# Multiplex neurodegeneration proteotoxicity platform reveals DNAJB6 promotes non-toxic FUS condensate gelation and inhibits neurotoxicity

Neurodegenerative disorders (NDDs) are a family of diseases that remain poorly treated despite their growing global health burden. To gain insight into the mechanisms and modulators of neurodegeneration, we developed a yeast-based multiplex genetic screening platform. Using this platform, 32 NDD-associated proteins are probed against a library of 132 molecular chaperones from both yeast and humans, and an unbiased set of ~900 human proteins. We identify both broadly active and specific modifiers of our various cellular models. To illustrate the translatability of this platform, we extensively characterize a potent hit from our screens, the human chaperone DNAJB6. We show that DNAJB6 modifies the toxicity and solubility of multiple amyotrophic lateral sclerosis and frontotemporal dementia (ALS/FTD)-linked RNA-binding proteins (RBPs). Biophysical examination of DNAJB6 demonstrated that it co-phase separates with, and alters the behavior of FUS containing condensates by locking them into a loose gel-like state which prevents their fibrilization. Domain mapping and a deep mutational scan of DNAJB6 revealed key residues required for its activity and identified variants with enhanced activity. Finally, we show that overexpression of DNAJB6 prevents motor neuron loss and the associated microglia activation in a mouse model of FUS-ALS.

Human neurodegenerative diseases (NDDs) are a major source of morbidity and mortality worldwide and represent a significant unmet medical need[1–3]. A hallmark of many NDDs is the intracellular accumulation of misfolded protein aggregates[4,5]. To model the proteotoxicity imposed by NDD-associated proteins with an intrinsic propensity to aggregate such as Fused in Sarcoma (FUS), TAR DNA-binding protein (TDP-43), and alpha-synuclein, researchers have repeatedly turned to the yeast, *Saccharomyces cerevisiae*[6–8]. In yeast, expression of these aggregation-prone proteins results in slow growth. By screening for genes that restore growth upon overexpression, researchers have been able to identify pathways involved in modulating the underlying proteotoxicity, and validate their findings in lower throughput mammalian cell systems. As a whole, these screens have informed our understanding of disease pathobiology, and served as the basis for numerous therapeutic intervention strategies, some of which are being advanced clinically[7–18].

While previous yeast-based studies have proven insightful, only a subset of NDD models have been screened, likely reflecting the difficulty involved in conduction large genetic screens. In addition, due to differences in the testing environment, genetic background, and method of screening, it is difficult to definitively compare results across studies in order to gain insight into broad versus narrow-acting regulators of proteotoxicity[7,8,18]. When comparisons between screens were performed, few or no shared modifiers of toxicity were identified, even among related disease proteins such as TDP-43 and FUS[7,8,10]. This lack of correlation conflicts with overlaps in clinical presentation

✉e-mail: ps2764@cumc.columbia.edu; alc042@health.ucsd.edu

between TDP-43 and FUS patients and the shared biochemical and biophysical properties of both proteins[19]. Furthermore, as previous approaches were only able to study one model at a time, they mainly focused on screening wild-type versions of NDD associated proteins, preventing our ability to assess if patient mutations alter the underlying molecular processes, which if found, would have significant implications in the treatment of this family of disorders[8,15]. Finally, most screens performed in yeast have searched for yeast genes that rescue the toxicity of NDD models, leading to identified hits with unclear or no known orthologous human counterpart to advance as a potential therapeutic candidate[8,10].

To overcome these limitations, we created a multiplex screening platform capable of simultaneously identifying genetic suppressors of 32 cell-based models expressing proteins that are implicated in neurodegeneration. To interpret the wealth of data obtained, we built a custom analysis pipeline that prioritizes interactions for subsequent validation. Using this platform, we were able to identify previously elusive, broadly active rescuers, along with highly selective rescuers that only impact a single model. These studies revealed numerous genetic modifiers for future investigation, while highlighting the diverse array of pathways and mechanisms that could exploited for therapeutic benefit.

Further examination of our results identified the human HSP40 co-chaperone, DNAJB6, as a potent rescuer of the toxicity caused by the expression of multiple RNA-binding proteins associated with ALS/FTD. Through subsequent studies in mammalian cells, we show that DNAJB6 can modulate the solubility of FUS, TDP-43, and heterogeneous nuclear ribonucleoprotein A1 (hnRNPA1). We also use purified proteins to demonstrate that DNAJB6 can phase separate and alter the liquid-liquid phase separation properties of FUS. We show that DNAJB6 is able to maintain FUS in a loose gel-like state that prevents its fibrilization over time. This mechanism is unique among modifiers of biologic condensates and suggests an additional mechanism by which chaperones prevent the aggregation of clients. Domain mapping and a deep mutational scan of DNAJB6 revealed key residues required for its activity and identified variants with enhanced activity. Finally, we demonstrate that AAV-based delivery of DNAJB6 rescues motor neuron loss and associated microgliosis in a mouse model of ALS-FUS, demonstrating its therapeutic potential in vivo.

## Results

### Development of a multiplexed screening strategy to identify rescuers of proteotoxicity

To enable our multiplex screening approach, we make use of isogenic yeast strains that each contain a unique DNA-barcode inserted into a neutral genomic locus[20]. Into each of these barcoded strains, we deliver a construct encoding an NDD-associated protein with a propensity to cause toxicity either through its misfolding or perturbation of core cellular processes (e.g., TDP-43, FUS, alpha-synuclein). Growth of these strains in media that induces the overexpression of the toxic disease-associated protein causes a reduction in cell proliferation and provides a facile method of modeling the dysfunction elicited by these proteins. As each of our models (i.e., yeast expressing a unique protein of interest) are linked to a particular DNA-barcode, we can combine them into a single mixed pool and track the growth of each member by measuring its barcode abundance using next-generation sequencing. To identify regulators of neurodegeneration, we probe the pool of disease models against a library of genetic modifiers (Supplementary Fig 1). In cases where a genetic modifier suppresses the toxicity of a particular NDD-associated protein, we observe a marked increase in the abundance of the model's barcode as compared to the control condition where the pool is exposed to an inert genetic modifier like mCherry. Taking advantage of the scalability afforded by the use of DNA-barcoding, each examined protein is placed into several different DNA-barcoded strains (i.e., redundant barcoding). This decreases

assay noise by allowing us to use the collective behaviors of all uniquely barcoded strains containing the same protein to derive our conclusions and enables us to confidently identify significant interactions between our disease models and genetic modifiers[21].

To develop our approach, we analyzed multiple screening parameters such as strategies for pooling DNA-barcoded strains based on their toxicity, appropriate number of experimental replicates, and the amount of redundant barcoding required to detect interactions with high sensitivity (Supplementary Note 1-3, Supplementary Fig 1–7). We then assembled a collection of NDD-associated proteins based on previous publications (Fig. 1a and Supplementary Data 1)[13,22–24]. For a subset of these proteins, we also engineered a panel of point mutants based on familial variants that increase the likelihood of disease to determine whether they might influence the observed rescue (Fig. 1a). To assist in interpreting screening results, the final pool of cells was supplemented with a series of DNA-barcoded controls including cells that express a non-toxic fluorescent protein (mCherry), other aggregation-prone proteins not associated with neurodegeneration (e.g., SUP35, RNQ1), or proteins linked to neurodegeneration but which themselves are not prone to misfolding (e.g., ANG, OPTN). In total, our final screening library contains 32 NDD-associated proteins plus multiple controls, each placed within 5–7 uniquely DNA-barcoded strains and mixed at equal ratios into a single pool for testing (For detailed descriptions of models and pool composition, see Supplementary Data 1 and Supplementary Note 3).

### Functional characterization of chaperone interactions with neurodegenerative disease models

Molecular chaperones have been previously implicated in the refolding, turnover, and mitigation of the toxicity of aggregation-prone proteins associated with neurodegeneration[5,25,26]. However, a comprehensive map of the functional interactions between chaperones and their disease-associated clients remains elusive, limiting the field's ability to identify broadly active members of this class of proteins. We hypothesized that the approach described here would be uniquely suited towards assessing this interaction space. In addition, it would also serve as a data-rich set to extensively validate our methodology. Using our final pool and optimized screening strategy, we probed a targeted library of 132 molecular chaperones, 62 of which were from yeast and 70 from humans. Overall, this screen represents 5850 genetic interactions between models in the pool and the corresponding library of molecular chaperones, including 4224 interactions directly relevant to human neurodegenerative disease models.

Within the yeast chaperone set, we observed 112 strong interactions that resulted in a greater than 0.5 log2 fold increase in barcode abundance (Fig. 1b, Supplementary Fig 8a, and Supplementary Data 2, 3). Notably, potentiated HSP104 chaperones were broadly active with the ability to rescue the proteotoxicity of a number of models including two beta-amyloid models, hnRNPA1, and the dipeptide repeat PR50 associated with C9orf72 RAN-translation, in addition to their previously reported rescue of TDP-43, FUS, and alpha-synuclein models[27]. Outside of strong interactions, such as those observed with the HSP104 variants, 74 interactions with mild-to-moderate positive log2 fold changes between 0.25 and 0.5 were also observed. The combined 186 interactions were prioritized for further analysis using an in-house pipeline designed to call interactions with statistically significant enrichment (Fig. 1c, and Supplementary Fig 8b). This pipeline uses the control wells tested against inert rescuers (e.g., mCherry) to develop an expectation for the abundance of each barcode in the pool, determines which barcodes significantly increase in abundance in test wells, and combines information from barcodes associated with the same model to identify the most potent and significant hits (see "Methods" for details). Upon application of our data analysis pipeline, 100 interactions were deemed significant

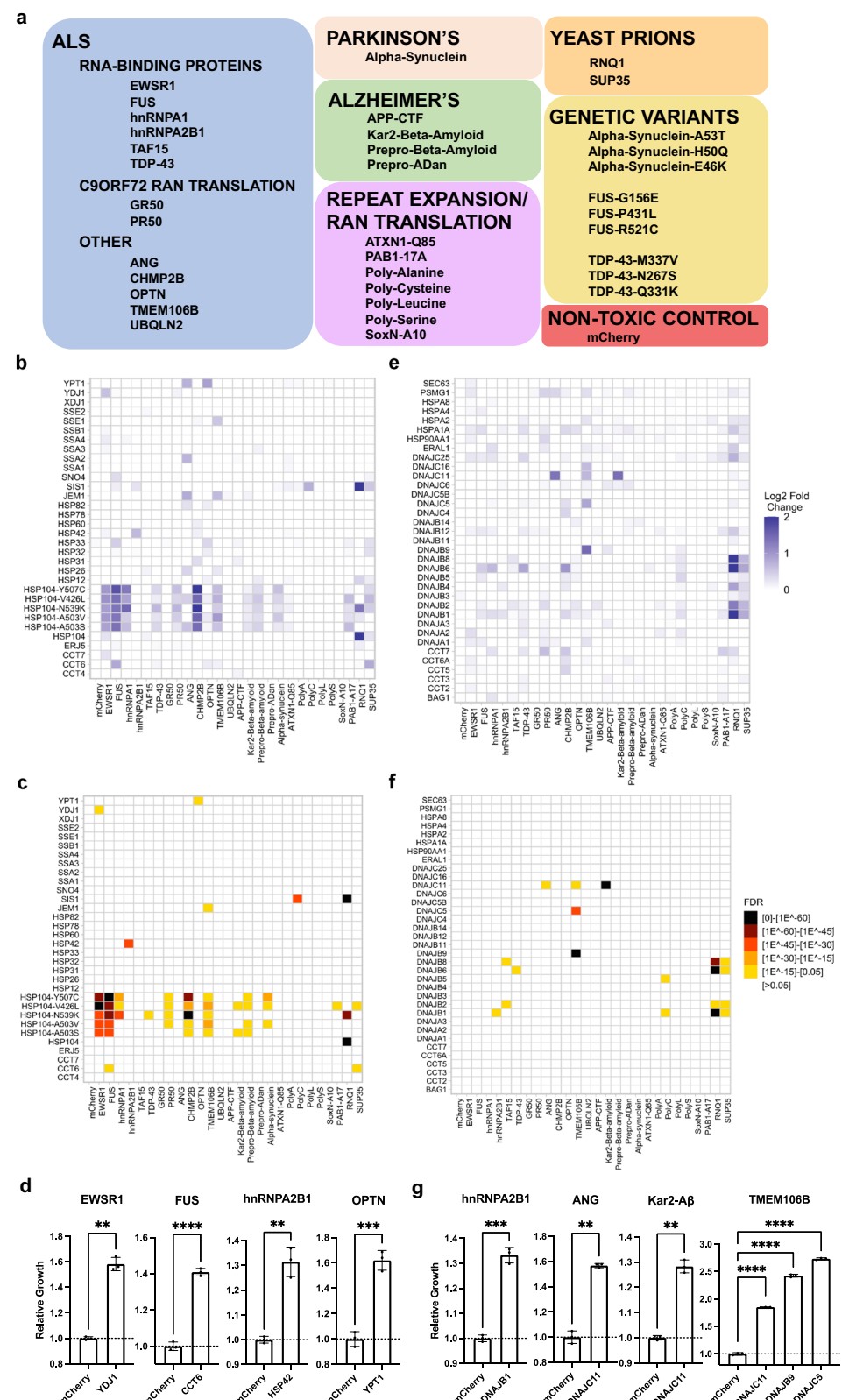

(Supplementary Data 2). Among these significant interactions, we identified specific interactions between the ALS/FTD-associated RNA-binding proteins, EWSR1, FUS, and hnRNAP2B1 and the type I HSP40 chaperone YDJ1, the chaperonin containing TCP-1 (CCT) subunit CCT6, and the small heat shock protein HSP42, respectively. We also observed an interaction between the Golgi-maintenance protein and autophagosomal receptor OPTN and YPT1, a Rab family GTPase

involved with ER-to-Golgi transport that localizes to pre-autophagosomal structures (Fig. 1d).

The mammalian chaperone set contained 131 interactions that resulted in mild-to-moderate or strong log2 fold changes in barcode abundance (Fig. 1e and Supplementary Fig. 9). However, only 45 of the 131 interactions led to strong changes in barcode abundance, possibly reflecting the suboptimal function of mammalian chaperones within

**Fig. 1 | Screening of molecular chaperones from yeast and humans for their ability to rescue the proteotoxicity of various neurodegenerative disease and protein misfolding models. a** Models included in screen and disease association. **b** Log2 fold change plotted for interactions between selected yeast chaperones and models. Log2 fold changes are shown as an average of all barcoded strains associated with that model. **c** Statistically significant interactions between selected yeast chaperones and models. **d** Validation of interactions between yeast chaperones and models. Data are shown as mean ± s.d. for three biological replicates. **P = 0.0017, ****P = < 0.0001, **P = 0.0091, ***P = 0.0006. **e** Log2 fold change

plotted for interactions between selected human chaperones and models. **f** Statistically significant interactions between selected human chaperones and models. **g** Validation of interactions between human chaperones and models. Data are shown as mean ± s.d. for three biological replicates. ***P = 0.0007, **P = 0.001, **P = 0.0011, ****P = < 0.0001, ****P = < 0.0001, ****P = 0.0001. Comparisons between two conditions were conducted with two-sided Welch's *t* tests while multiple comparisons were conducted with ordinary one-way ANOVA; **$P \le 0.01$, ***$P \le 0.001$, ****$P \le 0.0001$.

---

yeast cells (Fig. 1e and Supplementary Fig. 9a). Nevertheless, 20 significant interactions were identified, including rescue of two membrane associated proteotoxicity models, TMEM106B and Kar2-beta-amyloid, mediated by the primarily mitochondria-localized type III HSP40 chaperone, DNAJC11 (Fig. 1f-g and Supplementary Fig. 9b). These results are of interest as DNAJC11 mutant mice show prominent neuronal pathology with vacuolization of the endoplasmic reticulum and disruption to mitochondrial membranes[28]. Furthermore, both TMEM106b and beta-amyloid cell-based models have been associated with mitochondrial stress and dysfunction[23,29]. These findings suggest that DNAJC11 mediated rescue may relate to its ability to buffer against the effects of amylogenic proteins on mitochondrial function. The breadth of our screening also revealed a general trend of interaction between multiple human type II HSP40 chaperones (DNAJB1, DNAJB2, DNAJB4, DNAJB6, DNAJB8) and two yeast prions, RNQ1 and SUP35. These findings indicate that the misfolded intermediates produced by these two yeast prions may have similar properties, which is in agreement with data showing the ability of these two yeast prions to cross-seed each other's aggregation[30]. Furthermore, the fact that SIS1, the yeast orthologue of these human DNAJB proteins, shows prominent activity against RNQ1 and SUP35, suggests that this class of misfolded prion species may be evolutionarily conserved substrates for the DNAJB family of chaperones[31].

### Secondary validation of multiplexed screening results

To validate the interactions identified in the screen, we applied a testing paradigm similar to the screen that was conducted. Using a previously reported passaging-based growth assay, which we first verified against a set of known interactions (Supplementary Fig 10)[32]. We individually validated two sets of hits, namely, those judged as statistically significant by our analytical pipeline and those with positive log2 fold changes that did not reach our significance threshold but represent suspected interactions. Hits identified as statistically significant with an FDR adjusted *p*-value of <0.05 were validated at a high rate, with 116/120 (96.7%) of these interactions reproducing upon individual testing (Supplementary Data 4). We next verified a number of suspected interactions with positive log2 fold changes that did not meet statistical significance, with 59/95 (62.1%) of these suspected interactions showing rescue upon individual testing (Supplementary Data 4). This indicates that the hit-calling algorithm can prioritize interactions that are likely to validate compared to using log2 fold changes alone, although some true hits may be missed as these likely do not survive our adjustments for multiple hypothesis testing. To estimate the sensitivity and specificity of the statistical pipeline, we assumed that the 175 validated interactions (116 significant and 59 suspected) represent most of the hits in the matrix of 5850 interactions. Holding this to be the case, it suggests that our screening platform has an estimated sensitivity and specificity of ~66% and ~99%, respectively, which is on par with or better than previously reported one-model-at-a-time screening approaches[33–36]. To ensure that the pipeline did not miss interactions that resulted in a growth rate increase, we manually tested 77 interactions predicted not to cause a growth rescue across 7 models and found that 0/77 (0%) resulted in a growth rescue (Supplementary Fig. 11). Furthermore, highlighting the power of our redundant barcoding strategy, if the number of barcodes

analyzed for each model is progressively reduced from 5 to 7 redundant barcodes per model to 1 per model, a striking decrease in the number of hits captured is seen with each barcode removed (Supplementary Fig. 12). These results reinforce the dramatic improvement in data quality afforded by transforming each model into multiple DNA-barcoded strains and analyzing their collective behavior.

### A large, unbiased screen of human ORFs identifies interactions

To evaluate the performance of our platform against a larger, unbiased library of potential modifiers, we screened the hORFeome V8.1 human cDNA library against our pool. We chose to screen human ORFs not only to identify hits that are more likely to translate directly to mammalian models of disease, but also because few human genes had been tested before in this paradigm, suggesting that many interactions remain uncovered. Screening 899 members of the hORFeome library enabled the examination of 40,455 interactions, representing ~9 times more interactions surveyed within our system than is typical[7,8,11,14,15]. Screening and subsequent validation of this collection of rescuers resulted in the identification of 54 confirmed genetic interactions (Supplementary Data 5–7). As expected with a library comprising random human genes, the occurrence of genetic interactions was significantly lower (0.14%) compared to the curated molecular chaperone screen (2.9%). Furthermore, when compared to other unbiased screens overexpressing yeast ORFs instead of human genes our hit rate remains lower (previously reported hit rates of 0.24–1.13% vs. 0.14%)[8,10,11,14]. One of the main drivers of this difference is likely the failure of human genes to function in yeast cells, highlighting the benefits of using a rapid high-throughput, quantitative screening approach to explore this sparsely populated interaction space[37].

Despite a lower rate of interactions, rescuers were identified for a broad range of yeast models, including ones that did not have rescuers from the initial molecular chaperone screen (Supplementary Fig. 13a). Several hits captured known interactions. For example, two 14-3-3 proteins, YWHAG and YWHAB were found to be rescuers of the ATXN1 polyglutamine (ATXN1-Q85) yeast model (Supplementary Fig. 13b). Cytoplasmic 14-3-3 proteins bind ATXN1 via a motif centered around a phosphorylated serine at position 776 in ATXN1. This binding masks the ATXN1 NLS and diminishes its transport into the nucleus which is necessary for ATXN1 toxicity[38,39]. Other rescuers of ATXN1-Q85 in our screen were RNPS1 and JMJD6, two proteins involved in RNA-splicing, which ATXN1 also actively participates in (Supplementary Fig. 13b). Furthermore, both RNPS1 and JMJD6 were recently identified as interactors of ATXN1-Q85 via pulldown and BIO-ID approaches, respectively[40]. We also identified two rescuers of the alpha-synuclein expressing model, one of which is a member of the Regulators of G protein signaling (RGS) family, RGS20 (Supplementary Fig. 13c). RGS family proteins are structural proteins that accelerate the GTPase activity of α-subunits in G coupled-protein receptors, reducing the duration of downstream signaling[41]. A number of RGS family proteins have been implicated in the development of Parkinson's Disease including a study of RGS6 in which expression of RGS6 in substantia nigra pars compacta dopaminergic neurons suppressed Parkinson's Disease phenotypes in aged mice[42]. Within our larger screen, we also discovered several rescuers of the FUS yeast model (Supplementary Fig. 13d). These rescuers included RAD23B, a ubiquitin-binding

proteasome-shuttle protein localized to the nucleus that is suspected to undergo phase separation, HIST1H1E, a member of the H1 linker histone family with predicted disordered regions, TGIF2LX, a DNA-binding homeobox protein with predicted disordered regions, and OAS1, a stress granule localized 2′5′-oligoadenylate synthetase protein. TMEM106B, a protein implicated in FTD[43–46] and brain aging[47,48] was strongly rescued by COX7B, a member of the cytochrome c oxidase complex involved in the mitochondrial respiratory chain (Supplementary Fig. 13e)[43,47]. Taken together, these results suggest that human proteins beyond molecular chaperones can be screened to identify interactions.

## DNAJB6 is a rescuer of multiple RNA-binding proteins implicated in ALS/FTD

During screening and subsequent validation, we identified a human chaperone, DNAJB6 that rescued the toxicity of cells expressing ALS/FTD-associated aggregation-prone RNA-binding proteins FUS, TDP-43, and hnRNPA1 (Fig. 2a). DNAJB6 is a type II HSP40 co-chaperone expressed in several tissues, including ubiquitously throughout the brain and spinal cord[49]. However, DNAJB6 expression is decreased in the brain with aging potentially sensitizing neurons to misfolded protein stress[50,51]. HSP40 co-chaperones comprise a large class of approximately 50 proteins in humans with an array of reported activities. However, they primarily function by binding to misfolded proteins, trafficking them to HSP70 co-chaperones, and regulating protein-protein interactions[52,53]. While DNAJB6 was the only type II HSP40 that was able to rescue TDP-43 toxicity, DNAJB1 and DNAJB2 could rescue the FUS and hnRNPA1 models, although their magnitude of rescue was ~1/3 that of DNAJB6, suggesting DNAJB6 has properties that are unique among its family (Supplementary Fig. 14). DNAJB6 has been previously shown to suppress the aggregation of polyglutamine repeat containing proteins[54–57]. In addition, mutations in DNAJB6 cause Limb Girdle Muscular Dystrophy D1 (LGMDD1), in which affected muscle tissue accumulates TDP-43 aggregates, suggesting it plays a role in the maintenance of TDP-43 solubility within humans[58,59]. Nonetheless, it remains unclear as to whether DNAJB6 is a broadly active modifier of other disease relevant clients and how DNAJB6 modulates misfolding[60,61].

To determine if the interaction between DNAJB6 and the RNA-binding proteins FUS, TDP-43, and hnRNPA1 are relevant in mammalian context, we employed an in vivo protein aggregation assay[60,62]. In this assay, when FUS, TDP-43, or hnRNPA1 are overexpressed within human embryonic kidney 293 T (HEK293T) cells, they form SDS-insoluble species (RIPA buffer insoluble), that can be solubilized in urea. As compared to control cells co-transfected with enhanced yellow fluorescent protein (EYFP), cells co-transfected with DNAJB6 contain less SDS-insoluble FUS, TDP-43, and hnRNPA1 (Fig. 2b, e and Supplementary Fig. 15). This result demonstrates that DNAJB6 can reduce the formation of insoluble species for multiple RNA-binding proteins within mammalian cells.

## Endogenous DNAJB6 participates in the response to accumulation of insoluble FUS, TDP-43, and hnRNPA1

To investigate further the role—and mechanisms by which—DNAJB6 responds to increasing cellular concentration of ALS/FTD-associated RNA-binding proteins, we transfected HEK293T cells with EYFP, FUS, or TDP-43 expression constructs and performed unbiased RNA-sequencing. As compared to the EYFP control condition, DNAJB6 was among the most strongly and significantly upregulated chaperones within the ~270 molecular chaperones observed upon overexpression of FUS and TDP-43. However, only overexpression of TDP-43 resulted in an increased amount of DNAJB6 protein at the same time point (Supplementary Fig. 16, 17 and Supplementary Data 8).

To determine what role endogenous levels of DNAJB6 play in regulating FUS, TDP-43 and hnRNPA1 misfolding, Cas9 was used to generate multiple independent HEK293T cell clones in which DNAJB6 was knocked out (Fig. 2f). In DNAJB6 knockout lines, SDS-insoluble FUS, TDP-43, or hnRNPA1 species were not observed when these proteins were endogenously expressed, and we did not observe significant changes in the levels of soluble forms off these proteins (Supplementary Fig. 18). However, upon overexpression of FUS, TDP-43, or hnRNPA1, greater amounts of SDS-insoluble species within DNAJB6 knockout lines were observed compared to the non-targeting gRNA control (NTC) lines (Fig. 2g-j and Supplementary Fig. 19). Soluble levels of FUS, TDP-43, and hnRNPA1 were not different between the DNAJB6 KO and NTC lines upon overexpression of each RBP.

Taken together, these data suggest that DNAJB6 is part of a programmed cellular response to rising levels of multiple aggregation-prone RNA-binding proteins and that physiological levels of DNAJB6 can regulate the solubility of these proteins.

## Endogenous DNAJB6 colocalizes with endogenous FUS in selected FUS ribonucleoprotein (RNP) granules in projections of human motor neurons

To confirm an interaction between FUS and DNAJB6 we undertook both co-immunoprecipitation and co-localization experiments. Co-immunoprecipitation experiments in HEK293T cells failed to capture an interaction between exogenously expressed FUS with DNAJB6, although interactions with TDP-43 and hnRNAP1 were captured (Supplementary Fig. 20). In contrast, co-localization experiments in disease relevant human motor neurons derived from induced pluripotent stem cells (hiPSCs) expressing endogenous FUS and DNAJB6 at physiological levels, revealed that a small percentage (~4%) of FUS ribonucleoprotein (RNP) granules contain co-localizing DNAJB6 (Fig. 3 and Supplementary Data 9). Specifically, we used Airyscan immunofluorescence microscopy (implementing the Airyscan2 module to improve resolution from 240 nm to 125 nm) to visualize endogenous FUS and DNAJB6 in the nucleus, cytoplasm and neurites of two independent iPSC-derived human motor neuron lines. We determined the location of each RNP as the peak (maximum) of the fluorescent intensity distribution (Gaussian distribution) emanating from each FUS+ or DNAJB6+ fluorescent particle (see methods). FUS RNPs with a peak max within 71 nm (2 pixels) of a DNAJB6 RNP peak max across ≥ 4 consecutive, 150 nm Z-stack sections were scored as colocalized. We observed that in neurites, $3.94 \pm 1.06$ percent of FUS-positive granules co-localized with DNAJB6-positive granules (mean and standard deviation, respectively; $n = 407$ granules in five Z-stacks, and 341 granules in four Z-stacks in the two independent wild type hiPSC-derived motor neuron lines). This overlap was significantly greater than random chance ($0.55 \pm 0.07\%$, mean and standard deviation; right-tailed two-sample $t$ test; $p = 0.025$). We interpret these results to mean that FUS and DNAJB6 do indeed co-exist within cells. However, these interactions are likely to be transient and/or weak and do not survive our immunoprecipitation conditions.

## DNAJB6 undergoes phase separation and can prevent aberrant FUS condensation in vitro

DNAJB6 contains multiple domains with low amino acid complexity. This feature, which is a characteristic of proteins that undergo liquid-liquid phase separation (LLPS), prompted us to investigate whether DNAJB6 might itself undergo LLPS[63–65]. To address this question, we purified recombinant human DNAJB6 from insect cells. We then incubated near physiological intracellular concentrations of purified DNAJB6 ($0.25\,\mu M$–$3.00\,\mu M$ as determined from the literature) in 50 mM NaCl. Under these conditions, DNAJB6 forms liquid–liquid phase-separated droplets, thereby confirming our hypothesis (Supplementary Fig. 21a-b)[62,66,67].

Next, to determine whether DNAJB6 might co-partition with phase separating client proteins such as FUS, we mixed FUS and DNAJB6 at concentrations approaching those physiologically found in

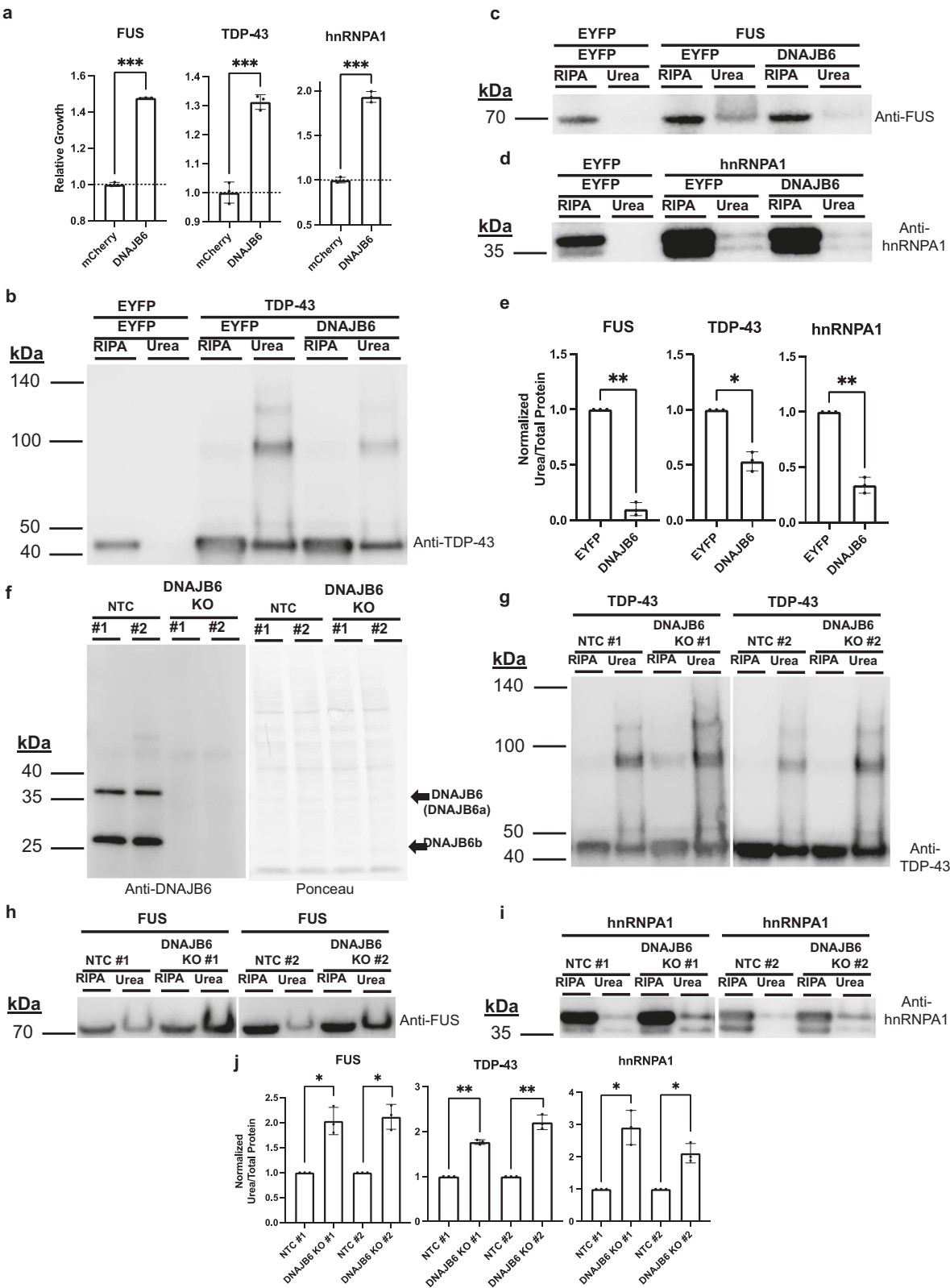

cells as determined from the literature (FUS: 1.5 μM; DNAJB6: 0.25 μM) in 50 mM NaCl[68]. Under these conditions, as well as at a 1:1 molecular ratio (not shown), DNAJB6 and FUS co-partition into liquid–liquid phase-separated droplets (Supplementary Fig. 21c). Intriguingly these FUS + DNAJB6 droplets are both more numerous and noticeably smaller than the FUS-only condensates (Fig. 4a, b). Furthermore, they remain small and spherical even after 48 h incubation. In contrast, the

FUS only condensates initially form large spherical condensates, but become increasingly irregular in shape and by 36–48 h, begin to form aged fibrillary aggregates (Fig. 4a, b). This observation suggests that DNAJB6 might prevent progression of FUS-FUS interactions that drive time-dependent solidification of FUS condensates.

To discover the biophysical basis of this effect, we next performed a series of orthogonal biophysical experiments starting with

**Fig. 2 | DNAJB6 is a rescuer of FUS, TDP-43, and hnRNPA1. a** DNAJB6 rescues proteotoxicity of FUS, TDP-43, and hnRNPA1 in yeast. Data are shown as mean ± s.d. for three biological replicates. ***$P$ = 0.0001, ***$P$ = 0.0005, ***$P$ = 0.0002. **b–d** Overexpression of FUS, TDP-43, and hnRNPA1 in HEK293T cells results in formation of SDS-insoluble (RIPA buffer insoluble), urea-soluble species. The formation of these species can be reduced by co-expression of DNAJB6. **e** Quantification of the urea soluble species normalized to total protein detected by Ponceau S staining in (**b**–**d**). For TDP-43 blots, all bands at or above 43 kDa were included in quantification. Each independent replicate was normalized to its associated EYFP rescued sample. Data are shown as mean ± s.d. for three biological replicates. **$P$ = 0.0014,

*$P$ = 0.0114, **$P$ = 0.0039. **f** Validation of non-target control (NTC) and DNAJB6 KO lines. DNAJB6 has two isoforms; DNAJB6a and DNAJB6b, which are 36 kDa and 27 kDa in size, respectively. **g–i** Overexpression of FUS, TDP-43, and hnRNPA1 in DNAJB6 KO lines shows an increase in the propensity to form SDS-insoluble, urea-soluble species. **j** Quantification of the urea soluble species normalized to total protein detected by Ponceau S staining in (**g**–**j**). Each independent replicate was normalized its associated EYFP rescued sample. All data are shown as mean ± s.d. for three biological replicates. *$P$ = 0.0225, *$P$ = 0.0162, **$P$ = 0.0014, **$P$ = 0.006, *$P$ = 0.0249, *$P$ = 0.0231. All statistical tests were conducted with two-sided Welch's $t$ tests; ns not significant $P$ > 0.05, *$P$ ≤ 0.05, **$P$ ≤ 0.01, ***$P$ ≤ 0.001, ****$P$ ≤ 0.0001.

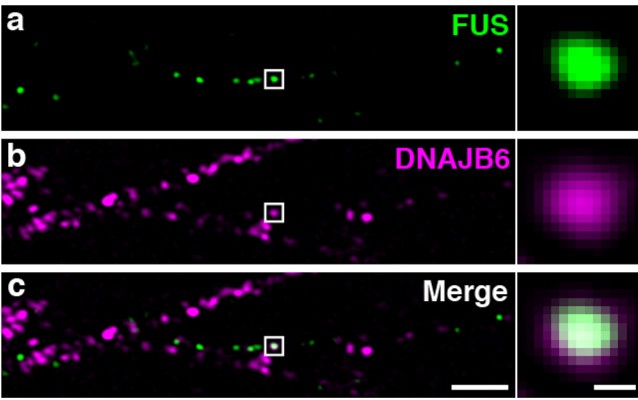

**Fig. 3 | FUS and DNAJB6 coincide in a subset of neuritic motor neuron granules.** Immunostain of (**a**). FUS, **b** DNAJB6, **c** FUS and DNAJB6 merge, in hiPSC human differentiated motor neurons. Left panel: A single Airyscan Z-stack section is shown, out of three 150 nm sections with continuous overlap between FUS and DNAJB6 (scale bar = 2 μm). Right panel: magnification of a FUS granule that co-labels with DNAJB6 (scale bar = 0.176 μm). Data shown is one of three biological replicates.

Fluorescence Recovery after Photobleaching (FRAP). FRAP measures the dynamic exchange of tagged FUS molecules between the dilute solute phase and the concentrated condensed phase. In freshly prepared (30 min) condensates, the fluorescence recovery rates of FUS + DNAJB6 condensates are significantly faster than FUS-only condensates (Supplementary Fig. 21d). However, by 24 h, there is little fluorescence recovery in either FUS-only or FUS + DNAJB6 condensates after photobleaching (data not shown). This result reflects a clear impact of DNAJB6 on the dynamics of the early stages of FUS condensation. However, FRAP is a relatively crude assay of condensation biophysics, and does not provide insight into the packing state or the chemical state of FUS in these condensates. Consequently, we next applied Fluorescence Lifetime Imaging Microscopy (FLIM) to our samples. FLIM measures the average duration that a fluorophore resides in the excited state before emitting a photon and, by that, can provide insights into the fluorophore's environment and dynamics. Using FLIM, we have recently shown that the drop in the fluorescence lifetime of fluorescent protein labeled FUS directly correlates with an increase in viscosity of FUS condensates[69]. Here, using mEmerald instead of GFP, we show that FUS-only preparations cause a progressive and time-dependent reduction in fluorescence lifetimes (Fig. 4c). This indicates that, over time, FUS polymers become increasingly closely packed and more viscous in the condensates—a result that is fully congruent with the microscopic observation that FUS-only preparations eventually harden into fibrillar aggregates. In contrast, from 20 min to 48 h, the FUS + DNAJB6 condensates continue to demonstrate significantly longer fluorescence lifetimes that are similar to those of control experiments containing dispersed solutions of DNAJB6 and mEmerald (Emerald does not form condensates;

Fig. 4c, Supplementary Fig. 21e–g). Accordingly, these FLIM experiments suggest that DNAJB6 permits FUS condensation to occur, but limits close homotypic FUS-FUS interactions, and prevents further condensation (packing) into dense fibrillar condensates. We interpret the FRAP and FLIM experiments to indicate that in both FUS-only and in FUS + DNAJB6 condensates, the various polymer interactions create networks that reduce the ability of FUS to exchange with the dilute solute phase. However, in the case of FUS-only condensates, the unfettered FUS-FUS interactions drive hardening of condensation to the point where FUS polymers are increasingly more viscous, which leads to a decrease in the fluorescence lifetime. In contrast, in FUS + DNAJB6 condensates the FUS-DNAJB6 interaction prevents this closer packing (allowing persistently longer fluorescence lifetimes).

To understand the biophysical mechanism by which DNAJB6 alters FUS condensation, we next used a tie line gradient analysis approach[70]. This method characterizes the collective interactions between FUS and DNAJB6 in the condensed phase[70] and does so by examining changes in the concentration of FUS in the dilute phase after phase separation. We apply samples containing mEmerald-tagged FUS to a confocal single photon detection unit using a microfluidic flow cell. We then record time traces as the sample flows under the detector (Supplementary Fig. 21h). The resulting times traces display a stable baseline intensity (corresponding to the FUS dilute phase concentration) with short bursts in intensity (caused by condensates passing through the confocal volume) (Fig. 4d, left panel). By performing these measurements under varying DNAJB6 concentrations (0-4 μM), we find that the addition of DNAJB6 significantly decreases the dilute phase concentration of FUS (Fig. 4d, middle panel), and does so in a FUS and DNAJB6 concentration-dependent manner even at nanomolar concentrations. An initial conclusion to be drawn from this experiment is that DNAJB6 enhances FUS phase separation because a larger fraction of FUS is converted into the condensed state.

Next, by identifying a set of points with different total DNAJB6 and FUS concentrations but equivalent FUS dilute phase concentrations we can characterize the tie line gradient between DNAJB6 and FUS (see Methods). We determined this gradient to be 3.0 ± 1.2 μM DNAJB6 / μM FUS. This result indicates that the dense phase concentration of DNAJB6 exceeds its dilute phase concentration. Thus, DNAJB6 displays preferential partitioning into FUS condensates (dense phase) at comparable concentrations with an attractive interaction with FUS in the condensed phase[63]. This result indicates that DNAJB6 prevents the overall liquid-to-solid transition of FUS by replacing on-pathway homotypic FUS-FUS interactions with preferential heterotypic FUS-DNAJB6 interactions (Fig. 4d, right panel). Hence, despite the overall increase in the phase separation propensity of FUS, the FUS molecules interact with DNAJB6 rather than themselves, which preserves their liquid / gel-like state. This conclusion is fully congruent with the conclusions drawn from the FLIM experiments.

Classical amyloidophyllic dyes such as Thioflavin T (ThT) do not bind well to pathological FUS aggregates. In contrast, the oligothiophene dye pentameric formyl thiophene acetic acid (pFTAA)[71,72] does bind to pathologically condensed gels and fibrillar aggregates of FUS and tau, and does so in both biochemical and cellular preparations[62,66].

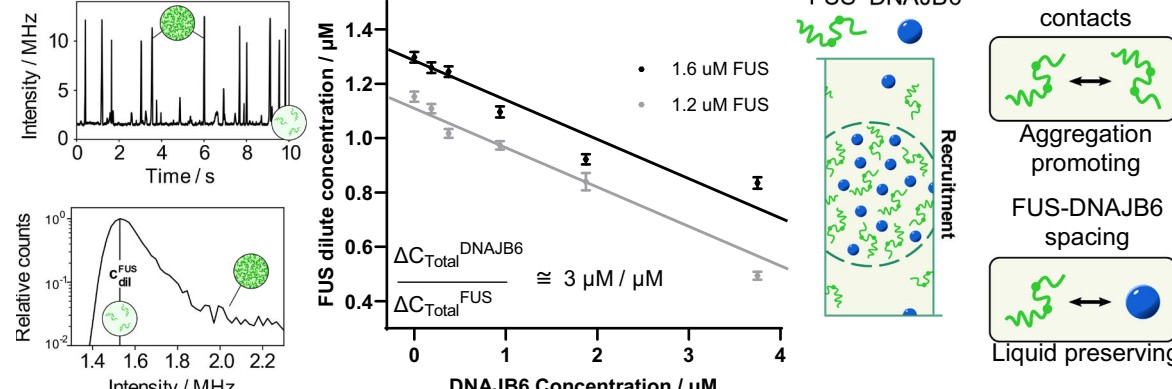

This property appears to relate to the high affinity of pFTAA for repeated crossed-β sheet structures. To evaluate the impact of DNAJB6 on pFTAA-positive crossed-β sheet content in hardened (also termed aged) FUS condensates, we compared the time-dependent changes in pFTAA fluorescence in biochemical preparations of FUS-only versus FUS + DNAJB6 condensates as they age over 48 h (Fig. 5a and Supplementary Fig. 22a). Over time, FUS-only condensates increasingly form large, non-spherical shapes with strong pFTAA fluorescence. Many of these hardened / aged FUS-only condensates display strongly pFTAA-labeled foci dispersed within the aggregates. In contrast, over the same time interval, FUS + DNAJB6 condensates remain smaller with significantly weaker pFTAA signals (Fig. 5a).

To further characterize both the conformational state and the nanomechanical and material properties of FUS-only and FUS +

**Fig. 4 | Biophysical characterization of FUS condensates with and without DNAJB6. a** DNAJB6 promotes formation of smaller spherical condensates that are stable during aging, and prevent formation of dense irregular spiculated / fibrillary condensates. FUS and FUS + DNAJB6 condensates at 20 min, 24 h and 36-48 h. FUS condensates form fibrillar aggregates when incubated for 36 h. Scale bar = 10 μm at 20 min and 24 h, scale bar = 20 μm at 36 h. Data were obtained using epi-fluorescence microscopy. **b** Quantification of condensate size. One micron squared minimum size cutoff was used when quantifying condensates. Average condensate size was determined for multiple condensates within several different fields of view. For FUS, $n = 343$ (20 min), 792 (24 h); for FUS + DNAJB6, $n = 1455$ (20 min), 1459 (24 h). ****$P = < 0.0001$, ****$P = < 0.0001$. Comparisons were conducted with two-sided Welch's *t* test. ****$P \leq 0.0001$ Plots are shown as mean ± s.e.m. **c** Fluorescence lifetime measurements (FLIM) of mEmerald-FUS condensates ± DNAJB6 over time. The condensate-averaged fluorescence lifetime of mEmerald fluorophores linked to FUS significantly decreases over time, indicating FUS condensation and aggregation over time. However, in the presence of DNAJB6, FUS aggregation is prevented, as shown by the significantly higher fluorescence lifetime detected with DNAJB6. For mEmerald-FUS, $n = 86$ (20 min), 100 (2 h), 100 (24 h), 100 (48 h); for

mEmrald-FUS + DNAJB6, $n = 100$ (20 min), 100 (2 h), 100 (24 h), 100 (48 h). Data are shown as mean ± s.d. for three biological replicates. *$P = 0.0310$, ****$P = < 0.0001$, ****$P = < 0.0001$, ****$P = < 0.0001$, ****$P = < 0.0001$, ****$P = < 0.0001$, ****$P = < 0.0001$, ****$P = < 0.0001$, ****$P = < 0.0001$. Non-parametric Kruskal-Wallis test with Dunn's multiple comparison where * is $P < 0.05$ and **** is $P < 0.0001$. Error bars denote ± s.d. **d** Tie line gradient analysis reveals preferential partitioning and interaction between DNAJB6 and FUS. (left panel) Intensity time trace of a phase separated FUS sample upon flowing through a confocal detection set-up using microfluidic technology (top) and corresponding intensity histogram (bottom). Bursts of high intensity correspond to condensates passing through the confocal volume. The baseline signal forms a peak in the intensity histogram and is used to extract dilute phase concentrations. For details, see "Methods". (middle panel) Dilute phase concentration changes of FUS with varying DNAJB6 concentrations allowing for determination of the tie line gradient value between FUS and DNAJB6 at the measured conditions. Data are presented as mean ± s.d., parameter errors are estimated by fitting to concentration intensity histograms obtained from the raw signal trace. (right panel) DNAJB6 is recruited into FUS condensates, effectively spacing out aggregation promoting FUS-FUS interactions to induce liquid state preservation.

DNAJB6 condensates, we next applied atomic force microscopy (AFM) and both bulk Fourier Transform Infrared spectroscopy (FTIR) and single molecule infrared nanospectroscopy (AFM-IR). These methods require surface deposition, which could potentially disrupt the structural and conformational properties of condensates, confounding results[73]. To circumvent this problem, we employed microfluidic spray deposition which, by minimizing the time for surface-induced structural deformations, retains relevant structural features of condensates on a solid support[74,75].

AFM allows direct measurement of the material properties of condensates. The elastic response of condensates can then be quantified via the acquisition of force-distance curves that depict how a sample responds to local deformations by the AFM tip[76,77]. Because the elastic response to deformation is a characteristic behavior of solid materials, the Young's modulus reports on the emergence of solid-like behavior within ageing condensates. FUS-only condensates display a time-dependent increase in the Young's modulus (Fig. 5b). Thus, at 2 h, the Young's modulus of FUS-only condensates is ~9 kPa at 2 h, which is typical of aged fluids[78–80]. However, at 48 h, the Young's modulus of FUS-only condensates increases to ~25,000 kPa, which is consistent with the emergence of solid-like behavior[78]. In contrast, FUS + DNAJB6 condensates initially display a higher elastic response of ~31 kPa at 2 h, which then remains essentially unchanged up to 48 h (~40 kPa) (Fig. 5b). Thus, on the timescale explored, partitioning of DNAJB6 into the FUS condensates prevents hardening / ageing into solid-like assemblies—a conclusion that is also in good agreement with the FLIM and tie-line experiments.

We next wished to understand whether DNAJB6 altered the ageing behavior of FUS condensates by inducing changes in the secondary structure and conformation of the component polypeptide chains as a function of time. To accomplish this, we initially turned to bulk FTIR. Bulk FTIR of FUS-only condensates reveals that the dominant peak at 2 h is from random coil/α-helical conformations (1657 cm$^{-1}$)[81]. After 24 h of ageing, the intensity of this peak decreases, and is accompanied by the appearance of a peak corresponding to intermolecular β-sheet at 1626 cm$^{-1}$ (Fig. 5c). This peak further increases in intensity at 48 h (Fig. 5c). These results suggest a gradual transition from a random coil to a solid intermolecular β-sheet conformation[75]. The same experiment on FUS + DNAJB6 condensates reveals two dominant peaks at 2 h. One reflects random coil/α-helix (1655 cm$^{-1}$)[75,81] (Fig. 5c). The other reflects intermolecular, parallel β-sheet (1626 cm$^{-1}$)[75,81], which did not further increase in intensity up to 48 h (Fig. 5c). These FTIR results are consistent with DNAJB6 stabilizing the presence of gel-like condensates that cannot further develop to solid-like states. Indeed, in the presence of DNAJB6, most of the FUS polypeptide chains remain in a random

coil conformation as indicated by the lack of change in the peak intensity at ~1655 cm$^{-1}$. Interestingly, we observed a ~2 cm$^{-1}$ downward shift of the random coil peak after 48 h of ageing, which we attribute to a decrease of electron density in the carbonyl bond of the peptide backbone[82]. As a control, we confirmed that the secondary structure changes seen in the FUS + DNAJB6 condition were not due to DNAJB6 alone (Supplementary Fig. 22b, c). These FTIR results suggest that DNAJB6 alters the chemical environment and the interaction network between disordered FUS polypeptide chains. These results are consistent with both the FLIM and tie line results.

We next focused on understanding how DNAJB6 affects the conformational state of FUS and FUS + DNAJB6 gelled condensates. To accomplish this, we performed nano-chemical imaging and spectroscopy via infrared nanospectroscopy (AFM-IR, Fig. 6 and Supplementary Fig 23). This technique first maps the 3D morphology, vibrational IR absorption and stiffness of individual condensates (Fig. 6a and Supplementary Fig. 23a–c) and then acquires nano-localised IR spectra allowing the measurement of their secondary structure (Fig. 6b and Supplementary Fig. 23d-g), thereby circumventing the averaging effects of heterogeneity amongst condensates and fibrillary aggregates present in bulk FTIR measurements. While we observe the rare occurrence of fibrillar aggregates with increased intermolecular β-sheet content (Supplementary Fig 23h-j), we focused here on the less characterized and more abundant condensates at the initial time-points of formation (0-20 min). These single-condensate experiments confirmed that, compared to FUS-only condensates, the FUS + DNAJB6 condensates contain more intermolecular parallel β-sheet and less intermolecular antiparallel β-sheet (Fig. 6b, c). Furthermore, the parallel β-sheet content of FUS + DNAJB6 shifted at a lower wavenumber, indicating the presence of more extended strands and more hydrogen bonding[83] (Fig. 6c). These features have previously been associated with formation of gels[66], and are consistent with the results from bulk FTIR experiments. Taken together, the bulk FTIR and IR spectroscopy experiments support the notion that DNAJB6 alters the interactions between component FUS polypeptide chains within the condensates and thereby prevents their complete structural conversion into solid-like assemblies.

## Characterization of DNAJB6 via domain deletions and deep mutational scanning

Having established the ability of DNAJB6 to phase separate and directly interact with its clients to prevent their misfolding, we sought to decipher the regions within the protein required for its effect within cells. A series of DNAJB6 deletion mutants were constructed. These include versions lacking the J-domain which is necessary for activation

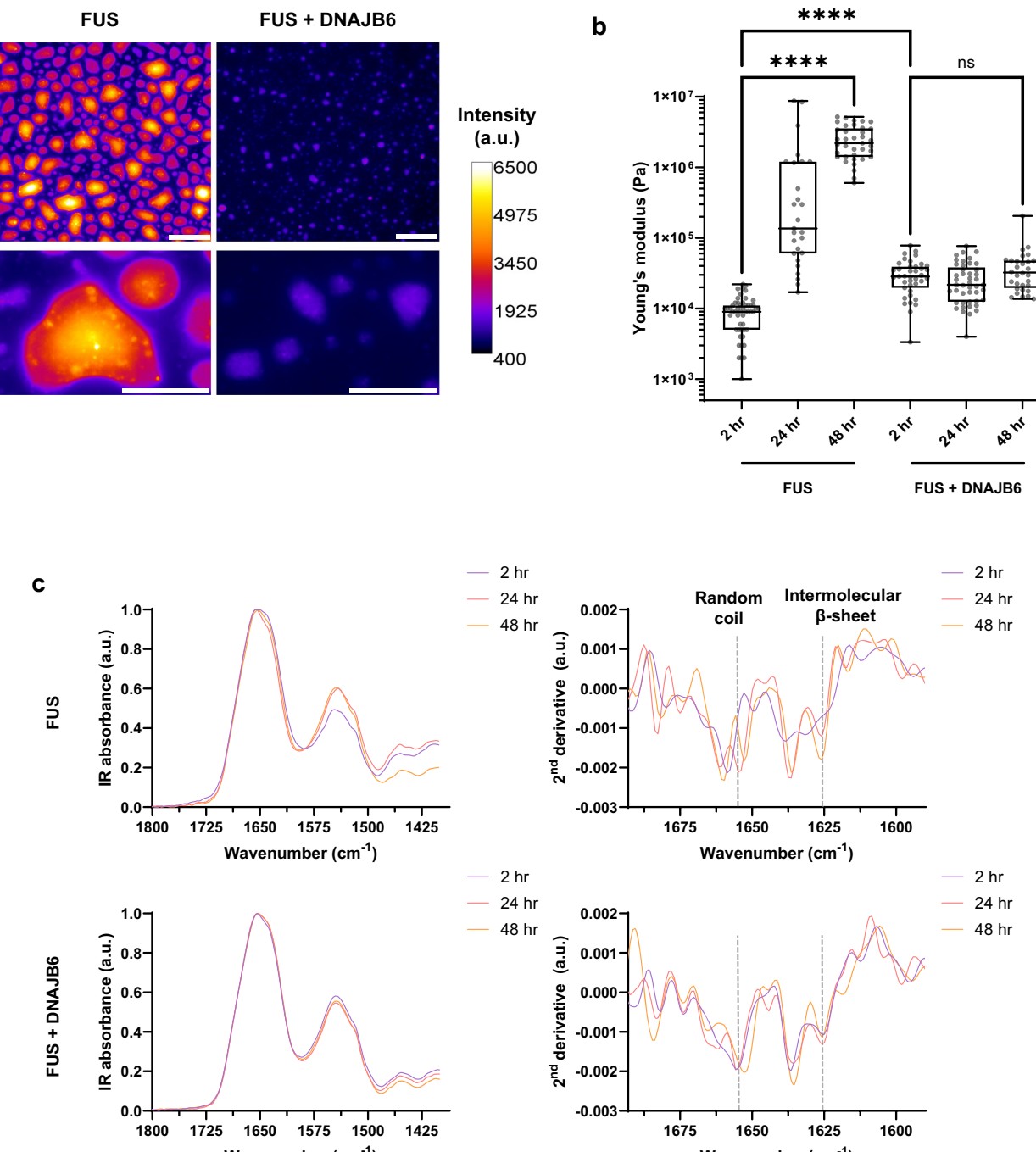

**Fig. 5 | pFTAA staining, AFM, and FTIR studies of the effect of DNAJB6 on FUS condensates. a** Aged FUS condensates have higher content of pFTAA-positive condensates than FUS with DNAJB6. pFTAA staining of mCherry-FUS and mCherry-FUS with DNAJB6 after 48 h, with corresponding zoom-in regions below. Scale bars correspond to 25 μm for top panels and 10 μm for bottom zoomed-in panels. a.u. is arbitrary units. **b**, **c** Complementary AFM and FTIR studies reveal that DNAJB6 has significant effects on the nanomechanical properties of aging FUS condensates but has modest effects on bulk IR spectral signatures. **b** Measurements of the Young's (elastic) modulus of single condensates by AFM reveal that DNAJB6 prevents the progressive age-associated increase in elastic modulus in FUS only preparations. FUS only condensates undergo liquid-to-solid transitions characterized by a progressive increase in elastic modulus. In contrast, FUS + DNAJB6 condensates at 2 h have a higher elastic modulus than FUS only (consistent with the presence of β-sheet content measured via FTIR), but there is then no significant increase in Young's modulus values at 48 h. All assays used 2 mM FUS, 0.34 mM DNAJB6,

50 mM NaCl, Brown-Forsythe and Welch ANOVA with Dunnett's T3 test for multiple comparisons. ****$P$ = < 0.0001, ****$P$ = < 0.0001. ns not significant, **** $p < 0.0001$ FUS $n = 51$ (2 h), 27 (24 h), 37 (48 h); FUS + DNAJB6 $n = 39$ (2 h), 45 (24 h), 32 (48 h). Box plots show the median (center line), the interquartile range (IQR; box bounds represent the 25th and 75th percentiles), and minimum and maximum values (whiskers). **c** Bulk FTIR spectra (left) and the corresponding second derivatives (right) were obtained at 2, 24, and 48 h. This reveals that initially random coil FUS only condensates undergo a gradual increase in intermolecular β-sheet content at 24 and 48 h of ageing (*top panel*). Whereas, in the presence of DNAJB6, there is already the modest presence of intermolecular β-sheet at 2 h, but there is no further increase with aging (*bottom panel*). We also detect differences in the random coil peak position between the two preparations, indicating a different chemical environment of intrinsically-disordered polypeptide chains in the presence of DNAJB6.

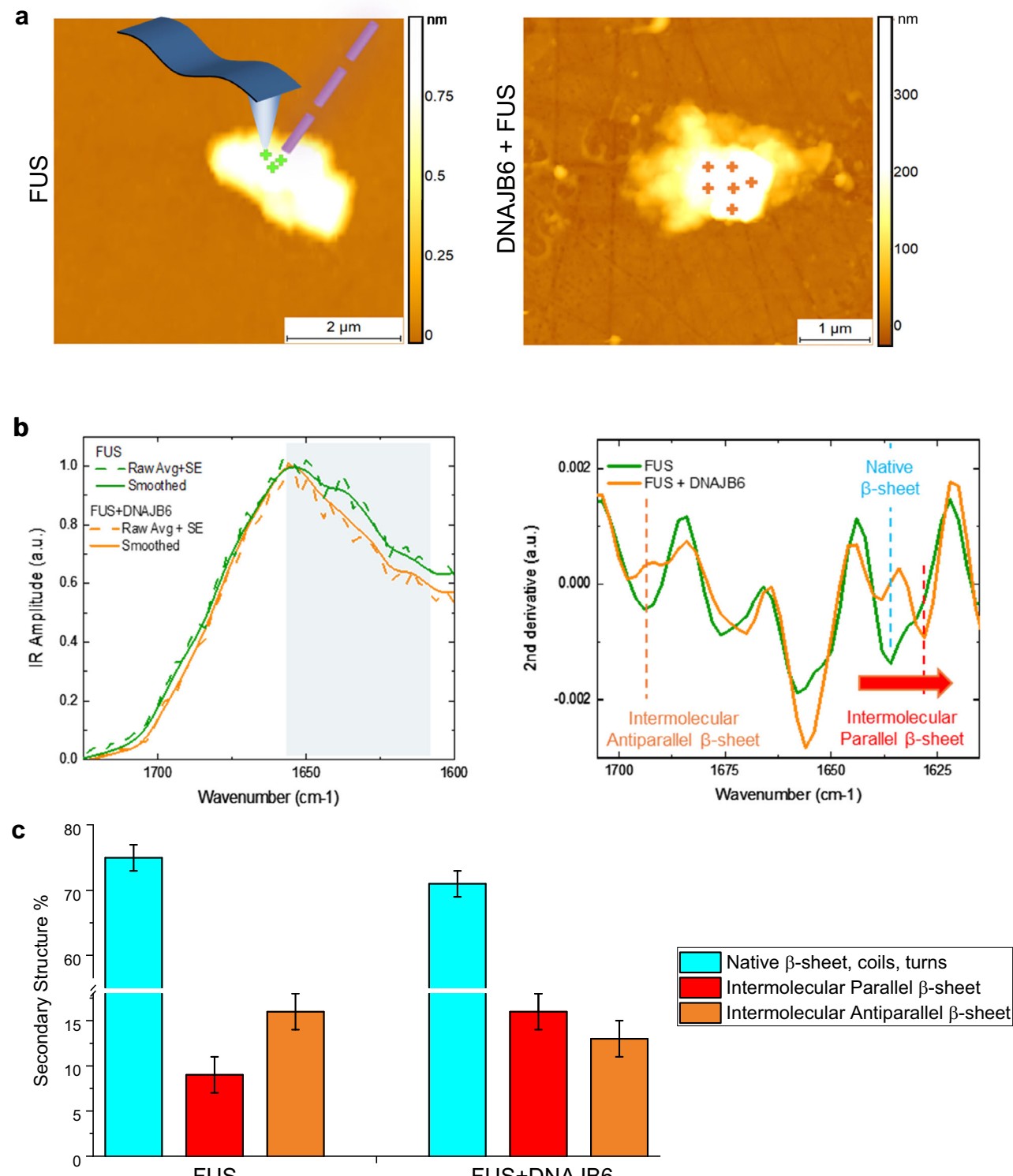

**Fig. 6 | AFM-IR nano-chemical analysis of FUS and FUS + DNAJB6 condensates.** **a** Maps of 3D morphology. **b** IR average spectra from 3 independent condensates and second derivative of the amide I band to deconvolve protein secondary structure contributions. a.u. is arbitrary units. **c** Quantification of secondary structure. Quantification of secondary structure. Data are shown as integration of the area of the second derivative spectra that is used for the determination of the average secondary structure ± the error of area integration for three biological replicates. FUS, $n > 100$ spectra; FUS + DNAJB6, $n > 80$ spectra.

of HSP70 partners, the glycine/phenylalanine rich region which contributes to its ability to phase separate, and the serine rich domain that is implicated in client recognition and binding[54,56]. Deletion of any of these domains prevented DNAJB6 from rescuing FUS toxicity within yeast, consistent with the recent results using purified proteins for a related HSP40 chaperone, DNAJB1, and its interaction with FUS (Supplementary Fig. 24a)[84]. All the examined deletion mutants were expressed in yeast except for the variant with the J-domain removed

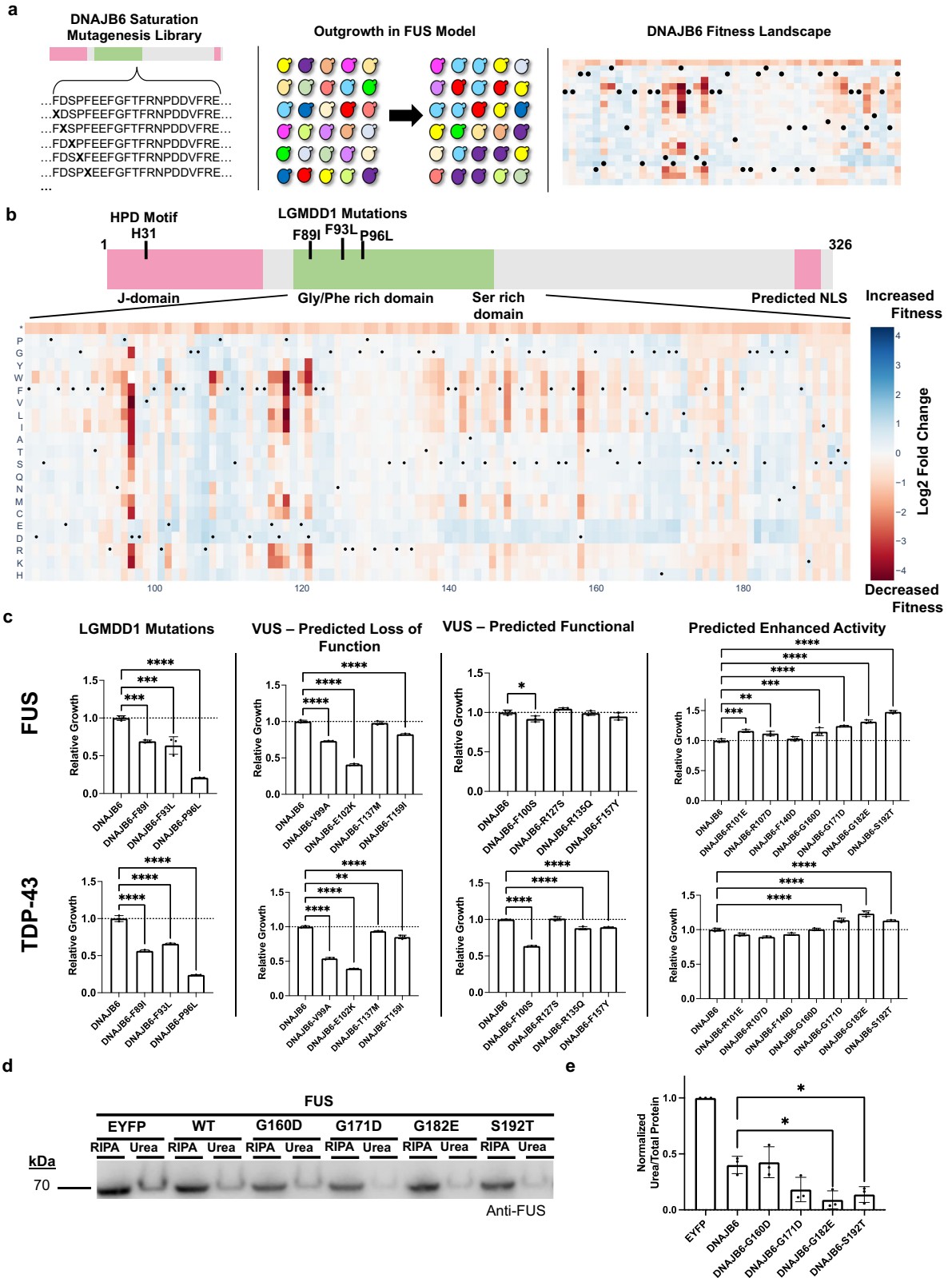

(Supplementary Fig. 24b, c). To better examine the role for the J-domain a point mutant within the conserved HPD motif of the J-domain (H31Q) was created. This mutant expresses well but blocks the ability of DNAJB6 to stimulate the ATPase activity of, and transfer misfolded substrate to HSP70 family members[53,85]. The H31Q mutation also rendered DNAJB6 non-functional for rescuing FUS-mediated toxicity within yeast (Supplementary Fig. 24). These results suggest that all domains within DNAJB6 are required for its effects and that cooperating with a HSP70 partner may be necessary for its full function in cells. We also verified that the interaction between DNAJB6 and an HSP70 partner is required for it to reduce the formation of SDS-insoluble species in mammalian cell culture (Supplementary Fig. 25, 26).

**Fig. 7 | Deep mutational scan (DMS) of DNAJB6 identifies potentiated variants.**
**a.** Overview of DMS approach for testing DNAJB6 variants against the FUS model. A library of DNAJB6 mutants is generated using a site directed mutagenesis-based approach, the resulting plasmid library is transformed into yeast containing the FUS model and grown under inducing conditions, finally the relative growth rates of all mutants are determined and normalized to the wild-type DNAJB6 variant. **b** DMS heatmap for residues 83-194 of DNAJB6. Intensity of blue or red colored boxes indicates increased or decreased activity as compared to the wild-type DNAJB6 protein. * indicates stop codon mutation, black dots (•) mark the wild-type residue for each site in the protein. **c** Validation of DMS results in the FUS model and testing of variants against the TDP-43 model. ***$P = 0.0007$, ***$P = 0.0002$, ****$P = < 0.0001$, ****$P = < 0.0001$, ****$P = < 0.0001$, ****$P = < 0.0001$, *$P = 0.0398$,

***$P = 0.0002$, **$P = 0.0047$, ***$P = 0.0006$, ****$P = < 0.0001$, ****$P = < 0.0001$, ****$P = < 0.0001$, ****$P = < 0.0001$, ****$P = < 0.0001$, ****$P = < 0.0001$, ****$P = < 0.0001$, ****$P = < 0.0001$, ****$P = < 0.0001$, **$P = 0.0015$, ****$P = < 0.0001$, ****$P = < 0.0001$, ****$P = < 0.0001$, ****$P = < 0.0001$, ****$P = < 0.0001$, ****$P = < 0.0001$, ****$P = < 0.0001$. **d.** Testing of potentiated DNAJB6 variants in mammalian cells for their ability to reduce SDS-insoluble, urea-soluble FUS species upon overexpression with 200 ng of FUS plasmid. **e** Quantification of the urea soluble species normalized to total protein detected by Ponceau S staining in (**d**). Each independent replicate was normalized its associated EYFP rescued sample. All bar chart data are shown as mean ± s.d. for three biological replicates. *$P = 0.0102$, *$P = 0.0266$. Comparisons were conducted with ordinary one-way ANOVA with display of significant comparisons; *$P \leq 0.05$, **$P \leq 0.01$, ***$P \leq 0.001$, ****$P \leq 0.0001$.

During our deletion analysis, we observed that loss of the glycine/phenylalanine (G/F) rich domain or the serine (S) rich domain from DNAJB6 enhanced the toxicity of the FUS model (Supplementary Fig. 24). These regions have been implicated in substrate recognition and are also the site of mutation within LGMDD1 patients as well as multiple variants of uncertain significance within the ClinVar database. To decipher the function of this critical region at increased resolution, we conducted a deep mutational scan (DMS) across the 112 amino acids of the G/F and S-rich regions. Deep mutational scanning libraries were constructed by performing comprehensive mutagenesis of each codon, yielding versions of DNAJB6 with all possible amino acids or a stop codon at each interrogated position (Fig. 7a). The library of variants was then tested for their ability to rescue the proteotoxicity induced upon FUS overexpression, creating a comprehensive fitness landscape of all single amino acid mutations in the G/F and S rich region on DNAJB6 activity (Fig. 7b). Mutagenesis libraries were constructed and screened in biological duplicates, with strong correlation between replicates observed (see Methods, Supplementary Fig. 27). Notably, regions rich with patient mutations causing LGMDD1 (amino acids 89–100) frequently resulted in a decrease in rescue, with subsequent validation of these mutations confirming our screening results (Fig. 7b, c). The loss of rescue may be driven by an inability of a given DNAJB6 variant to interact with their RBP clients or the DNAJB6 variant gaining an intrinsic toxicity that then synergizes with the RBP toxicity. Expressing several of our examined DNAJB6 variants alone in cells revealed that the loss of rescue was due to both of the mentioned possibilities (Supplementary Fig. 28). Future studies will be required to understand the mechanism driving this variable toxicity between mutants. Numerous variants of uncertain significance (VUS) within ClinVar were also captured within the DMS data. Follow-up studies individually examining these mutants revealed a subset that were defective in their ability to rescue FUS expressing models, suggesting that additional disease associated mutants may exist beyond what is currently annotated (Fig. 7c). Within the DNAJB6 mutational landscape, variants with enhanced activity against FUS were observed. Outside of the conservative S192T mutation which showed one of the strongest effects on rescue, a general trend was seen where mutation to an acidic residue in stretches between 138-171 and 182-189 appeared to enhance activity compared to WT DNAJB6 (Fig. 7b). In line with this observation, a single acidic amino acid within these stretches, D158, was critical for function as mutation of D158 to almost any other amino acid other than glutamic acid reduced the activity of the protein (Fig. 7b). Based on the screening results, DNAJB6 mutants with enhanced activity were selected for validation against FUS, revealing several mutations that showed clear gains in activity as compared to the wild-type protein (Fig. 7c). Finally, when the various DNAJB6 mutants were tested against the TDP-43 model, similar results were obtained, in agreement with the existence of a common mechanism of interaction between DNAJB6 and its aggregation-prone clients (Fig. 7c).

To further establish the relevance of the enhanced DNAJB6 variants, we examined their ability to reduce the formation of SDS-insoluble, urea-soluble FUS species in mammalian HEK293T cells. Given that WT DNAJB6 almost entirely reduced the formation of SDS-insoluble, urea-soluble FUS, we modified the assay by transfecting more FUS expression plasmid into cells which resulted in higher levels of SDS-insoluble, urea-soluble FUS species (Supplementary Fig. 29). Utilizing higher FUS expression, the potentiated DNAJB6 variants, G182E and S192T showed a further reduction in the formation of SDS-insoluble, urea-soluble FUS species as compared to the wild-type protein (Fig. 7d, e and Supplementary Fig. 30). These findings demonstrate that the activity of DNAJB6 can be improved and that rescuer proteins optimized in yeast can be readily translated to mammalian cell systems[27].

## DNAJB6 rescues motor neuron loss and associated microgliosis in an animal model of ALS-FUS

Next, we set out to determine whether DNAJB6 could rescue the age-dependent motor neuron loss and associated microgliosis within an ALS-FUS mouse model. This knock-in model expresses two mutant forms of FUS (FUS P517L and FUSΔ14) which mimic early-onset patient variants. These animals show a consistent loss of motor neurons and increased microgliosis at 6 months of age, and have served as a preclinical model for therapeutic intervention with a FUS-targeting antisense oligonucleotide[86]. To determine the effect of overexpressing DNAJB6 within diseased motor neurons, an AAV vector was generated which contained a CAGGS promoter regulating DNAJB6 with an intervening lox-stop-lox (LSL) transcriptional termination cassette. When the LSL cassette is present, DNAJB6 is not transcribed due to the multiple copies of a transcriptional termination sequence within the LSL cassette. In the presence of Cre, which is expressed from the ChAT promoter in this ALS-FUS model, the LSL cassette is excised and DNAJB6 is expressed in ChAT+ motor neurons. DNAJB6 was then packaged into AAV9 and delivered via intracerebroventricular injection to newborn ALS-FUS or control pups. As an additional control, a cohort of animals was injected with a GFP containing virus. Animals were then analyzed at 6 months of age and samples from the L4-L5 region of the spinal cord were processed for immunostaining (Table 1). Robust Cre-dependent expression of DNAJB6 was confirmed in the spinal motor neurons (Supplementary Fig. 31). In the control animals there was no difference in the numbers of motor neurons or staining for the microglial activation marker Iba1 between DNAJB6 vs GFP treated animals (Fig. 8). Consistent with the expected phenotype, ALS-FUS animals that received the GFP control virus showed a marked ~20% decrease in motor neurons and a simultaneous increase in Iba1 positive microglia. In contrast, the overexpression of DNAJB6 in ALS-FUS animals was protective, resulting in preserved motor neuron numbers and no evidence of abnormal microglial activation (Fig. 8).

**Table 1 | Immunofluorescence antibody information**

| Antibody | Manufacturer | Catalog number | Dilution |
|----------|-------------|----------------|----------|
| ChAT | Millipore-Sigma | AB144P | 1:250 |
| Iba1 | Wako | 019-19741 | 1:5,000 |
| hDNAJB6 | Invitrogen | MA5-27342 | 1:1,000 |

Antibody name, manufacturer, catalog number and dilution details provided for all antibodies used in L4-L5 lumbar section immunostaining.

## Discussion

Modeling of neurodegeneration in simplified cellular systems has yielded insights into how proteotoxic aggregation-prone proteins disrupt cellular function. However, a great deal of additional information is required to understand these complex processes fully, and to provide practical information that can be translated into effective diagnostics and therapeutics for the associated human disorders. The work described here advances this goal in two ways. First, we developed a multiplexed approach in which numerous misfolding-prone proteins can be screened in parallel to identify a rich dataset of candidate modulators acting either in a protein-specific way, or in a more general class-specific manner. Second, our work has uncovered a previously unrecognized mechanism to the best of our knowledge whereby protein-protein interactions can modulate the propensity of misfolding-prone proteins to form species that injure cells - namely by facilitating their conversion into gel-like biomolecular condensates rather than injurious fibrillar assemblies. In the paragraphs below, we explore each of these concepts in greater detail.

### A platform for identifying modulators of proteotoxic species

Our platform provides several advantages over conventional approaches. Specifically, by inserting the same misfolding-prone protein into several uniquely barcoded strains and analyzing their collective response to a given genetic perturbation, we greatly enhance the sensitivity and specificity of our assay. Crucially, the quantitative nature of our approach enables us to capture subtle changes in growth by a putative rescuer as compared to traditional, semi-quantitative methods. Moreover, by simultaneously studying multiple models within the same genetic background and under the same testing paradigm, we can make broad observations about the nature of rescuers and the relationship between models. Finally, previous studies typically screen cDNA libraries derived from the model organism used for the screen (e.g. yeast or flies). Here, we demonstrate across dozens of models that our platform enables direct screening of human genes in simple organisms such as yeast to identify disease relevant suppressors. We screened less than 10% of known human genes as putative rescuers and nevertheless identified multiple interactions, suggesting that further screening would deliver many more genetic modifiers. We note that a limitation of our screening approach is that not all human genes will function appropriately in yeast. This may be due to the lack of a required protein partner that is present in humans but absent in yeast. Alternatively, yeast may contain a homolog of the required partner that is unable to interact with the expressed human protein because of the evolutionary distance between yeast and humans. By performing studies where we simultaneously deliver a set of factors known to cooperate, it may be possible to overcome this limitation. In a similar fashion, work has also been done creating humanized yeast which have yeast genes replaced with their human counterpart and these may be a better cellular chassis in which to conduct our studies[87]. In addition, codon optimization of each of the tested human genes for expression in yeast, could help to mitigate false negatives arising from the poor expression in yeast[88]. Finally, additional technical improvements to the platform, enabled by using more individually barcoded strains, would likely provide more

sensitive detection of weaker interactions like those that may emerge from the expression of a human gene within yeast.

Overall, this work establishes a high-throughput platform for identifying genetic suppressors applicable to any family of proteins so long as their expression in yeast causes a growth defect that is dependent upon the biological function the user intends to interrogate[89,90]. Furthermore, with minor modifications our approach can be readily applied toward screening against libraries of small molecules to identify compounds with specific versus broad activities against classes of disease relevant proteins such as the NDD-associated proteins studied in this work.

An additional finding from our screens was the lack of concordance between proteotoxic species of the same class, such as various dipeptide repeats and poly-alanine models. In contrast, RNA-binding proteins showed a general trend towards having rescuers that were active on several members of its group. No rescuers, however, showed clear activity on all RNA-binding proteins and each tended to show a preference for a particular subset (Fig. 1 and Supplementary Fig 8-9). These results imply that there is no universal feature that is conserved across a given family of NDD-associated proteins, but that there can still be shared features among the misfolded species that enable genetic modifiers to interact with several members of the family.

As our library also contained different pathogenic variants of the same protein of interest, we asked what, if any, was the effect of these mutations on our screening results. Interestingly, although many of the tested mutations have been shown to accelerate the rate of protein misfolding or to drastically alter the localization of the variant protein, they appeared to be rescued in a similar manner as the wild-type protein (Supplementary Fig 8, 9)[62,65,91,92]. These results suggest that while the tested mutations, increase the probability of misfolding (and thus the probability of disease), they do not appear to fundamentally change the underlying mechanism of toxicity or the misfolded state that is present within the cell. These results raise the possibility that a therapeutic modality designed against the wild-type form of a protein might also work for patients with rare disease-causing variants.

The finding that the overexpression of DNAJB6 within an ALS-FUS mouse model rescues motor neuron loss and the associated microgliosis, reinforces the value of using simple cellular systems to rapidly search for genetic modifiers of disease. Additional work will be required to verify our initial findings and understand the long-term safety and efficacy of overexpressing DNAJB6 or enhanced mutants with more specific and improved anti-FUS aggregation activity. Furthermore, given that DNAJB6 reduces the toxicity and misfolding of other ALS-associated RBPs, such as TDP-43, it will be interesting to determine if DNAJB6 rescues ALS models that are driven by dysfunction in other RNPs.

It is important not to overlook prior work which has demonstrated that DNAJB6 can prevent the aggregation of poly-glutamine repeat expansion containing proteins, along with acting on beta-amyloid and alpha-synuclein[50,54,55,57,93,94]. Together with our study, these results raise the possibility that DNAJB6 represents a generalized chaperone for several clinically relevant misfolded species. Furthermore, the ability to observe DNAJB6 activity in a tractable yeast model enables the high-throughput optimization of its activity. We demonstrate that extensive deep mutational scanning of DNAJB6 can identify enhanced variants. Additional screening of these mutant libraries against numerous protein misfolding models could be used to select for more broadly active versions of the protein (but with the caveat that such broadly active versions could have mechanism-based side effects arising from their effects on off-target proteins). Alternatively, these same mutant libraries could be used to selectively tune DNAJB6 towards a narrower class of targets should an increase in specificity be desired for eventual therapeutic applications. By examining, the

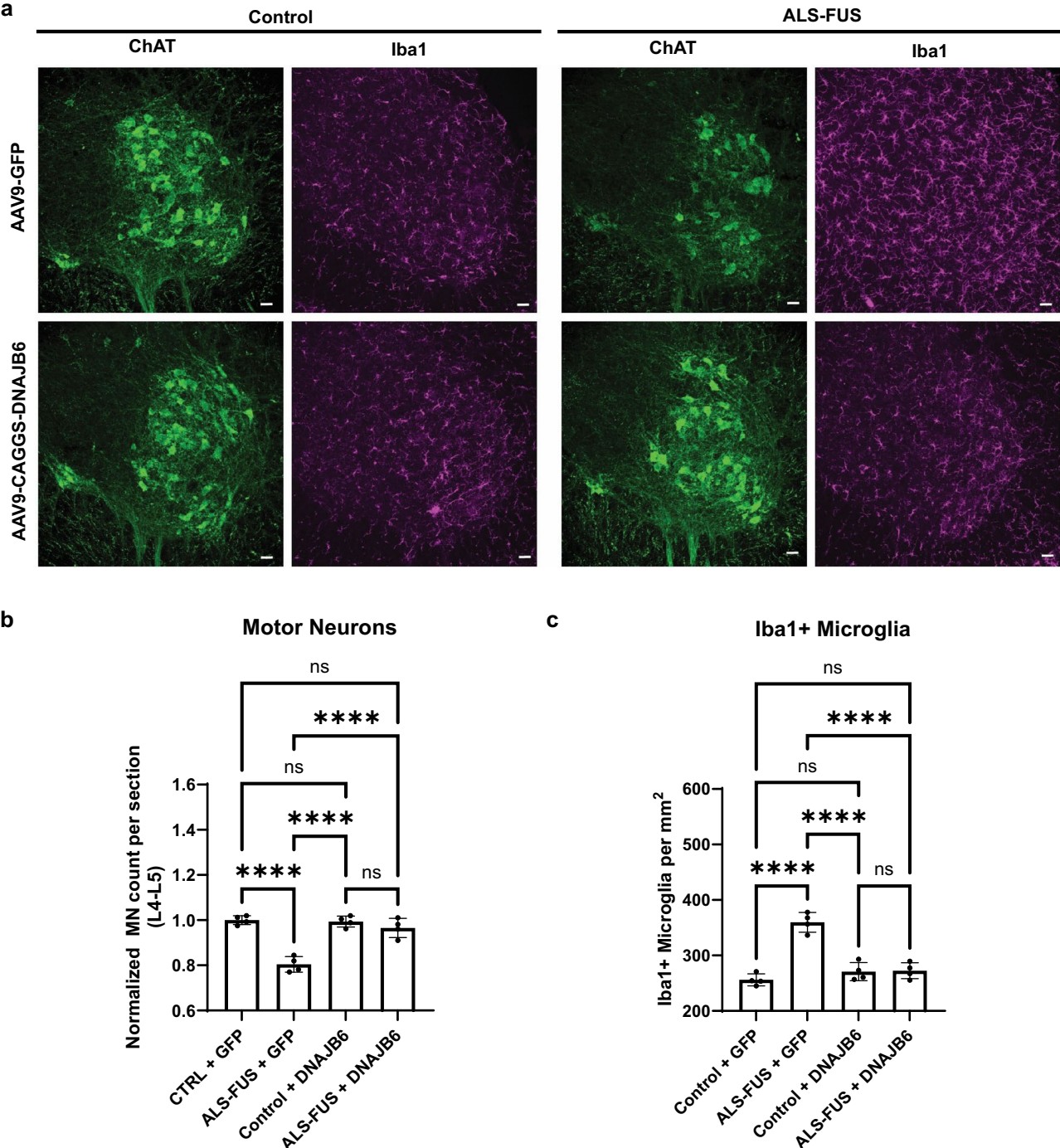

**Fig. 8 | DNAJB6 rescues FUS-mediated MN loss in a mouse model of ALS-FUS.**
**a** Representative immunofluorescence staining of motor neurons (ChAT positive staining) and activated microglia (Iba1 positive staining) in control or ALS-FUS mice exposed to GFP or DNAJB6 containing constructs delivered via AAV9. Scale bar represents 50 microns. **b**, **c** Quantification of ChAT positive MNs and Iba1+ microglia at the L4-L5 spinal cord level normalized to the control mice provided GFP. All data are shown as mean ± s.d. and were gathered from 4 independently exposed mice. ****$P$ = < 0.0001, ****$P$ = < 0.0001, ****$P$ = < 0.0001, ****$P$ = < 0.0001, ****$P$ = < 0.0001, ****$P$ = < 0.0001. Comparisons were conducted with one-way ANOVA; ns = not significant, ****$P$ < 0.0001.

properties and three-dimensional structure of the obtained enhanced DNAJB6 proteins, we can gain fundamental insights into how these classes of proteins may function and evolve.

**Changing the phase state of phase separating proteins forming biomolecular condensates**
There is a growing body of work describing critical functional interactions between components of biomolecular condensates (including intrinsically disordered proteins (IDPs) and their interacting RNAs) and molecular chaperones (especially homomeric/heteromeric assemblies of heat shock proteins - HSPs)[95–102]. Investigating potential interactions between these components (IDPs, RNAs, and chaperones) will be of interest to pursue in future studies. Yet, even in the absence of knowledge about whether or not RNAs play a role in FUS-DNAJB6 condensate formation, there is evidence that IDP-chaperone interactions play a key role both in forming biomolecular condensates and in

preventing rampant aggregation of their aggregation-prone components[95–102].

In support of this idea, we show that DNAJB6 colocalizes with FUS in neurites of cultured human iPSC-derived motor neurons expressing endogenous FUS and DNAJB6 at physiological levels. In biochemical and biophysical experiments, we show that DNAJB6 co-partitions with FUS, and can prevent hardening (also termed aging) of FUS into dense fibrillary aggregates. These biophysical results, when taken in conjunction with the results of the yeast and mouse model experiments potentially position DNAJB6 to play an important functional role in modulating FUS phase state both physiologically and in the presence of ALS-associated mutations. More generally, our work strongly supports the emerging notion that the interaction of HSPs with aggregation-prone components of the biomolecular condensates helps to maintain the reversible phase dependent properties of these condensates.

Our data also raise several interesting but currently unresolved questions, including the immediate question of which residues or domains of FUS and DNAJB6 are necessary for stabilizing FUS in a gelled state that does not progress to hardened fibrillar aggregates? DNAJB6 shares broad similarities with several ATP-independent sHSPs (e.g. HSPB1 and HSPB8). It is a ~ 36 kDa soluble protein comprised of four domains. At the N-terminus, the α-helical domain (J domain) interacts with HSP70 and stimulates HSP70 ATPase activity. Next to it, there is an intrinsically disordered glycine/phenylalanine-rich (G/F) domain that also contains multiple arginine and tyrosine residues. This domain is likely to be involved in interactions with the intrinsically disordered domains of client proteins like FUS and TDP-43. However, this domain also forms an inhibitory loop that docks with the J-domain, and regulates binding and subsequent activation of the partnered HSP70[103] and HSP70-related nucleotide exchange factors (NEFs) like BAG2 and BAG3[104], which also interact with DNAJB6[105]. The C-terminally adjacent serine rich (S) domain is responsible for DNAJB6 oligomerization, along with client recognition. Finally, there is a structured C-terminal (CT) domain containing a single β-sheet motif, which likely interacts with structured domains of client proteins. Our domain deletion and deep mutational scan studies confirm that all of these key domains in DNAJB6 are necessary for the ability of DNAJB6 to rescue FUS toxicity in yeast. The mutagenesis also identifies a key role for D158, and highlights enhanced rescue by substitution of acidic amino acids in residues 138–171 and 182–189, as well as the specific S192T mutant. However, additional studies combining biophysical and cell biological analyses are required to understand whether they primarily affect DNAJB6-FUS interactions or DNAJB6 HSP70-related interactions. Additional studies using either time-resolved cross-linking mass spectrometry or hydrogen-deuterium exchange methods, and possibly structural studies using cryo-electron microscopy will be required to precisely map the protein-protein interaction sites. However, because of the broad architectural similarity between DNAJB6 and other ATP independent sHSPs, some testable hypotheses can be constructed. Prior work on ATP-independent sHSP chaperones reveals that they bind unfolded client proteins through both their intrinsically disordered and their structured domains[106,107]. Specifically, the low complexity domain (LCD) of HSPB8 (equivalent to the G/F and S domains of DNAJB6) facilitates partitioning of HSPB8 into FUS condensates via arginine / tyrosine cation-π interactions similar to those that drive FUS condensation[66,107]. The structured α-crystalline domain of HSPB8 (equivalent to the CT domain of DNAJB6) then interacts with and stabilizes the structured RNA Recognition Motif (RRM) in FUS, thereby abrogating its propensity to form amyloid fibrils[107]. A similar mechanism has also been proposed for how HSPB1 modulates conversion of TDP-43 into a gel-like state rather than into a fibrillar amyloid state[104]. We propose here that DNAJB6 likely follows a similar pathway, with the initial FUS-DNAJB6 interaction driven by cation-π (arginine, tyrosine/phenylalanine) and charge-charge (lysine/arginine,

aspartate/glutamate) interactions between FUS and the disordered G/F and S domains of DNAJB6. We hypothesize that the ordered CT domain of DNAJB6 then interacts with the RRM domain of FUS in the same way as HSPB8. This hypothesis on how DNAJB6 might affect condensate proteins phase state is broadly in agreement with studies on how DNAJB6 mitigates Aβ aggregation. In this case, the disordered N-terminal domain of DNAJB6 inhibits primary nucleation of Aβ at non-stoichiometric ratios (as observed here), while the β-strands of the structured CT domain of DNAJB6 are involved in mitigating secondary nucleation of Aβ fibrils[108]. Finally, like HSPB8 and HSPB1, DNAJB6 is known to interact with HSP70 and several NEFs including BAG2 and BAG3[105]. If correct, this would provide a mechanism by which DNAJB6 mediates refolding and dispersal of gelled (but not hardened fibrillar) FUS condensates (see below).

A second question raised by our data is whether the gel-like state of FUS-DNAJB6 condensates is the physiological endpoint of pathological FUS condensates rescued by DNAJB6 interaction. Again, our data does not explicitly address this issue, however, we suspect that this is unlikely. It is more likely that the FUS-DNAJB6 interaction forms gel-like condensates that prevent further hardening of FUS into fibrillar aggregates. By preventing further hardening of FUS, the FUS-DNAJB6 gels might facilitate the subsequent interaction with HSP70 family chaperones and their related NEFs to mediate refolding and dispersal of FUS[105]. This hypothesis is supported by our data showing that the ability of DNAJB6 to ameliorate RBP toxicity in yeast and prevent RBP misfolding in mammalian systems is dependent upon its interaction with HSP70 family chaperones (Supplementary Fig. 24, 25). In addition, recent observations show that in cells recovering from heat shock, HSP70 is required for the DNAJB6-mediated conversion of solid orphan ribosome protein (ORP) condensates into a liquid-like state in the nucleolar periphery[97,109]. This observation suggests that the gel-like state formed by interaction with DNAJB6 prevents aggregation of ORP components into hardened fibrillar states and allows engagement with HSP70 and HSP70-related NEFs to then disperse the gel-like state.

Intriguingly, in contrast to conventional chaperone-mediated disaggregation, the chaperone-mediated processes appear to be much faster in biomolecular condensates. Studies in yeast show that the Hsp104-, Hsp70-, and Sis1-chaperone mediated dispersal of Pab1 biomolecular condensates is unexpectedly rapid[96]. This rapid dispersal and refolding appears to reflect the fact that unfolded condensate proteins (like Pab1), are only partially threaded through the central canal of Hsp104[96]. By contrast, conventional misfolded proteins (e.g. luciferase) require complete threading through the central canal of Hsp104[96]. If the same situation holds true for mammalian proteins, the rapid dispersal of FUS-DNAJB6 gelled condensates may account for the relative infrequency of FUS-DNAJB6 condensates observed in neurons under physiological conditions. Further work will be required to investigate the dynamics of the engagement of DNAJB6, HSP70 and NEFs, and correlate them with changes in the biophysical state of FUS condensates.

A third question raised by our work is how the FUS-DNAJB6 interaction might mitigate ALS/FTD progression. As previously shown by us and others, one very likely mechanism by which FUS aggregates causes ALS and FTD is by sequestration of RNP granule cargo (RNA and RNA processing machinery) in the pathologically hardened cytoplasmic FUS condensates[62,66,110]. By preventing or dispersing these hardened pathological condensates, DNAJB6 and its partners would liberate FUS and its RNA/RNA processing cargo, which would restore new protein synthesis in distal neuronal compartments. Similar effects on nuclear transcriptional condensates would likely also occur. The combined effects of preventing or dispersing the hardening FUS condensates would likely preserve the normal roles of FUS in transcription and translation. However, an additional beneficial effect of DNAJB6 mediated inhibition of hardened FUS condensates relates to a potential impact on seeding and spreading of FUS aggregates during

disease progression. Recent work reveals that pathological FUS aggregates can seed further aggregation and promote the spreading of pathological FUS aggregates across the brain[111] – a phenomenon first shown for prion-related neurodegeneration[112]. DNAJB6 and HSP70-mediated dispersal of intracellular pathological FUS condensates might be expected to attenuate the release of hardened pathological FUS condensates into the extracellular space. This would have the predictable effect of attenuating the subsequent seeding of further aggregation in other parts of the central nervous system.

While our work focuses on how DNAJB6 prevents the irreversible hardening of FUS (and possibly other) biomolecular condensates, other chaperones also play keys roles in modulating the formation of biomolecular condensates. For example, cytoplasmic Poly-A binding proteins (PABPC) bind to a short linear motif (SLiM) in the Ataxin 2 (ATXN2). This interaction allows PABPC to serve as an ATXN2-chaperone, regulating ATXN2 condensation and its interaction with stress granules[113].

Finally, our work suggests that modulating the aggregation of intrinsically disordered proteins in biomolecular condensates could be targeted therapeutically. However, before that can be achieved, additional biophysical experiments are needed to explore the impact of chaperone partitioning and their interface and surface effects on aggregation of biomolecular condensate components. This information, together with information about the molecular grammar and specific motifs driving chaperone- condensate interactions will be needed in order to develop peptidic and small molecular mimics that specifically target pathological condensates, and do not disrupt the physiological chaperone-client interactions.

## Methods

### Yeast strains and media
Barcoded *S. cerevisiae* yeast BY4741 *MATa his3Δ1 leu2Δ0 ura3Δ0 met15Δ0* strains were purchased from Horizon (Cat. #YSC5117). To introduce rescuers through mating, rescuer containing *S. cerevisiae* yeast BY4742 strains *MATα his3Δ1 leu2Δ0 ura3Δ0 lys2Δ0* were used. Individual barcoded strains containing expression vectors were maintained in Synthetic Complete (SC) -ura media (20 g/L glucose, 1.5 g/L Drop Out mix [US Biological D0539-09A], 1.7 g/L Yeast Nitrogen Base [US Biological Y2030], 5 g/L Ammonium Sulfate [Fisher H8N2O45], supplemented with 18 mg/L Leucine and 9 mg/L Histidine). Individual rescuer BY4742 strains were maintained in SC -his media (20 g/L glucose, 1.5 g/L Drop Out mix [US Biological D0539-09A], 1.7 g/L Yeast Nitrogen Base [US Biological Y2030], 5 g/L Ammonium Sulfate [Fisher H8N2O45], supplemented with 18 mg/L Leucine and 1.8 mg/L Uracil). Mating was conducted in YPD media (20 g/L glucose, 20 g/L peptone, and 10 g/L yeast extract). Selection for mated strains was conducted in SC -ura -his media (20 g/L glucose, 1.5 g/L Drop Out mix [US Biological D0539-09A], 1.7 g/L Yeast Nitrogen Base [US Biological Y2030], 5 g/L Ammonium Sulfate [Fisher H8N2O45], supplemented with 18 mg/L Leucine), while outgrowth of induced mated strains was carried out in SC -ura -his gal media (20 g/L galactose, 1.5 g/L Drop Out mix [US Biological D0539-09A], 1.7 g/L Yeast Nitrogen Base [US Biological Y2030], 5 g/L Ammonium Sulfate [Fisher H8N2O45], supplemented with 18 mg/L Leucine).

### Plasmids
Proteotoxic genes and controls were cloned into either the pAG416GAL-ccdb (Addgene #14147) or pAG426GAL-ccdb (Addgene #14155) using Gateway LR II Clonase Enzyme mix (Invitrogen). Once expression plasmids were sequence verified, they were transformed into barcoded BY4741 strains using standard LiOAc transformation protocols and plated on SC -ura agar plates. Yeast rescuer genes and control rescuer genes were cloned into pAG413GAL-ccdb (addgene #14141) using Gateway cloning. Human rescuer genes from the hOrfeome V8.1 Library collection were cloned into a derivative of

pAG413GAL-ccdb, pAG413GAL-ccdb-6Stop, wherein the 3' attR2 site was modified to encode a stop codon 6 amino acids downstream of the last codon to compensate for a lack of a stop codon in the ORFeome.

During the validation of the plasmids for public distribution, a deletion at position S326 was identified in the EWSR1 gene. When compared to the wild-type ORF, the S326 deletion was found to cause similar toxicity levels.

All mammalian expression vectors were cloned into the pLEX307 backbone (Addgene #41392) using Gateway LR II Clonase Enzyme mix (Invitrogen).

Plasmid DNA was isolated using standard miniprep buffers (Omega Biotek) and silica membrane columns (Biobasic). All expression plasmids were Sanger sequenced to confirm the appropriate insert (Genewiz).

### Yeast multiplexed screening
Each plate of rescuers was screened in biological duplicates. A fresh aliquot (500 μL) of frozen barcoded yeast pool was inoculated into 5 mL of SC -ura media and rotated at 30 °C. At the same time, 5 μL of each rescuer strain was inoculated into 500 μL of SC -his media in 96 well 2 mL deep well plate format (VWR) and shaken at 900 rpm at 30 °C. 24 h later, 5 μL of the saturated barcoded yeast pool was mixed individually with 5 μL of rescuer strain in a new 96 well plate where each well was filled with 500 μL of YPD and shaken at 900 rpm at 30 °C. For selection of mated strains, 20 h later, 5 μL of mated barcoded yeast pool was transferred into a new 2 mL deep well plate filled with 500 μL of SC -ura -his media and shaken at 900 rpm at 30 °C for 24 h. For outgrowth, 2 μL of the mated and selected pool was inoculated into 1 mL of SC -ura -his galactose media and shaken at 1000 rpm at 30 °C for 30 h.

After growth, 100 μL of yeast culture was removed and the optical density (OD595) of the culture was determined in a 96 well plate reader (Tecan). After measurement of culture density, genomic DNA was extracted using a modified LiOAc-SDS extraction method. Briefly, plates were centrifuged for 5 min at 4000 rpm. Supernatant was discarded and the pellet was resuspended in 200 μL of 200 mM LiOAc with 1% SDS with rigorous pipetting. Plates were sealed with aluminum foil and incubated at 70 °C for 20 min to enable lysis. 600 μL 100% ethanol was added to each well and pipetted up and down rigorously before being centrifuged for 10 min at 4000 rpm. Supernatant was discarded and pellets were air dried for 30 min under flame. Pellets were then resuspended in 200 μL 1X TE and incubated at 42 °C for 30 min. The plates were centrifuged for 10 min at 4000 rpm and the supernatant containing DNA was pipetted into a new plate for storage at −20 °C. Raw sequencing reads from the chaperone screen have been uploaded to the NCBI SRA under BioProject PRJNA769721 (SUB10508463). Raw sequencing reads from the orfeome screen have been uploaded to the NCBI SRA under BioProject PRJNA769721 (SUB10508562).

### Sequencing library preparation
For sequencing on NextSeq 500/550 (Illumina), libraries were prepared from genomic DNA in two PCR steps. The first step amplifies genomic DNA containing the DNA barcode and attaches an internal index to designate which column the well was amplified from. The second PCR attaches Illumina indexes to the amplicon, wherein the combination of Illumina indexes indicates the row and plate location of the well. The first PCR step was done in technical duplicates unless otherwise stated with Taq polymerase (Enzymatics). The following reaction mix was used: 2 μL 10X Taq buffer, 0.1 μL 100 μM forward primer, 0.1 μL 100 μM reverse primer, 0.1 μL Taq polymerase, 0.4 μL 10 mM dNTPs, 0.5 μL DNA, and 16.8 μL H$_2$O. The following cycling conditions were used: 1. 94 °C, 180 s, 2. 94 °C, 30 s, 3. 60 °C, 20 s, 4. 72 °C, 30 s, 5. Return to step 2 27X, 6. 72 °C, 180 s. After the first round of PCR, technical replicates of each individual well were pooled. For

the second round of PCR where Illumina indexes were attached, the following reaction mix was used: 2 μL 10X Taq buffer, 0.1 μL 100 μM forward primer, 0.1 μL 100 μM reverse primer, 0.1 μL Taq polymerase, 0.4 μL 10 mM dNTPs, 0.5 μL DNA from first round PCR, and 16.8 μL H$_2$O. The following cycling conditions were used: 1. 94 °C, 180 s, 2. 94 °C, 30 s, 3. 56 °C, 20 s, 4. 72 °C, 30 s, 5. Return to step 2 7X, 6. 72 °C, 180 s. After the second round PCR, all reactions corresponding to a plate of screening were pooled together. The reaction products were run out on a gel and a band corresponding to the right size was gel extracted. Libraries were quantified with the NEBNext Library Quant Kit for Illumina according to manufacturer instructions (NEB). Pooled libraries were combined and sequenced with a 75 cycles NextSeq 500/550 High Output Kit on a NextSeq 500/550 machine (Illumina).

### Analysis of multiplexed screening

Raw reads in fastq format were trimmed and assigned to wells via combinations of Illumina indexes and column designating internal indexes. 20 bp barcode sequences were aligned to a reference genome allowing for +1 or −1 shifts in the sequencing phase using bowtie2. Raw counts of exact matches for each barcode were determined and well-read counts were normalized by the total number of reads in that well and converted to counts per million (CPM) unless otherwise stated. Wells were analyzed in batches with other wells in the same plate. Wells with less than 15,000 total reads were discarded in addition to wells where 1 biological replicated received less than 15,000 total reads. After CPM normalization, the estimated actual abundance of reads was calculated by normalizing against the optical density (OD595) of that well, which was measured immediately prior to harvesting with a 96 well Infinite F50 plate reader (Tecan). Wells containing control or inert rescuers were identified and the average read counts of each barcode in controls was determined. The variance in the number of reads between control wells for each barcode was determined. The mean-variance relationship was modeled using the equation $\log(\sigma\text{variance-mean}) = \log(k)+b*\log(\text{mean})\sigma$ as previously described[114]. The barcode mean and adjusted variance were used to determine whether a barcode in a test well was significantly upregulated using a one-sided cumulative density function assuming a normal distribution. The associated $p$-value was adjusted using a Benjamini-Hochberg procedure correcting for the number of tests in that well. To obtain model level information from individual barcode strains, p-values from independent barcode strains associated with the same model were combined using Stouffer's method. After this summary value was obtained, a further Benjamini-Hochberg procedure was used to correct summary values for the number of models and the number of wells in each plate. Average log2 fold change was calculated as the base 2 logarithm of the average change in counts over the expected value in the control wells. Analysis was conducted with custom scripts in R Version 4.0.2.

### Spot assays

Yeast strains to be assayed were grown overnight in selective synthetic complete media until saturation was reached. Saturated cultures were serially diluted 1:5 in sterile PBS (Gibco). To SC −ura -his glucose or galactose agar plates, 5 μL of diluted culture was spotted. Plates were left for 30 min to dry before being inverted and incubated at 30 °C for 48 h, followed by being scanned to document growth.

### Yeast liquid culture growth assay

Proteotoxic yeast strains were grown in 500 μL SC -ura media in plate format shaken at 1000 rpm at 30 °C for 24 h. At the same time, rescuer yeast strains were grown in 500 μL SC -his media shaken at 1000 rpm at 30 °C for 24 h. After growth, 5 μL of appropriate proteotoxic and rescuer yeast strains were mixed in 500 μL YPD and shaken at 1000 rpm at 30 °C for 24 h. To 500 μL of dual selective SC -ura -his media, 5 μL of mated strains were inoculated and shaken at 1000 rpm at 30 °C for

30 h. 48 h prior to reading, mated and selected strains were inoculated in SC -ura -his galactose media at one of 3 dilution factors (Supplementary Data 1) depending on their growth rate and shaken at 1,000 rpm at 30 °C for 24 h. 24 h after initial inoculation, strains were passaged depending on their specified dilution factor into fresh SC -ura -his galactose media. Upon reaching the assay endpoint, 100 μL of each well was transferred to a 96 well plate (Greiner) and the optical density was determined on a 96 well plate reader (Tecan). Multiple media only wells were also quantified as a baseline and these values were subtracted from optical density measurements. All statistics were performed in GraphPad Prism Version 9.2.0 or 10.2.1.

### Protein harvesting and western blotting

**RIPA/urea extractions.** Cells in 24-well dishes were washed with ice cold PBS (Gibco) after media was removed. 250 μL of RIPA buffer (50 mM Tris-HCl (pH 7.4), 150 mM NaCl 1% NP-40, 0.5% sodium deoxycholate and 0.1% SDS, Alfa Aesar) was added to each well and allowed to sit for 2 min. Cells were resuspended in RIPA buffer and moved to conical tubes. To lyse cells further, cells were sonicated for 10 s while kept on ice. Cell lysate was centrifuged for 20 min at 12,000 x $g$ at 4 °C and supernatant was saved as RIPA soluble fraction. Pellets were washed 1X with RIPA buffer and 50 μL urea buffer was added (8 M Urea, 2 M Thiourea, 4% CHAPS,) with vigorous pipetting to resuspend pellet and spun for 20 min at 12,000 x $g$ at 4 °C. The concentration of protein in the RIPA buffer was determined with a Bradford assay and lysates were adjusted to a final concentration between 250-350 μg/μL depending on the yield of the lowest concentration of the lysate in the set in which it was processed with 1X LDS loading buffer (Invitrogen). RIPA soluble fractions were boiled for 5 min and stored at −80 °C until use. Urea fractions were stored at −80 °C without boiling.

**Yeast lysate extraction.** Cells were collected and centrifuged at 2300 × $g$ for 2 min and washed with 1 mL of dH$_2$O. To pelleted yeast cells, 200 μL of 0.1 M NaOH was added and cells were resuspended by vortexing. Cells were allowed to lyse for 10 min at room temperature. Lysed cells were spun at 13,000 × $g$ for 1 min and supernatant was discarded. Pellets were resuspended in 50 μL of dH$_2$O and 25 μL of 200 mM DTT (Fisher) was added along with 25 μL of 4x LDS loading buffer (Invitrogen). Samples were boiled at 95 °C for 5 min and subsequently centrifuged at 800 x $g$ for 10 min at 4 °C. Supernatants were collected and moved to a new tube for storage at −20 °C.

**Western blotting.** 10 μL of normalized lysate with loading buffer was loaded into NuPAGE 4−12% Bis-Tris protein gels (Invitrogen) and subjected to 100 V electrophoresis for 65 min. Separated proteins were transferred onto a 0.2 μM PVDF membrane and blocked with SuperBlock (Invitrogen). Primary antibodies were diluted in SuperBlock with 0.1% Tween-20 and incubated overnight at 4 °C with gentle rotation. Blots were washed with TBST before secondary antibody incubation. Blots were imaged with the Odyssey XF imaging system (Li-Cor) using the chemiluminescent detection. After transfer, blots were stained with Ponceau S stain (G Biosciences) for 15 min. Total protein was imaged on a LAS-4000 imager (Fujifilm). Band intensities were quantified with Image Studio Lite (Li-Cor). For TDP-43 blots, high molecular weight (HMW) species were included in quantification of TDP-43 blots[60].

TDP-43 was detected with a polyclonal rabbit antibody at a 1:2,500 dilution (Proteintech 10782-2-AP). FUS was detected with a polyclonal rabbit antibody at a 1:2,500 dilution (Proteintech 11570-1-AP). hnRNPA1 was detected with a polyclonal rabbit antibody at a 1:5000 dilution (Proteintech 11176-1-AP). DNAJB6 was detected with a monoclonal mouse antibody at a 1:2500 dilution (Proteintech 66587-1-Ig). A goat anti-rabbit HRP conjugated antibody was used at a 1:50,000 dilution (Invitrogen G21234). A goat anti-mouse HRP conjugated antibody was

used at a 1:10,000 dilution (Invitrogen 31430). All statistics were performed in GraphPad Prism Version 9.2.0 or 10.2.1.

## Protein purification and in vitro LLPS experiments

FUS-mEmerald was purified as previously described[66]. His-Sumo tagged DNAJB6 expression vector was transformed into *E. coli* BL21(DE3) (NEB) for protein expression and purification. DNAJB6 expressing cells were grown at 37 °C until an OD600 of 0.6 was reached. Expression was induced with 0.5 mM IPTG overnight at 16 °C. Cell pellets were collected and subjected to high pressure lysis (Constant System) in lysis buffer (50 mMTris pH 7.5, 20 mM imidazole, 500 mM NaCl with 1x protease inhibitor cocktail). Lysate was centrifuged at 100,000 x *g* and collected supernatant was applied to a 10 mL Ni-Advance (BioServ, UK) column. After washing, His-Sumo tagged DNAJB6 was eluted in 250 mM imidazole containing buffer and cleaved overnight with ULP-protease at 4 °C. Cleaved DNAJB6 was diluted 5-fold before running through a cation exchange column. SP Sepharose chromatography was conducted in 50 mM HEPES, pH 7.5 with a salt gradient from 5 M – 1 M NaCl. Fractions containing the protein were concentrated and subjected to size-exclusion on a Superdex-75 16/600 column in 50 mM HEPES and 100 mM NaCl, pH 7.5. Through all stages of purification, presence of DNAJB6 was monitored via SDS-PAGE.

DNAJB6 was labeled with Alexa Fluor™ 555 $C_2$ Maleimide (Thermo Scientific) following the manufacturer's guidelines.

For LLPS experiments, concentrated, purified proteins (FUS, DNAJB6, and/or BSA) were diluted to 1.5 μM in a 50 mM NaCl solution with 50 mM Tris pH7.5 unless otherwise stated. For imaging, condensates were maintained in PEG-silane coated Ibidi™ coverslides to avoid wetting. Imaging was conducted with Zeiss Axiovert 200 M microscope with Improvision Openlab software using 100X magnification objective.

## TCSPC-FLIM

mEmerald-tagged WT-FUS in NaCl solution (and DNAJB6) were mixed in milli-Q water, to give final protein concentrations of 1 μM of WT-FUS, 0.16 μM WT-DNAJB6, and 60 mM NaCl. 7 μL of each condensate mixture was deposited in individual silicon wells (Press-to-Seal, Thermo-Fisher Scientific) attached on 1.5 thickness coverslips (Superior Marienfeld, Lauda-Konigshofen, Germany) for imaging. During ageing, samples were stored in the fridge at 5 °C. Samples were imaged on a home-built confocal fluorescence microscope equipped with a time-correlated single photon counting (TCSPC) module. A pulsed, super-continuum laser (Fianium Whitelase, NKT Photonics, Copenhagen, Denmark) provided excitation a repetition rate of 40 MHz. This was passed into a commercial microscope frame (IX83, Olympus, Tokyo, Japan) through a 60x oil objective (PlanApo 60XOSC2, 1.4 NA, Olympus). The excitation and emission beams are filtered through GFP-appropriate bandpass filters centered at 474 and 542 (FF01-474/27-25, FF01-542/27, Semrock Inc., NY, USA). Laser scanning was performed using a galvanometric mirror system (Quadscanner, Aberrior, Gottingen, Germany). Emission photons were collected on a photon multiplier tube (PMT, PMC150, B&H GmBH, Berlin, Germany) and relayed to a time-correlated single photon counting card (SPC830, B&H GmBH). Images were acquired at 256×256 pixels for 120 s (i.e., 10 cycles of 12 s). Photon counts were kept below 1% of laser emission photon (i.e., SYNC) rates to prevent photon pile-up. TCSPC images were analysed using an in-house, MATLAB-based (MathWorks, Natnick, MA, USA) phasor plot analysis script (https://github.com/LAG-MNG-CambridgeUniversity/TCSPCPhasor) as well as FLIMPA[115], from which fluorescence lifetime maps and values were generated. Fluorescence lifetimes are, where applicable, presented as those from individually segmented condensates from 9 images (giving total a total of 86−209 condensates analysed per sample) taken over 3 fully independent experiments. Statistical analysis was performed on Prism ver. Prism 9.5.1 (GraphPad, San Diego, CA, USA), where, depending on the data distribution, either non-parametric Kruskal-Wallis tests with Dunn's multiple comparisons or one-way ANOVA tests with Tukey's multiple comparisons were applied.

**Fabrication of microfluidic devices.** A two-step photolithographic process was used to fabricate the master used for casting microfluidic spray devices[116]. In brief, a 25 μm thick structure was fabricated (3025, MicroChem) was spin-coated onto a silicon wafer. This was then soft-baked for 15 min at 95 °C. An appropriate mask was placed onto the wafer, exposed under ultraviolet light to induce polymerization, and then post-baked at 95 °C. A second 50 μm thick layer (SU-8 3050, MicroChem) was then spin-coated onto the wafer and soft-baked for 15 min at 95 °C. A second mask was then aligned with respect to the structures formed from the first mask, and the same procedure was followed, i.e. exposure to UV light and post-baking for 15 min at 95 °C. Finally, the master was developed in propylene glycol methyl ether acetate (Sigma-Aldrich) to remove any photoresist which had not cross-linked. A 1:10 ratio of PDMS curing agent to elastomer (SYLGARD 184, Dow Corning, Midland, MI) was used to fabricate microfluidic devices. The mixture was cured for 3 h at 65 °C. The hardened PDMS was cut and peeled off the master. The two complementary PDMS chips are then activated with O2 plasma (Diener Electronic, Ebhausen, Germany) and put in contact with each other and aligned precisely such that the gas inlet intersects with the liquid inlet to form a 3D nozzle[74,117,118].

**Microfluidic spray deposition.** Aliquots of FUS solution was thawed immediately before use. Phase separation was induced in an Eppendorf tube by lowering the salt concentration, via sample dilution to a final concentration of 50 mM NaCl, 25 mM Tris. Aliquots were then taken from the solution at desired time points (i.e. 2, 24, and 48 h) and promptly deposited via microfluidic spray deposition.

Prior to introduction of sample, each device was tested and washed with MilliQ water for 5 min. Sample was then loaded into 200 μL air-tight glass syringes (Hamilton) and driven into the spray device using a syringe pump (Harvard apparatus). Solutions containing sample were pumped into the device with a maximum flow rate of 100 μl/h to minimize sample shearing, while the nitrogen gas inlet pressure was maintained at 3 bar. Deposition was conducted for a maximum of 10 s at a distance of 3.5 cm to ensure coalescence of droplets did not occur. Samples were sprayed directly onto the relevant surfaces (i.e. ZnSe crystals, FTIR prism) with no further washing steps required before measurements.

**Atomic force microscopy.** Samples were deposited onto plasma-cleaned ZnSe crystals via microfluidic spray deposition. Nanomechanical characterization was performed in 25 mM Tris buffer using a MultiMode 8 (Bruker, USA) operating in either force-volume or quantitative nanomechanical mapping mode. ScanAsyst Fluid probes were used, with a spring constant of 0.7 N/m and a nominal radius of ~20 nm. Images were acquired at scan rates of 0.3−0.5 Hz at 512 × 512 pixels and the frequency of oscillation of the Z-piezo was set to 1 kHz. Probes were calibrated using the thermal tune method. Peak force values were chosen between 200 and 1000 pN. Deformation was monitored to ensure consistent indentation values across condensates with different material properties, and to ensure indentation depths did not result in excessive sampling of the substrate. Only condensates above 30 nm in height were considered. All measurements were performed at room temperature. Force-distance curves were analyzed in Nanoscope Analysis Software Version 2.0 (Bruker, USA). A Hertz-DMT contact mechanic model was used to fit the contact region of withdrawal curves. The contact region of the withdrawal curve was analyzed via second derivative analysis; those curves which showed a biphasic response were rejected, as this was taken as an indication of contribution from the hard substrate.

**Fourier transform infrared spectroscopy.** Measurements were performed on a Vertex 70 FTIR spectrometer (Bruker, USA) equipped with a DiamondATR unit and a deuterated lanthanum a-alanine- doped triglycine sulfate (DLaTGS) detector. Each spectrum was acquired with a scanner velocity of 20 kHz over 4000 to 400 $cm^{-1}$ as an average of 256 scans. Thin films were deposited on the prism using microfluidic spray deposition. New background spectra were acquired before each measurement. All spectra were normalized and analyzed using OriginPRO 2025 (OriginLab, USA). To determine the secondary structure composition of proteins, a second derivative analysis was performed. Spectra were first smoothed by applying a Savitzky-Golay filter.

**Infrared nanospectroscopy (AFM-IR)**
A nanoIR3 platform (Bruker) combining high resolution and low-noise AFM with a tunable quantum cascade laser (QCL) with top illumination configuration was used. The sample morphology was scanned by the nanoIR3 system, with a line rate within 0.1–0.4 Hz and in contact mode. A silicon gold coated probe with a nominal radius of 30 nm and a cantilever with an elastic constant of about 0.2 N $m^{-1}$ was used. Both infrared (IR) spectra and maps were acquired by using phase loop (PLL) tracking of contact resonance, the phase was zeroed to the desired off-resonant frequency on the left of the IR amplitude maximum and tracked with an integral gain I = 0.1–5 and proportional gain P = 1–5. All images were acquired with a resolution above 500 × 100 pixels.

The AFM images were treated and analyzed using SPIP Version 6.7.3 (Surface Metrology, Denmark). The height images were first order flattened, while IR and stiffness related maps were only flattened by a zero-order algorithm (offset). Nanoscale-localized spectra were collected by placing the AFM tip on the top of the condensates with a laser wavelength sampling of 2 $cm^{-1}$ and a spectral speed of 100 $cm^{-1}$/s within the range 1462–1800 $cm^{-1}$. Within a single condensate, the spectra were acquired at multiple nanoscale localized positions, the spectrum at each position being the co-average of 5 spectra.

Successively, the spectra were treated by OriginPRO 2025 (OriginLab, USA). They were smoothed by an adjacent averaging filter (5 pts) and a Savitzky-Golay filter (second order, 7 points) and normalised. Spectra second derivatives were calculated, smoothed by a Savitzky-Golay filter (second order, 5 points). Relative secondary and quaternary organisation was evaluated by integrating the area of the different secondary structural contributions in the amide band I.

The spectra from 3 different condensates (FUS, n > 100; FUS + DNAJB6, n > 80) were averaged and used to determine the secondary structure of the condensates. The error in the determination of the relative secondary structure was calculated over the average of at least 5 independent spectra and it is < ±3%.

Spectra were analysed using the microscope's built-in Analysis Studio (Bruker) and OriginPRO 2025 (OriginLab, USA). All measurements were performed at room temperature, with laser power <2 mW and under controlled Nitrogen atmosphere with residual real humidity below 5%.

**Mammalian cell lines and cell culture**
HEK293T cells used in this study were obtained from ATCC. Cells were maintained at 37 °C in a humidified atmosphere with 5% $CO_2$. HEK293T cells were grown in Dulbecco's Modified Eagle Medium (DMEM, Invitrogen) which was supplemented with 10% fetal bovine serum (Gibco) and penicillin-streptomycin (Invitrogen).

**Mammalian transfection**
24 h prior to transfection, 293 T cells were seeded at 40–60% confluency into 24-well plates coated for 30 min with a 0.1 mg/mL solution of poly-D-lysine (MP Biomedicals Inc.) and washed with PBS (Gibco) once prior to media and subsequent HEK293T cell addition. The next day, expression plasmid was incubated with Opti-MEM (Gibco) and Lipofectamine 2000 (Invitrogen) for 30 min at room temperature

prior to addition to cells, per manufacturer protocol. 20 h after transfection, media was changed. Cells were harvested for protein extraction and western blotting 48 h after transfection.

**Co-immunoprecipitation assay**
**Transfection.** 24 h prior to transfection, 293 T cells were seeded at 40–60% confluency into 6-well plates coated for 30 min with a 0.1 mg/mL solution of poly-D-lysine (MP Biomedicals Inc.) and washed with PBS (Gibco) once prior to media and subsequent HEK293T cell addition. The next day, expression plasmid was incubated with Opti-MEM (Gibco), P300 Reagent (Invitrogen) and Lipofectamine 3000 (Invitrogen) for 30 min at room temperature prior to addition to cells, per manufacturer protocol. Media was changed 24 h after transfection.

**Bead preparation.** 20 μL of Pierce™ Anti-DYKDDDDK Magnetic Agarose Beads (Thermo Scientific) were washed 3x in 250 μL. 5% NP40 lysis buffer (50 mM Tris-HCl (pH 7.5), 150 mM NaCl. 5% NP-40). Beads were resuspended in 1 mL of blocking buffer (50 mM Tris-HCl (pH 7.5), 150 mM NaCl. 5% NP-40, 1% BSA w/v) and incubated for 1 h at 4 °C.

**Preclearing.** 30 μL of Pierce™ A/G Magnetic Agarose Beads (Thermo Scientific) were washed 3x in 250 μL lysis buffer (50 mM Tris-HCl (pH 7.5), 150 mM NaCl. 5% NP-40). Beads were resuspended in 1 mL of blocking buffer (50 mM Tris-HCl (pH 7.5), 150 mM NaCl. 5% NP-40, 1% BSA w/v) and incubated for 1 h at 4 °C. Beads were then incubated with cell lysate for 1 h.

**Co-immunoprecipitation.** HEK293T cells were transfected as described above with 2 μg of expression plasmid and grown for 48 h post-transfection. Cells were washed with ice cold PBS (Gibco) after media was removed and 500 μL of trypsin was added to each well before incubation for 5 min at 37 °C. Cells were transferred to conical tubes and washed 1X with ice cold PBS (Gibco) before being resuspended in 400 μL of lysis buffer (50 mM Tris-HCl (pH 7.5), 150 mM NaCl. 5% NP-40). Cells were rotated for 30 min at 4 °C before centrifugation for 15 min at 16,000 × g at 4 °C. 10% of the lysate was transferred to a conical tube and processed with 1x LDS loading buffer (Invitrogen) with 5% added β-mercaptoethanol. The remaining lysate was either precleared (hnRNPA1) as described previously or immediately (TDP-43) incubated with prepared Pierce™ Anti-DYKDDDDK Magnetic Agarose Beads (Thermo Scientific) for 2 h. Beads were washed either 3x (hnRNPA1) or 6x (TDP-43) with wash buffer (50 mM Tris-HCl (pH 7.5), 150 mM NaCl, 5% NP-40, 1% BSA w/v). Beads were resuspended with 100 uL of 1x LDS loading buffer (Invitrogen) and boiled at 95 °C for 5 min. Samples were transferred to conical tubes and stored with input lysates at −80 °C.

**Granule immunofluorescence imaging.** Two wild type human iPSC lines (CS29iALS-C9n1.ISOxx, Cedars Sinai; NSB3234, RUCDR) were differentiated into motor neurons and fixed in 4% PFA at day 40 post-dissociation[119]. After permeabilization in 0.2% Triton X-100 and block in PBS/1% BSA, the motor neurons were stained for two hours at room temperature with anti-FUS (Proteintech 11570-1-AP; 1:300), anti-DNAJB6 (Proteintech 66587-1; 1:1000), and anti-Beta III Tubulin (EMD Millipore AB9354; 1:1000) and subsequently with appropriate secondary antibodies in PBS/1% BSA. Airyscan experiments were conducted on an inverted Zeiss LSM 980 microscope equipped with an Airyscan2 module. Samples were excited with standard laser lines at 633 nm, 561 nm, 488 nm, and 405 nm excitation wavelengths. Images were acquired using a plan apochromat 63×/1.4 NA DIC M27 oil objective. Multi-color Z-stack images (150 nm step size) were acquired by using a piezo-motor, a frame-scanning mode with 4-line averaging, and processed using automatic 3D Airyscan processing in ZEN Blue Version 3.10 (Zeiss).

**Granule detection and colocalization.** Airyscan Z-stack slice image-areas containing FUS-positive neuritic particles were duplicated in Fiji, and used as input for granule detection and colocalization analysis. Each individual FUS granule was represented only once within the complete image data set. Granules containing FUS and/or DNAJB6 were detected using a custom script in the Fiji distribution of ImageJ[120]. The ImageJ code was based on a script previously developed to analyze synapses in confocal images[121]. A Difference of Gaussians (DoG) detector from the ImgLib2 algorithm library[122] was used to locate intensity peaks in the image data set. Inspection of the stacks confirmed that there was no detectable chromatic aberration that would cause artifactual absence of colocalization. Size and intensity criteria for FUS and DNAJB6 granule detection were determined manually, and the same parameters were applied to all images in the two wild type motor neuron datasets. FUS puncta within 0.071 μm of DNAJB6 puncta were scored as colocalized. All granules determined as colocalized overlapped in several 150 nm sections. Peak count data was consolidated using R Statistical Software (v4.4; R Core Team 2024). Neurite areas were manually delineated as ROIs in Fiji and included to allow comparison with stochastic colocalization. Similar results were obtained independently of masking the neuritic areas.

**Colocalization simulation and statistical analysis.** A stochastic colocalization simulation was generated to investigate the random chance of overlap between signals from FUS and DNAJB6. ROIs outlining neurite areas were imported as masks into MATLAB (The MathWorks Inc., version R2021b). A custom MATLAB script was used to generate random positions of signals of two channels, representing simulated FUS and DNAJB6 granules, within each ROI mask at the experimentally obtained density for each image and channel. The randperm function was used to ensure the positions were unique (with no overlap). An analysis of the percentage of colocalization versus number of simulations was performed to determine the number of simulations required for a stable colocalization percentage in all masks. For each ROI, the simulations were conducted 20 times, and the average number of colocalized FUS puncta was calculated based on the same criteria used for experimental colocalization, i.e., simulated FUS puncta within 2 pixels (0.071 μm) of simulated DNAJB6 puncta were considered colocalized. The results of the simulation revealed an overlap rate by chance of 0.55 ± 0.07%, mean and standard deviation. The number of colocalized puncta in experimental and simulated datasets was compared using a right-tailed two-sample $t$ test designed to test whether the experimentally determined colocalization was greater than the simulated colocalization.

**Deep mutational scanning.** Deep mutational scanning libraries were prepared in biological duplicates, wherein PCR mutagenesis, construction of bacterial libraries, and construction of yeast libraries were completed as independent replicates. The DNAJB6 yeast expression vector, pAG413GAL-DNAJB6 was miniprepped immediately prior to use. Variant versions of DNAJB6 were made by a single primer site-directed mutagenesis protocol. Oligos were designed to introduce a degenerate codon, NNK, at each amino acid position. Additionally, each oligo was designed to introduce 2–4 synonymous mutations at the codon immediately prior to the degenerate codon to increase sampling diversity. For each codon, an individual mutagenesis single primer PCR reaction was conducted in technical duplicates. The following PCR mix was used for mutagenesis: 5 μL 5X Q5 reaction buffer, 0.5 μL 10 mM dNTPs, 150 ng DNA, 1.25 μL 10 μM primer, and $H_2O$ to 25 μL. The following cycling conditions were used: 1. 98 °C, 45 s, 2. 98 °C, 15 s, 3. 60 °C, 15 s, 4. 72 °C, 260 s, 5. Return to step 2 29X, 6. 72 °C, 240 s. After PCR, the unmodified backbone was digested with 1 μL DpnI at 37 °C for 1 h. After digestion, independent PCR replicates were pooled and sets corresponding to 14 contiguous amino acids were pooled together to enable data analysis using short read Illumina

sequencing. 8 total sets were created encompassing 112 mutagenized amino acids. The combined sets were column purified with the Zymo DNA Clean & Concentrator Kit. Each biological replicate of each set was transformed into electrocompetent 10-beta *E. Coli* (New England Biolabs) in triplicate according to manufacturer instructions. Cells were plated following outgrowth and recovery on 15 cm LB agar plates containing ampicillin and colonies were allowed to form for 24 h at 30 °C. An individual typical transformation yielded 3-10 million colonies for a total of approximately 9-30 million colonies for each biological replicate of each set. As a quality control measure, 20 colonies from each set were sequenced to ensure editing. All colonies were scraped off plates and plasmid libraries were purified by Midiprep (Zymo). Plasmid libraries were transformed into BY4741 containing the expression plasmid pAG416GAL-FUS. Each biological replicate of each set was transformed in 96 separate transformation reactions to ensure appropriate coverage and allowed 48 h for outgrowth on SC -ura -his plates at 30 °C. Yeast libraries were scraped into 10 mL sterilized PBS and frozen in 20% glycerol. For outgrowth, 600 μL of frozen yeast library was inoculated into 6 mL of SC -ura -his for 18 h in triplicate at 30 °C with rotation. After inoculation into galactose media, the remaining cells from the overnight cultures were spun down at 4000 rpm for 5 min and the pellets were frozen at −20 °C. For each independent culture outgrowth, 12 μL of saturated overnight culture was inoculated into 6 mL of SC -ura -his galactose for 48 h at 30 °C with rotation. Each tube was centrifuged at 4000 rpm for 5 min and the supernatant was discarded. Pellets were resuspended in 300 μL of 200 mM LiOAc with 1% SDS and incubated for 15 min at 70 °C with shaking at 800 rpm. Afterwards, 900 μL 100% ethanol was added, tubes were vortexed, and centrifuged at 13,000 rpm for 10 min. Supernatant was discarded and pellets were allowed to air dry for 20 min under flame. Pellets were then resuspended in 200 μL TE and incubated at 42 °C for 20 min and then centrifuged at 13,000 rpm for 10 min. The supernatant containing DNA was collected and stored for further use. Each sample was then independently amplified and subsequently indexed for sequencing on a NextSeq 500/550 (Illumina). Depending on the set, amplification of the mutagenized region was done in 8 technical replicates that were pooled after amplification. The following mix was used for all PCR reactions: 5 μL 5X Q5 reaction buffer, 0.5 μL 10 mM dNTPs, 0.5 μL DNA, 0.125 μL 100 μM forward primer, 0.125 μL 100 μM reverse primer, 0.25 μL Q5 polymerase, and 18.5 μL $H_2O$. Amplification was done for 24 cycles for non-induced libraries and 28 cycles for induced samples with the following conditions: 1. 98 °C, 45 s, 2. 98 °C, 15 s, 3. 58 °C, 15 s, 4. 72 °C, 30 s, 5. Return to step 2, either 24x or 28×72 °C, 240 s. After pooling, index sequences were attached using the same PCR mix and cycling conditions, but for 8 cycles of amplification. Amplicons were pooled according to set and amplicon length and gel purified to remove primers.

Sequencing data were processed on Illumina Basespace according to default QC settings and downloaded as fastq files. Sequences were aligned using custom Python code. A raw activity score was calculated as:

$$\text{Activity}_i = \log_2\left(\frac{\text{Mut}_i}{\text{WT}}\right)_{\text{induced}} - \log_2\left(\frac{\text{Mut}_i}{\text{WT}}\right)_{\text{uninduced}}$$

where the subscript i denotes individual unique variants. $\text{Mut}_i$ denotes the average number of counts of the particular mutant codon of interest, averaged over all codings, while WT denotes the number of counts of the wild-type nucleotide sequence. Raw activity scores were then normalized across sets by anchoring stop codon mutants to -1 and WT to 0 to eliminate set by set variation that may have arisen due to experimental fluctuation. The final normalized activity scores are presented in heatmap format for easy visualization. Plotted wild-type values are derived from the recoded versions of the wild-type residue at a given position and thus do not always have a 0 value. Raw

sequencing reads have been uploaded to the NCBI SRA under BioProject PRJNA769721 (SUB10503160).

### Cas9 knockout of DNAJB6

A Cas9 expressing HEK293T cell line was generated in a well of a 24-well dish by transfecting 300 ng of pB-CAGGS-Cas9-SV40-BPSV40 vector a long with 100 ng of a plasmid expressing the Piggybac transposase (System Biosciences, LLC) with Lipofectamine 2000 according to manufacturer protocols (Invitrogen). Media was changed 24 h after transfection before selecting with Noursethricin N-Acetyl Transferase (NAT) at 300 µg/mL 48 h after transfection. Cells were expanded and continuously selected with NAT for 2 weeks before being frozen down for further use.

A gRNA lentivirus compatible plasmid encoding two guide RNAs from the Brunello library (GCATATGAAGTGCTGTCGGA and GACTTCTTTGGGAATCGAAG) targeting DNAJB6 was created. Lentivirus was created from this plasmid by co-transfecting HEK293T cells with this plasmid alongside psPAX2 (Addgene #12260) and MD2.G (Addgene #12259). After transfection and a media change 24 h after transfection, media containing lentivirus was harvested 72 h later. Lentivirus containing media was added to Cas9 containing cells for 24 h before a media change. Beginning 48 h after the media change, cells were exposed to 2 weeks of alternating drug selections of NAT at 500 µg/mL and Blasticidin at 2 µg/mL every 48 h with regular splitting, to ensure both Cas9 and gRNA maintained good expression and were not silenced during the outgrowth process. After 2 weeks, single cells were sorted into 96 well plates using the Bigfoot Spectral Cell Sorter (Thermo). To gate on single cells, forward scatter and side scatter were used to isolate single cells and sort them into 100 µL of media. Single cells were allowed to expand for 2 weeks under alternating drug selection. DNA was harvested with QuickExtract (Lucigen) according to manufacturer protocols. PCR primers spanning individual cut sites in addition to primers spanning a potential deletion were used to amplify out the region of DNAJB6 subject to cutting. PCR products were sanger sequenced and TIDE was used to analyze sanger fragments to confirm disruption of DNAJB6[123]. To confirm loss of DNAJB6, western blots were performed. The same process was repeated with non-targeting control guide RNAs (AAAAAGCTTCCGCCTGATGG and AAAACAGGACGATGTGCGGC).

### RNA-seq

HEK293T cells were transfected as previously described with 50 ng of expression plasmid and grown for 72 h in a 24 well dish after transfection. Cells were harvested in TRIzol and stored at −80 °C. RNA was harvested from cells with the Direct-zol miniprep kit (Zymo). Harvested RNA was prepared for sequencing with the NEBNext® Ultra™ II RNA Library Prep Kit for Illumina (NEB). Two biological replicates were performed for each condition. Each individual replicate was amplified with a unique combination of indexing primers after the adaptor ligation step to uniquely identify it. Pooled libraries were combined and sequenced with a 75 cycles NextSeq 500/550 High Output Kit on a NextSeq 550 machine (Illumina). Each replicate was allocated ~30 million reads. Reads were aligned to the hg19 genome using HISAT2 Version 2.1.0 to obtain counts. Differential expression was calculated using limma. Raw sequencing reads have been uploaded to the NCBI SRA under BioProject PRJNA769721 (SUB10426285).

### Human ORFeome library construction

The pooled hORFeome V8.1 library was inserted into the pAG413GAL-ccdb-6Stop vector with Gateway LR II Clonase Enzyme mix (Invitrogen) at a ratio of 150 ng:50 ng. The reaction was incubated overnight at 25 °C. Expression plasmids were electroporated into electrocompetent 10-beta E. Coli (NEB) and ~200,000 colonies were harvested and Miniprepped. The expression plasmid library was transformed into BY4742 and selected in SC -his glucose plates for 48 h. Individual

colonies were picked and arrayed into 96 well plates and saved. To identify the ORF present in each well, each plate was process individually. A total of 20 pools per plate were made, consisting of 12 column pools and 8 row pools. DNA from each of these pools was then obtained using a LiOAc-based extraction. ORFs within each pool were amplified for 30 cycles with general primers binding to the galactose promoter and cyc terminator and subsequently column purified. 250 ng of the purified PCR product was processed with the NEBNext Ultra II FS DNA Library Prep Kit for Illumina (NEB) according to manufacturer protocols. After adaptor ligation and prior to indexing, a forward primer placed 60 bp upstream from the ATG start codon was used in combination with an adaptor reverse primer and amplified for 13 cycles to selectively enrich for the human ORF containing fragment in preparation for sequencing. Each individual pool was then uniquely indexed and sequenced with a 150 cycles NextSeq 500/550 High Output Kit on a NextSeq 500/550 machine (Illumina). After sequencing, the identity of each well was determined by its presence in a specific combination of row and column wells. Some wells were not able to be determined and are demarcated with a "ORF identity unknown" designation in Supplementary Data 5, Supplementary Data 6, and Supplementary Data 7.

### Mouse lines and procedures

C57Bl/6J (000664) and ChAT-CreΔneo (031661) mouse lines were obtained from Jackson Laboratory. Mutant FUS knock-in mice were generated as previously described[86]. All strains were backcrossed to C57Bl/6J background for at least five generations. All animals were housed in a specific pathogen-free facility with ambient temperature of 18–23 °C and 40–60% humidity and a 12-h light/dark cycle. For neonatal intracerebroventricular injections, animals were anesthetized on ice, and then 5*10^10 vg of AAV9-LSL-DNAJB6 or AAV9-GFP in a total volume of 5 µl was injected into the right lateral ventricle using a Hamilton Neuros syringe. AAVs were generated by Packgene. To collect fixed tissue samples, the animals were transcardially perfused with PBS-heparin solution (10U/ml) followed by 4% paraformaldehyde and spinal cords were subsequently dissected.

### Immunofluorescence

Lumbar regions at L4-L5 levels were sectioned at 70-µm thickness on a Leica VT 1000S vibratome. Immunostaining procedure was carried out as previously described[86].

### Ethical statement

All experiments involving live animals were approved by the Institutional Animal Care and Use Committee at Columbia University Irving Medical Center.

### Reporting summary

Further information on research design is available in the Nature Portfolio Reporting Summary linked to this article.

## Data availability

All reagents generated in this study will be deposited to Addgene. Raw sequencing reads from the chaperone screen have been uploaded to the NCBI SRA under BioProject PRJNA769721 (SUB10508463). Raw sequencing reads from the orfeome screen have been uploaded to the NCBI SRA under BioProject PRJNA769721 (SUB10508562). Raw sequencing reads from the deep mutational scanning have been uploaded to the NCBI SRA under BioProject PRJNA769721 (SUB10503160). Raw sequencing reads from the RNA-seq have been uploaded to the NCBI SRA under BioProject PRJNA769721 (SUB10426285). Hyperlink: NCBI SRA PRJNA769721. Uncropped blots are provided as Supplementary Figs. The source data underlying all graphs presented within the main text and Supplementary Figs. are

provided as a Source Data File. Source data are provided with this paper.

## Code availability

Code used for analysis of the screening approach is available at the Chavez group github account alejandrochavezlab/MultiplexNDD [10.5281/zenodo.16276180]. Custom scripts used for analysis of hiPSCs derived motor neurons are available [https://github.com/CUIMC-Confocal/burguete].

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

## Acknowledgements

We thank members of the Chavez lab for helpful discussions and insights regarding the project. Aaron Gitler provided initial insight and a

number of yeast disease models, for which we are immensely grateful. We thank the ALS Stem Cell Core Program at Columbia University Irving Medical Center and its members, Hynek Wichterle, Jon Costa, Emily Lowry, and Niraj Ramsamooj for helpful discussions regarding cell-based assays and for valuable reagents. Matt Harms provided helpful discussion regarding DNAJB6 patient mutations and insight into inter-pretation of DMS results. Edwin Chan provided the SoxN plasmids. Richard Gardner provided the PAB1 poly-alanine plasmids. Priya Bane-rjee provided insight into the phase separation properties of DNAJB6 and FUS. Divyansh Argawal provided helpful discussions regarding statis-tical analysis of the screening platform. Debbie Hong and Sho Iketani provided valuable guidance on how to perform our deep mutational scanning studies. A.C. is supported by a Career Awards for Medical Scientists from the Burroughs Wellcome Fund and a Therapeutic Idea Award (#AL190073) from the DoD, and a fellowship award to the laboratory from Project ALS. S.J.R is supported by NIH grant F31NS111851. J.W. is supported by NSF grant 2113646. N.Z. is supported by NIH grant 2R01HG006137-10. Mouse work was supported by the National Institute of Neurological Disorders and Stroke by NIH grant 5R01NS106236 to N.A.S. This work was supported by grants from Canadian Institutes of Health Research (406915 Foundation Grant and Canadian Consortium on Neurodegeneration in Aging Grant), Wellcome Trust Collaborative Award 203249/Z/16/Z, US Alzheimer Society Zenith Grant ZEN-18-529769, Alzheimer Society of Ontario Chair in Alzheimer's Disease Research (PStGH); Cambridge Trust and Wolfson College (C.W.C); Wellcome Trust (065807/Z/01/Z) (203249/Z/16/Z), the UK Medical Research Council (MRC) (MR/K02292X/1), and Michael J Fox Foundation (16238 and 022159)(G.S.K.S.); the European Research Council (#804581) and Alzheimer's Society (#AS-PhD-19a-016) (J.A.V. and P.E.), Wellcome Trust (203249/Z/16/Z) and UKRI (10061100 and 10059436) (M.V.); the Dutch Ministry of Education—Sector Plan Beta for science and technology (FSR), ERC grant DiProPhys (agreement ID 10100161, T.P.J.K.); Global Research Technologies Novo Nordisk A/S (H.A., T.P.J.K.); the European Research Council under the European Union's Seventh Framework Program (FP7/2007-2013) through the ERC grants PhysProt (agreement no. 337969; T.P.J.K.). This work was sup-ported by a multiyear ALS Association award (21-IIA-571) to A.S.B. This study used the resources of the Herbert Irving Comprehensive Cancer Center Confocal and Specialized Microscopy Shared Resource, funded in part through NIH/NCI Cancer Center Support Grant P30CA013696. We are grateful to Dr. Barbara Corneo and Dario Sirabella for motor neuron differentiation.

## Author contributions

A.C. conceived the project. A.C. and StGH supervised the project. A.C., G.S.K.S., J.A.V., T.P.J.K., M.V., F.S.R and P.StGH. planned and designed experiments. S.J.R. and A.C. performed yeast multiplexed screening. S.J.R. and L.H.H. performed secondary validation testing. S.J.R. and J.S. performed and conducted analysis on deep mutational scanning approaches. P.K. assisted with follow-up characterization of DNAJB6 within yeast and mammalian cells. V.K., M.M.A., and N.A.S. performed in vivo studies examining the effect of DNAJB6 in ALS-FUS mice. S.J.R. and S.M. made cell lines. S.J.R. performed mammalian cell culture assays. S.Q. and J.N-A. purified and performed in vitro condensation and FRAP studies with DNAJB6 and FUS. C.W.C., N.F.L., S.K., C.F.K., and G.S.K.S. performed FLIM experiments. H.A. and T.P.J.K performed the tie line experiments. P.E. and J.A.V, performed the pFTAA fluorescence experiments. A.M. and M.V. performed the AFM and FTIR experiments. X.L. and F.S.R. performed AFM-IR experiments. S.J.R., N.Z., and J.W. designed and performed analysis. For hiPSC derived motor neuron studies T.S. and Y.C. developed the code for analyzing and simulating colocalization and performed the statistical analysis. Y-C.L. contributed to the conception, design, and acquisition of imaging data for Airyscan experiments. A.S.B. conceived and designed the colocalization analysis experiments in hiPSC derived motor neurons, acquired, analyzed and interpreted the data, and wrote the corresponding parts of the manu-script. S.J.R., P.StGH., and A.C. wrote the manuscript with input from all authors.

## Competing interests

A.C. and S.J.R. are inventors on a patent application submitted on the screening technology described in this work. The remaining authors declare no competing interests.

## Additional information

**Samuel J. Resnick**[1,2,3], **Seema Qamar**[4,5], **Pushya Krishna**[6], **Vladislav Korobeynikov** ⦿[1,7], **Hannes Ausserwoger** ⦿[5], **Alyssa Miller**[5], **Pietro Esposito** ⦿[8], **Juan A. Varela** ⦿[9], **Jenny Sheng**[1,3], **Lei Haley Huang** ⦿[1], **Jonathon Nixon-Abell** ⦿[4], **Schuyler Melore** ⦿[1], **Chyi Wei Chung** ⦿[10], **Nino F. Läubli** ⦿[10], **Sofia Kapsiani** ⦿[10], **Xuecong Li** ⦿[11,12], **Jingshu Wang**[13], **Nancy Zhang** ⦿[14], **Mahabub Maraj Alam**[7,15], **Alondra S. Burguete** ⦿[16], **Theresa C. Swayne**[17,18], **Yanyan Chen**[18], **Ya-Cheng Liao**[19], **Neil A. Shneider**[7,15], **Michele Vendruscolo** ⦿[5], **Tuomas P. J. Knowles** ⦿[5,20], **Clemens F. Kaminski** ⦿[10],

Francesco Simone Ruggeri ⓘ [11,12], Gabriele S. Kaminski Schierle ⓘ [10], Peter St George-Hyslop ⓘ [5,16,21] ✉ &
Alejandro Chavez ⓘ [6] ✉

[1]Department of Pathology and Cell Biology, Columbia University Irving Medical Center, New York, NY, USA. [2]Medical Scientist Training Program, Columbia University Irving Medical Center, New York, NY, USA. [3]Integrated Program in Cellular, Molecular, and Biomedical Studies, Columbia University Irving Medical Center, New York, NY, USA. [4]Cambridge Institute for Medical Research, Department of Clinical Neurosciences, University of Cambridge, Cambridge, UK. [5]Yusuf Hamied Department of Chemistry, Centre for Misfolding Diseases, University of Cambridge, Lensfield Road, Cambridge, UK. [6]Department of Pediatrics, University of California San Diego, La Jolla, CA, USA. [7]Center for Motor Neuron Biology and Disease, Columbia University Irving Medical Center, New York, NY, USA. [8]School of Biology, University of St Andrews, St Andrews, UK. [9]School of Physics and Astronomy, University of St Andrews, St Andrews, UK. [10]Department of Chemical Engineering and Biotechnology, University of Cambridge, Philippa Fawcett Drive, Cambridge, UK. [11]Laboratory of Organic Chemistry, Stippeneng 4, 6703 WE, Wageningen University & Research, Wageningen, Netherlands. [12]Physical Chemistry and Soft Matter, Stippeneng 4, 6703 WE, Wageningen University & Research, Wageningen, Netherlands. [13]Department of Statistics, The University of Chicago, Chicago, IL, USA. [14]Department of Statistics, University of Pennsylvania, Philadelphia, PA, USA. [15]Department of Neurology, Eleanor and Lou Gehrig ALS Center, Columbia University Irving Medical Center, New York, NY, USA. [16]Department of Neurology, Taub Institute for Research on Alzheimer's Disease and the Aging Brain, Columbia University Irving Medical Center, New York, NY, USA. [17]Department of Pathology and Cell Biology, Columbia University Irving Medical Center; Columbia University Irving Medical Center, New York, NY, USA. [18]Confocal and Specialized Microscopy Shared Resource in the Herbert Irving Comprehensive Cancer Center, Columbia University Irving Medical Center, New York, NY, USA. [19]Department of Biochemistry and Molecular Biophysics, Columbia University Medical Center the Taub Institute for Research on Alzheimer's Disease and the Aging Brain, Columbia University Medical Center, New York, NY, USA. [20]Cavendish Laboratory, Department of Physics, University of Cambridge, J J Thomson Ave, Cambridge, UK. [21]Department of Medicine, Division of Neurology, University Health Network and University of Toronto, Toronto, ON, Canada. ✉e-mail: ps2764@cumc.columbia.edu; alc042@health.ucsd.edu

