## [Transparent Peer Review file · Nature Communications]

Multiplex Neurodegeneration Proteotoxicity Platform Reveals DNAJB6 Promotes non-toxic FUS Condensate Gelation and Inhibits Neurotoxicity

Corresponding Author: Dr Alejandro Chavez

Version 0:

Reviewer comments:

Reviewer #1

(Remarks to the Author)

In this manuscript, Resnick and co-authors developed a multiplex platform using barcoded yeast strains to simultaneously screen genetic toxic suppressors of 29 proteins that misfold in neurodegenerative diseases. A custom-built analysis pipeline was developed for data analysis. Using this method, the authors identified and validated the human HSP40 co-chaperone, DNAJB6, as a toxicity rescuer for FUS, TDP-43, and hnRNPA1. The authors proposed that the mechanism of the rescue of FUS toxicity is through maintaining FUS in a loose gel-like state that prevents its fibrilization. Finally, truncation mutants were tested and deep mutational scan was performed to further understand the mechanism of the rescue activity of DNAJB6. The multiplex platform developed in this manuscript have the potential to be broadly used to test other genetic modifiers of toxic proteins. The deep mutational scan method could also be useful to understand the mechanism and sequence space of other genetic modifiers of toxic proteins. The identification of DNAJB6 as a toxicity modifier for ALS associated RBP raises the possibility that DNAJB6 represents a generalized chaperone for a number of clinically relevant misfolded species. Further characterization of DNAJB6's activity could be done both in test tube and in mammalian cell model to confirm its mechanism and activity.

1. In line 40, it is mentioned that "...were probed against more than a thousand genetic modifiers.". In the main text, only the results of a targeted library of 132 molecular chaperones, 62 of which were from yeast and 70 from humans (line 131) were reported. I am wondering, what were the rest of the genetic modifiers that were tested.
2. One limitation of the study, as mentioned by the authors, is the failure of human genes to function in yeast cells. I would appreciate a discussion on alternate methods that could overcome this limitation.
3. DNAJB6 has been previously shown to suppress the aggregation of polyglutamine repeat containing proteins, but did not show rescue of the ATXN1-Q85 in the current screen. It is curious whether DNAJB6 suppressed the aggregation of ATXN-Q85 in the current study.
4. It would be helpful to confirm by fluorescence imaging that DNAJB6 can reduce the aggregation of FUS, TDP-43, and hnRNPA1 in the yeast models used.
5. In Figure 2B, there is a secondary band appear in the blot after TDP-43 transfection. Is this band the transfected TDP-43 and is it included in the quantification of TDP-43 aggregation?
6. The authors have shown that the amount of RBP protein in SDS-insoluble fraction is closely related to the RBP protein expression level. Therefore, it might be helpful to include a lane of the total RBP protein in the blots. This could be particularly helpful in some of the blots. For example, in Figure 2 G,H,I, it appears that in the KO#1 line, the total protein level of FUS, TDP-43, and hnRNPA1 is increased compared to the control line.
7. Fluorescence life time is very sensitive to the environment. Is it possible that other than the change of material state (local packing), the presence or binding of DNAJB6 can change fluorescence life time? This is especially curious for the 24 hours data, when FRAP experiment did not show difference in material state.
8. In order to validate that DNAJB6 is directly interacting with RBP for function, binding constant could be determined or pulldown assay could also be used to show direct interaction.
9. For the DNAJB6 mutants that have decreased activity in rescuing RBP toxicity, the authors attribute the mechanism to reduced chaperon activity or binding. It would be necessary to rule out the possibility that these mutants are more toxic to WT yeast than WT DNAJB6. For example, based on Figure S17A, deltaG/F domain is more toxic to yeast than the FUS alone strain. Therefore, there might be added toxicity from the mutants.

10. In Figure S13, RNA-seq demonstrates that DNAJB6 is significantly upregulated in response to FUS or TDP-43 overexpression in HEK293T cells. It would be helpful if this could be confirmed by western blot.

11. The authors showed that DNAJB6 can rescue FUS, TDP-43, and hnRNPA1 toxicity in yeast cells and decrease the insoluble fraction in HEK293 cells. It would be helpful to confirm the toxic reduction activity of DNAJB6 in other model systems. For example, does DNAJB6 rescue toxicity of these RBPs in HEK293 cells?

12. Similarly, in the yeast model, DNAJB6 was expressed without human HSP70. I am curious in mammalian cells, does DNAJB6's activity depends on the expression/activity of HSP70?

Minor points:

13. "liquid liquid droplet" has been used in a couple of places in the manuscript and it is not an accurate description. It should be replaced with either "liquid-liquid phase-separated droplets" or "liquid droplets".

14. Figure 3F is missing the label for Y axis.

15. In Figure S15, scale bar is missing from A.

16. In Figure S15, the droplets in B are very small and also appear to be moving, which prevents a detailed characterization such as circularity measurement or FRAP experiment to confirm the liquid nature of the droplets. I am wondering if an image with clearer droplets would be provided.

17. For the same reason as mentioned in point 6, the amount of FUS plasmid used for transfection should be mentioned for Figure 4D.

Reviewer #2

(Remarks to the Author)

Authors of the manuscript, "A multiplex platform to identify mechanisms and modulators of proteotoxicity in neurodegeneration," developed a screening platform to probe the interactions between neurodegenerative disease (ND)-associated proteins and potential toxicity rescuers in yeast cells. From this platform, DNAJB6 was found to modify the solubility and toxicity of proteins implicated in neurodegeneration, such as FUS and TDP-43. DNAJB6 was also demonstrated to co-phase separate with these proteins and prevented their fibrillation. Using a deep mutational scan, specific residues in the glycine/phenylalanine and serine rich regions were identified to reduce or enhance DNAJB6 activity.

Development of an efficient, extensive screening technique that can be used to characterize interactions in a living system is useful for finding potential therapeutic avenues, especially for neurodegenerative diseases. The text thoroughly discusses the successes and shortcomings of their platform; the figures are detailed and visually appealing. Overall, the paper is well-written, and the developed screening platform shows promise in identifying potential rescuers of ND-associated protein toxicity.

However, there are some concerns that the authors should consider:

1. The Results section mentions that DNA-barcoded strains were pooled 'based on toxicity'. Is there a measure of toxicity or a system of ranking the toxicity of ND-associated proteins in yeast? If so, is the toxicity similar to what would be observed in humans?

2. There is no evidence that these ND-associated proteins misfold in yeast. The burden of expressing exogenous proteins may be significant enough to cause the observed toxicity. This type of toxicity may be reversed by the presence of protein chaperones, but there is no evidence that rescuing toxicity in yeast is related to rescuing protein misfolding in ND.

3. In the manuscript, 'toxicity' is indirectly measured by the 'growth' rate, and 'rescue' is relative to toxicity. In other words, 'rescue' is assessed by the change in yeast abundance. The data in Fig. 1 b and 1e express 'rescue' as Log2 fold change. It would be useful to include a representative raw data in the supplementary information to depict the general landscape of the effects of ND-associated protein expression have on yeast growth.

4. The assumption that toxicity caused by ND-associated proteins is driven by proteins misfolding follows the toxic gain-of-function disease model; however, ND toxicity may also be caused by a loss-of-function or other disease pathways. Would the results from this screening platform be able to address other forms of ND pathology?

5. The yeast chaperones were demonstrated to be more effective at rescuing proteotoxicity. Does this suggest that the toxicity observed is yeast-specific? If these screens/experiments were conducted in other cell lines, would there be a significant difference in the results? How translatable are these results to human cells?

6. It is unclear why the data in Fig. 1d and 1g were selected from the statistically significant interactions in Fig. 1c and 1f. Why were the most significant interactions from the screening platform not chosen for validation?

7. It may be useful to show negative controls when validating the screening results (i.e., the relative growth of statistically non-significant samples from Fig. 1c and 1f)

8. Fig. 1e shows that the highest fold change was observed with the interaction between DNAJB6 and TDP-43, followed by FUS and then hnRNPA1. However, Fig. 2a shows the reverse: DNAJB6 rescues hnRNPA1 the most, followed by FUS and then TDP-43. Furthermore, only one of these interactions (TDP-43) were deemed as statistically significant in Fig. 1f. Why was DNAJB6 selected amongst other chaperones when most of its interactions were not statistically significant (Fig. 1f)?

What is the difference between 'suspected interactions' and false negatives? How applicable is this screening platform if the sensitivity is reported to be only 66%?

9. There is no significant difference in the AFM-IR data of FUS versus FUS+DNAJB6. Other techniques (e.g., electron microscopy, far UV CD spectroscopy, ThT fluorescence-based assays) would more accurately show DNAJB6 effect, if any, on FUS aggregation.

10. Microscopy images of irregularly shaped condensates in Fig. 3c do not necessarily prove that fibrils formed. Condensate formation and LLPS do not always result in fibrillation. Other experiments must be performed to confirm whether fibril formation has occurred.

11. There is evidence that FUS + DNAJB6 may form loose, gel-like structures, but fibrils can still be present in gel-like structures. The AFM-IR data showed an increase in intermolecular parallel and anti-parallel β -sheets in the FUS + DNAJB6 (Fig. 3f). Fibrils contain β -sheet structures that can be parallel or anti-parallel; therefore, more experiments need to be conducted to show whether fibrils are present in the FUS and FUS+DNAJB6 samples.

12. Fig 3a suggests that DNAJB6 affects LLPS characteristics of FUS (i.e., FUS+DNAJB6 LLPS is slower and results in smaller droplets), but there is no data that suggests that aggregation is affected. Kinetics data is required to support the hypothesis that DNAJB6 prevents fibrillation.

13. (SI Fig. 15) DNAJB6 does not phase separate in high salt concentrations. FUS is an RNA-binding protein, and nucleic acids can be conceptualized as long polyanions. How will the presence of RNA affect the phase separation of DNAJB6 alone versus DNAJB6 + FUS?

14. The manuscript mentions that FUS and DNAJB6 "directly interact" (line 319) but there is no experiment that proves that these two proteins bind. Co-localization demonstrates the two proteins temporally occupy the same vicinity; therefore, additional experiment is needed to provide binding and direct interaction between FUS and DNAJB6.

Reviewer #3

(Remarks to the Author)

I looked only the AFM-IR experiments and they were carried out carefully. I trust the results. The data collected support the conclusions.

One could argue that AFM-IR probes the surface, and the protein aggregates at the surface might be different than in the bulk of the aggregate. So one point that the authors should mention is the fact that AFM-IR probes the sample down to a limited depth, and therefore I would like that you mention this depth and compare it to the size of the observed aggregates.

A remark at the lines 275-..., Intriguingly.... The fact that you observe more aggregates in one case does not mean that there are more in solution, since what you observe are the aggregates that are adsorbed on the surface and not those which are in solution: this is especially the case after 20 minutes where the aggregates in the two experiments are almost the same size, in this case one should have the same number. There might be an effect of difference in adsorption on the surface for FUS and FUS-DNAJB6 aggregates.

For Figure 3a,b and c, the authors should mention with which technique the different sets of data were obtained.

Reviewer #4

(Remarks to the Author)

Resnick et al. develop a yeast-based platform in which they express one of 29 proteins associated with misfolding and neurodegenerative disorders, along with a second protein that may rescue the misfolding. These additional proteins initially include 62 yeast chaperones and 70 human ones, and subsequently ~900 human proteins from the hORFeome library. These many pairings result in 120 interactions called significant from the chaperone set and 54 from the hORFeome set. Although redoing – and extending – an experiment done multiple times over the last two decades, this work is an important accomplishment that allows consistent comparisons of disease-associated proteins and potential modifiers derived from disparate models. Furthermore, the platform employed here leverages redundant barcodes for each pairing to provide additional confidence in the results. Because the platform allows high throughput, Resnick et al. in several cases also screen mutant versions of misfolding proteins that are known to increase the probability of neurodegenerative disease. Finally, the work identifies the human HSP40 chaperone DNAJB6 as a rescuer of multiple RNA-binding proteins that are aggregation prone, and provides analysis of the mechanism by which this chaperone acts.

The results represent a large body of work that is done at a high standard. The platform will be of considerable value throughout the community studying these diseases, and its description and the results from these screens warrant publication in a broad-based journal.

My comments are generally minor and mostly pertain to the presentation.

1. The Abstract leaves out key information – e.g. that the platform is yeast-based – that would seem to be helpful to the

reader, and emphasizes a large number (“more than a thousand genetic modifiers”) that is not representative of the bulk of the results presented. The Abstract might better state some of the specific results; for example, that the 29 models were tested against 62 yeast and 70 human chaperones, and might provide some indication of what were the key findings besides those related to DNAJB6.

2. I could not find a list of what the 29 proteins are that undergo misfolding in neurodegeneration and that form the basis for the work. This list could be within Fig. 1a, but this panel shows 24 proteins associated with disease, 2 yeast proteins not associated with neurodegeneration, and 9 genetic variants of the 24 disease proteins.

3. 29 proteins are tested against 132 chaperones, which makes 3,828 combinations, plus controls. The manuscript cites 5,850 interactions (line 137) by lumping together models and controls, which are not clearly defined. There should be more precision in the descriptions of the controls and the pairings.

4. It is not obvious to me why the hORFeome data are included, other than to point to the potentially large scale (“~35,000 interactions”) of the platform. This experiment is described in a single paragraph on p. 8 (lines 202-215), and the 54 confirmed interactions are not discussed beyond a list of entries in Supp. Tables. These data should either be dealt with in a substantive manner or deleted.

5. Fig. 2 shows gels from a protein aggregation assay to compare the amount of SDS-insoluble species that are solubilized in urea with or without DNAJB6. The manuscript text (lines 234-241) and figure legend are sparse about the details of these experiments. What is the high molecular weight material in Fig. 2b, g and is it included in this comparison? Where is the Ponceau S staining for 2b, c, d and 2g, h, i that is used for normalization? Why are full gel lanes not shown for 2c, f, h, i, rather than only limited slices of the gels? This is an important experiment to validate the DNAJB6 rescue, and either Fig. 2 or a supplemental figure should show a complete depiction of what was found, including whole gels.

6. Sup. Fig. 13 shows that DNAJB6 was upregulated when FUS or TDP-43 was overexpressed. What are the other RNAs that are even more upregulated than DNAJB6?

7. Sup. Fig. 17a shows that deletion of domains from DNAJB6 prevents the protein from rescuing FUS toxicity in yeast. Why are the comparisons between growth in the presence of the deletion vs. mCherry, rather than growth in the presence of the deletion vs. the intact DNAJB6? Also, line 334 points to a result in Sup. Fig. 17b that should refer to Sup. Fig. 17a.

Version 1:

Reviewer comments:

Reviewer #1

(Remarks to the Author)

In the revised manuscript, Resnick and co-authors have adequately addressed my previous comments and concerns, making the manuscript suitable for publication. The additional data and reworded text further strengthen and clarify the conclusions.

(Remarks on code availability)

Reviewer #2

(Remarks to the Author)

The authors of, “Multiplex Neurodegeneration Proteotoxicity Platform Reveals DNAJB6 Promotes non-toxic FUS Condensate Gelation and Inhibits Neurotoxicity” have addressed many of my previous concerns to a satisfactory degree. The authors provided the data I had asked for in supplementary tables and figures. They were able to answer my question regarding the difference between observed L2FC from the initial screening platform and individual validation in subsequent assays and justified their investigation of DNAJB6 with many additional experiments. These experiments addressed my concerns regarding the validity of DNAJB6 effects, but also led to a few minor questions that the authors may choose to address:

1. It appears that the results from the screening platform designed by the authors require additional optimization steps before identifying strong candidates. Are the authors interested in further developing their screening platform? Do they have any ideas on how to improve this platform to make it more applicable for widespread laboratory use?
2. The results of aging condensates (either FUS alone or FUS + DNAJB6) are reported only up to 48 hours for all experiments. What happens beyond that timepoint? Is there a reason why data collection stops after two days?
3. Do the authors propose DNAJB6 as a potential therapeutic for neurodegeneration related to FUS toxicity? Have the authors investigated the effects of overexpression of DNAJB6 in neuronal cell models?
4. The bulk of biophysical characterization (AFM-IR, FLIM, etc) data study DNAJB6 and FUS interaction. What is the interaction between DNAJB6 and TDP-43/hnRNPA1? Are the gelation effects observed between DNAJB6 and FUS observed with TDP-43 and hnRNPA1?

(Remarks on code availability)

Reviewer #4

(Remarks to the Author)

The authors have done a careful and thorough job of addressing the concerns of the referees, and where they have declined to perform additional experimentation they provide credible rationales for doing so. They also generated new data in a mouse model that bolster the validity of the yeast screens. I have no additional concerns.

(Remarks on code availability)

Reviewer #5

(Remarks to the Author)

NCOMMS-22-42799A

This study presents a novel yeast genetic screening platform to identify modifiers of neurodegeneration-linked proteins, focusing on 32 NDD-associated targets. Key findings highlight the human chaperone DNAJB6 as a potent suppressor of toxicity in ALS/FTD models. The authors demonstrate DNAJB6's ability to alter phase separation dynamics of FUS protein condensates, preventing fibrilization, and validate its neuroprotective effects in a FUS-ALS mouse model.

Figure 8 & Supplementary Figure 31 Assessment

Figure 8: Robustly illustrates DNAJB6's in vivo efficacy, showing prevention of motor neuron loss and reduced microglia activation in FUS-ALS mice. Data presentation is clear, with appropriate controls and statistical rigor.

Supplementary Figure 31. Validation of DNAJB6 expression upon exposure to ChAT-Cre. Data presentation is clear, with appropriate controls and statistical rigor.

(Remarks on code availability)

Color coding for this document: Reviewer comments in **BLACK**, our responses in **BLUE**, words and edits that are in the manuscript are in **PURPLE**.

OPENING REMARKS TO THE REVIEWERS:

Dear Reviewers,

We want to thank you for your very helpful comments, which we address individually below.

We also want to highlight additional data that we have collected that is not related to an individual comment, but which provides strong support for the relevance of DNAJB6 as a modifier of FUS toxicity.

In the intervening months, we have tested the effects of DNAJB6 in a mouse model of ALS-FUS. These data show a marked rescue of motor neuron loss and a decreased inflammatory response in ALS-FUS animals that overexpress DNAJB6. We have included these data as Figure 8 within our revised manuscript as they further demonstrate the power of our platform and the translational relevance of the hits that are identified. Updates to the manuscript and figures for these animal data are included below for convenience.

Additions to results section:

[Line 561]

DNAJB6 rescues motor neuron loss and associated microgliosis in an animal model of ALS-FUS

Next, we set out to determine whether DNAJB6 could rescue the age-dependent motor neuron loss and associated microgliosis within an ALS-FUS mouse model. This knock-in model expresses two mutant forms of FUS (FUS P517L and FUS Δ 14) which mimic early-onset patient variants. These animals show a consistent loss of motor neurons and increased microgliosis at 6 months of age, and have served as a preclinical model for therapeutic intervention with a FUS-targeting antisense oligonucleotide⁸⁹. To determine the effect of overexpressing DNAJB6 within diseased motor neurons, an AAV vector was generated which contained a CAGGS promoter regulating DNAJB6 with an intervening lox-stop-lox (LSL) transcriptional termination cassette. When the LSL cassette is present, DNAJB6 is not transcribed due to the multiple copies of a transcriptional termination sequence within the LSL cassette. In the presence of Cre, which is expressed from the ChAT promoter in this ALS-FUS model, the LSL cassette is excised and DNAJB6 is expressed in ChAT+ motor neurons. DNAJB6 was then packaged into AAV9 and delivered via intracerebroventricular injection to newborn ALS-FUS or control pups. As an additional control, a cohort of animals was injected with a GFP containing virus. Animals were then analyzed at 6 months of age and samples from the L4-L5 region of the spinal cord were processed for immunostaining. Robust Cre-dependent expression of DNAJB6 was confirmed in the spinal motor neurons (Sup. Fig. 31). In the control animals there was no difference in the numbers of motor neurons or staining for the microglial activation marker Iba1 between DNAJB6 vs GFP treated animals (Fig. 8). Consistent with the expected phenotype, ALS-FUS animals that received the GFP control virus showed a marked ~20% decrease in motor neurons and a simultaneous increase in Iba1 positive microglia. In contrast, the overexpression of DNAJB6 in ALS-FUS animals was protective, resulting in preserved motor neuron numbers and no evidence of abnormal microglial activation (Fig. 8).

Additions to the discussion section:

[LINE 651]

The finding that the overexpression of DNAJB6 within an ALS-FUS mouse model rescues motor neuron loss and the associated microgliosis, reinforces the value of using simple cellular systems to rapidly search for genetic modifiers of disease. Additional work will be required to verify our initial findings and understand the long-term safety and efficacy of overexpressing DNAJB6 or enhanced mutants with more specific and improved anti-FUS aggregation activity. Furthermore, given that DNAJB6 reduces the toxicity and misfolding of other ALS-associated RBPs, such as TDP-43, it will be interesting to determine if DNAJB6 rescues ALS models that are driven by dysfunction in other RNPs.

Figure 8. DNAJB6 rescues FUS-mediated MN loss in a mouse model of ALS-FUS. a. Representative immunofluorescence staining of motor neurons (ChAT positive staining) and activated microglia (Iba1 positive staining) in control or ALS-FUS mice exposed to GFP or DNAJB6 containing constructs delivered via AAV9. Scale bar represents 50 microns. **b-c.**

Quantification of ChAT positive MNs and Iba1+ microglia at the L4-L5 spinal cord level normalized to the control mice provided GFP. All data are shown as mean \pm s.d. and were gathered from 4 independently exposed mice. Comparisons were conducted with one-way ANOVA; ns = not significant, ****P<0.0001.

Finally, we note that during the validation of the various plasmids for public distribution, a deletion at position S326 was identified in the EWSR1 gene. When compared to the wild-type ORF, the S326 deletion was found to cause similar toxicity levels. Additionally, several repeat-containing constructs were found to have altered lengths which has been documented in the updated supplementary materials provided with these revisions (Sup. Table 1).

REVIEWER #1 (REMARKS TO THE AUTHOR):

In this manuscript, Resnick and co-authors developed a multiplex platform using barcoded yeast strains to simultaneously screen genetic toxic suppressors of 29 proteins that misfold in neurodegenerative diseases. A custom-built analysis pipeline was developed for data analysis. Using this method, the authors identified and validated the human HSP40 co-chaperone, DNAJB6, as a toxicity rescuer for FUS, TDP-43, and hnRNPA1. The authors proposed that the mechanism of the rescue of FUS toxicity is through maintaining FUS in a loose gel-like state that prevents its fibrilization. Finally, truncation mutants were tested and deep mutational scan was performed to further understand the mechanism of the rescue activity of DNAJB6. The multiplex platform developed in this manuscript have the potential to be broadly used to test other genetic modifiers of toxic proteins. The deep mutational scan method could also be useful to understand the mechanism and sequence space of other genetic modifiers of toxic proteins. The identification of DNAJB6 as a toxicity modifier for ALS associated RBP raises the possibility that DNAJB6 represents a generalized chaperone for a number of clinically relevant misfolded species. Further characterization of DNAJB6's activity could be done both in test tube and in mammalian cell model to confirm its mechanism and activity.

We thank the reviewer for their astute summary of our work and its relevance to the field.

1. In line 40, it is mentioned that "...were probed against more than a thousand genetic modifiers.". In the main text, only the results of a targeted library of 132 molecular chaperones, 62 of which were from yeast and 70 from humans (line 131) were reported. I am wondering, what were the rest of the genetic modifiers that were tested.

We thank the reviewer for highlighting the use of our technology in both a smaller, targeted library of molecular chaperones and in a larger, unbiased screen, which together total to 1092 modifiers being investigated. The results of this larger screen are described in lines **202-215 of the originally submitted manuscript file** under the "**Secondary validation of multiplexed screening results**" heading with results of the screen and yeast validation presented in the supplementary information (Sup. Table 4 and Sup. Table 5) on our previous submission.

We acknowledge that this information is insufficiently highlighted in the previous version of the manuscript and have inserted a new section specifically highlighting these data called "**A large, unbiased screen of human ORFs identifies novel interactions,**" along with elaborating on our findings. We have added supplementary figures to support the previous supplementary tables that highlight the validation of rescue of proteotoxic species we observed in this screen. The hits highlighted from this larger screen are shown in **Supplementary Figure 13 and Sup. Tables 5-7**. We have also added discussion around some of the novel hits from this larger screen. The new section is pasted below:

[LINE 233]

A large, unbiased screen of human ORFs identifies novel interactions

To evaluate the performance of our platform against a larger, unbiased library of potential modifiers, we screened the hORFeome V8.1 human cDNA library against our pool. We chose to screen human ORFs not only to identify hits that are more likely to translate directly to mammalian models of disease, but also because few human genes had been tested before in this paradigm, suggesting that many novel interactions remain uncovered. Screening 899 members of the

hORFeome library enabled the examination of 40,455 interactions, representing ~9 times more interactions surveyed within our system than is typical^{7,8,11,14,15}. Screening and subsequent validation of this collection of rescuers resulted in the identification of 54 confirmed genetic interactions (Sup. Tables 5-7). As expected with a library comprising random human genes, the occurrence of genetic interactions was significantly lower (0.14%) compared to the curated molecular chaperone screen (2.9%). Furthermore, when compared to other unbiased screens overexpressing yeast ORFs instead of human genes our hit rate remains lower (previously reported hit rates of 0.24%-1.13% vs. 0.14%)^{8,10,11,14}. One of the main drivers of this difference is likely the failure of human genes to function in yeast cells, highlighting the benefits of using a rapid high-throughput, quantitative screening approach to explore this sparsely populated interaction space³⁷.

Despite a lower rate of interactions, rescuers were identified for a broad range of yeast models, including ones that did not have rescuers from the initial molecular chaperone screen (Sup. Fig. 13a). Several hits captured known interactions. For example, two 14-3-3 proteins, YWHAG and YWHAB were found to be rescuers of the ATXN1 polyglutamine (ATXN1-Q85) yeast model (Sup. Fig. 13b). Cytoplasmic 14-3-3 proteins bind ATXN1 via a motif centered around a phosphorylated serine at position 776 in ATXN1. This binding masks the ATXN1 NLS and diminishes its transport into the nucleus which is necessary for ATXN1 toxicity^{38,39}. Other rescuers of ATXN1-Q85 in our screen were RNPS1 and JMJD6, two proteins involved in RNA-splicing, which ATXN1 also actively participates in (Sup. Fig. 13b). Furthermore, both RNPS1 and JMJD6 were recently identified as interactors of ATXN1-Q85 via pulldown and BIO-ID approaches, respectively⁴⁰. We also identified two rescuers of the alpha-synuclein expressing model, one of which is a member of the Regulators of G protein signaling (RGS) family, RGS20 (Sup. Fig. 13c). RGS family proteins are structural proteins that accelerate the GTPase activity of α -subunits in G coupled-protein receptors, reducing the duration of downstream signaling⁴¹. A number of RGS family proteins have been implicated in the development of Parkinson's Disease including a study of RGS6 in which expression of RGS6 in substantia nigra pars compacta dopaminergic neurons suppressed Parkinson's Disease phenotypes in aged mice⁴². Within our larger screen, we also discovered several rescuers of the FUS yeast model (Sup. Fig. 13d). These rescuers included RAD23B, a ubiquitin-binding proteasome-shuttle protein localized to the nucleus that is suspected to undergo phase separation, HIST1H1E, a member of the H1 linker histone family with predicted disordered regions, TGIF2LX, a DNA-binding homeobox protein with predicted disordered regions, and OAS1, a stress granule localized 2'5'-oligoadenylate synthetase protein. TMEM106B, a protein

implicated in FTD^{43–46} and brain aging^{47,48} was strongly rescued by COX7B, a member of the cytochrome c oxidase complex involved in the mitochondrial respiratory chain (Sup. Fig. 13e)^{49,50}. Taken together, these results suggest that human proteins beyond molecular chaperones can be screened to identify novel interactions.

Supplementary Figure 13. Identification and validation of interactions from the hORFeome screen. **a.** Log2 fold change interactions between a select set of hits from the ORFeome screen and the models included in the pool. Manual validation in yeast of detected interactions for **b.** ATXN1-Q85 **c.** Alpha-synuclein **d.** FUS and **e.** TMEM106B. Data are shown as mean \pm s.d. for three biological replicates, independent outgrowths. Comparisons for **b.** and **d.** were conducted with ordinary one-way ANOVA and comparisons **c.** and **e.** were conducted with Welch's t test; * $P < 0.05$, ** $P < 0.01$, **** $P < 0.0001$.

2. One limitation of the study, as mentioned by the authors, is the failure of human genes to function in yeast cells. I would appreciate a discussion on alternate methods that could overcome this limitation.

This is a good point and one that is worthy of expansion. In the main text, we have elaborated on the types of interactions we observed in the ORFeome data. We have also included a more detailed discussion on the activity of human genes within yeast and have proposed methods that could overcome some of these limitations. The new discussion points within the edited manuscript are below:

[LINE 612]

We note that a limitation of our screening approach is that not all human genes will function appropriately in yeast. This may be due to the lack of a required protein partner that is present in

humans but absent in yeast. Alternatively, yeast may contain a homolog of the required partner that is unable to interact with the expressed human protein because of the evolutionary distance between yeast and humans. By performing studies where we simultaneously deliver a set of factors known to cooperate, it may be possible to overcome this limitation. In a similar fashion, work has also been done creating “humanized yeast” which have yeast genes replaced with their human counterpart and these may be a better cellular chassis in which to conduct our studies⁹⁰. In addition, codon optimization of each of the tested human genes for expression in yeast, could help to mitigate false negatives arising from the poor expression in yeast⁹¹. Finally, additional technical improvements to the platform, enabled by using more individually barcoded strains, would likely provide more sensitive detection of weaker interactions like those that may emerge from the expression of a human gene within yeast.

3. DNAJB6 has been previously shown to suppress the aggregation of polyglutamine repeat containing proteins, but did not show rescue of the ATXN1-Q85 in the current screen. It is curious whether DNAJB6 suppressed the aggregation of ATXN-Q85 in the current study.

The reviewer makes an interesting point. In our screen, we did not identify significant rescue of our ATXN1-Q85 model by DNAJB6. The reviewer also points to literature suggesting a role for DNAJB6 in suppressing the aggregation of polyQ containing proteins, which they find curious given our screening result. To address this question we manually tested the ability of DNAJB6 to rescue our ATXN1-Q85 construct alongside other polyQ containing proteins (see **Figure i** below). From these experiments, we were able to verify the result from our multiplex screen, with DNAJB6 failing to rescue ATXN1-Q85 growth defects, verifying that the lack of rescue within our screen is not due to a failure of the multiplex screening approach. A second polyQ model, a fragment of exon 1 from the HTT protein fused to a 97 amino acid polyQ stretch, also failed to be rescued by DNAJB6. In contrast, a 105 amino acid polyQ stretch fused to a FLAG epitope tag was rescued by DNAJB6 overexpression. While the biological basis of these findings require additional investigation, there is ample evidence in the literature that the sequence context surrounding the polyQ stretch can drastically alter its toxicity (presumably by changing its misfolding propensity or the conformation of the aggregated intermediate), potentially explaining why DNAJB6 rescues some polyQ proteins but not others (1-7 REF).

PolyQ Proteins - DNAJB6

Figure i. DNAJB6 rescue of polyQ containing proteins. PolyQ models were subject to rescue with either EYFP (control) or DNAJB6. Four independent biological replicates were assayed for each condition. All data are shown as mean \pm s.d. Independent t-tests with Welch's corrections were performed to calculate significance. ns - not significant, * $P < 0.05$, ** $P < 0.01$, **** $P < 0.0001$.

Reference:

1. Gillis, J. *et al.* The DNAJB6 and DNAJB8 protein chaperones prevent intracellular aggregation of polyglutamine peptides. *Journal of Biological Chemistry* **288**, 17225–17237 (2013).
2. Hageman, J. *et al.* A DNAJB chaperone subfamily with HDAC-dependent activities suppresses toxic protein aggregation. *Molecular Cell* **37**, 355–369 (2010).
3. Rodríguez-González, C., Lin, S., Arkan, S. & Hansen, C. Co-chaperones DNAJA1 and DNAJB6 are critical for regulation of polyglutamine aggregation. *Scientific Reports* **10**, (2020).

4. Kakkar, V. et al. The S/T-rich motif in the DNAJB6 chaperone delays polyglutamine aggregation and the onset of disease in a mouse model. *Molecular Cell* **62**, 272–283 (2016).
5. Månsson, C. et al. DNAJB6 is a peptide-binding chaperone which can suppress amyloid fibrillation of polyglutamine peptides at substoichiometric molar ratios. *Cell Stress and Chaperones* **19**, 227–239 (2014).
6. Duennwald, M. L., Jagadish, S., Muchowski, P. J. & Lindquist, S. Flanking sequences profoundly alter polyglutamine toxicity in yeast. *Proceedings of the National Academy of Sciences* **103**, 11045–11050 (2006).
7. Shen, K. et al. Control of the structural landscape and neuronal proteotoxicity of mutant Huntingtin by domains flanking the POLYQ tract. *eLife* **5**, (2016).

4. It would be helpful to confirm by fluorescence imaging that DNAJB6 can reduce the aggregation of FUS, TDP-43, and hnRNPA1 in the yeast models used.

We agree that greater insight into the biophysical behavior of FUS, TDP-43 and hnRNPA1 in yeast could inform the utility of such models. However, the validation of DNAJB6 as a rescuer of motor neuron loss in an ALS-FTD mouse model (**Figure 8**) in combination with the data generated in mammalian cells and with purified proteins, suggests that DNAJB6-mediated modulation of RBP toxicity in yeast is a sufficiently informative phenotype.

5. In Figure 2B, there is a secondary band appear in the blot after TDP-43 transfection. Is this band the transfected TDP-43 and is it included in the quantification of TDP-43 aggregation?

Thank you for bringing up this observation. This assay was developed based on a prior published protocol from the following paper in which high molecular weight (HMW) species of TDP-43 were also observed upon overexpression in mammalian cells:

Chen, H. J. et al. The heat shock response plays an important role in TDP-43 clearance: evidence for dysfunction in amyotrophic lateral sclerosis. *Brain* **139**, 1417-1432, doi:10.1093/brain/aww028 (2016)

This paper quantifies the abundance of HMW species which were observed at the same molecular weights in our study. We quantify all TDP-43 present in the urea lanes in our study. We have updated the methods and figure legends with this information and excerpts are pasted below:

In Figure 2 legend, the following statement has been added, **For TDP-43 blots, all bands at or above 43 kDa were included in quantification.**

In the methods, **For TDP-43 blots, high molecular weight (HMW) species were included in quantification of TDP-43 blots.**

6. The authors have shown that the amount of RBP protein in SDS-insoluble fraction is closely related to the RBP protein expression level. Therefore, it might be helpful to include a lane of the total RBP protein in the blots. This could be particularly helpful in some of the blots. For example, in Figure 2 G,H,I, it appears that in the KO#1 line, the total protein level of FUS, TDP-43, and hnRNPA1 is increased compared to the control line.

The reviewer brings up an interesting point regarding the total amount of RNA binding protein present and whether there is variability in their overall levels between the transfected DNAJB6 KO versus control lines. Unfortunately, a lane of the total RBP protein in the blots was not run as this protocol was derived from a previous study that did not run a total protein sample (1). However, we do believe that we can answer this concern with the data at hand. The below data demonstrates that endogenous levels of RBPs in the NTC and DNAJB6 KO lines are comparable.

We also went back and analyzed whether there was a difference in the amount of soluble RBPs between the NTC and KO lines when the RBPs are overexpressed. In all scenarios and comparisons between lines, there was no statistically significant difference between the levels of soluble RBP (line #1 FUS – $P= 0.772$, line #2 FUS – $P= 0.575$, line #1 TDP-43 – $P= 0.367$, line #2 TDP-43 – $P= 0.313$, line #1 hnRNPA1 – $P= 0.910$, line #2 hnRNPA1 – $P= 0.692$).

Reference:

1. Chen, H. J. *et al.* The heat shock response plays an important role in TDP-43 clearance: evidence for dysfunction in amyotrophic lateral sclerosis. *Brain* **139**, 1417-1432, doi:10.1093/brain/aww028 (2016)

7. Fluorescence life time is very sensitive to the environment. Is it possible that other than the change of material state (local packing), the presence or binding of DNAJB6 can change fluorescence life time? This is especially curious for the 24 hours data, when FRAP experiment did not show difference in material state.

We thank the reviewer for raising this point. We agree that FLIM measures the average duration that a fluorophore resides in the excited state before emitting a photon, and as a result can provide insights into the fluorophore’s environment and dynamics. As we describe in the revised manuscript and revised figure (now Fig 4c), we have recently shown that the drop in the fluorescence lifetime of fluorescent protein labelled FUS directly correlates with an increase in viscosity of FUS condensates (<https://doi.org/10.1101/2022.10.04.510756>). We show that FUS-only preparations cause a progressive and time-dependent reduction in fluorescence lifetimes (Fig. 4c, Supplementary Fig 21e-g). This indicates that, over time, FUS polymers become increasingly more viscous in the condensates – a result that is fully congruent with the microscopic observation that FUS-only preparations eventually form fibrillar aggregates (Fig 4a-b). In contrast, at all time points from 20 minutes to 48 hours, the FUS + DNAJB6 condensates demonstrate significantly longer fluorescence lifetimes that are similar to those of control experiments containing dispersed solutions of DNAJB6 and mEmerald which do not form condensates (Fig. 4c, Supplementary Fig 21e-g).

FRAP measures a different parameter, namely the ability of molecules in the dispersed phase to access the condensed phase. The FRAP assay has a floor effect, and by 24 hours, the ingress of new FUS molecules into either the gelled (FUS + DNAJB6) or the progressively aggregating (FUS only) condensates is so low that the FRAP assay is unable to distinguish the two states.

8. In order to validate that DNAJB6 is directly interacting with RBP for function, binding constant could be determined or pulldown assay could also be used to show direct interaction.

To address this question, we provide three types of data – Co-IP data from cells exogenously expressing FUS and DNAJB6 (Sup. Fig. 20), co-localization human iPSC derived neurons endogenously expressing FUS and DNAJB6 (Fig 3), and *in vitro* biophysical “tie-line” experiments (Fig 4d). Taken together, these experiments suggest that DNAJB6 forms transient, low avidity direct interactions with RBPs that prevent close packing of the RBP, and by doing so, prevents the homotypic inter- and intra-molecular interactions that drive aging of condensates from liquid droplet to aggregated states.

We thank the referee for the suggestion of attempting to measure the binding constant. We have thought about how to do this, but have come to the conclusion that it is not technically feasible for two broad reasons. First, the multivalent inter- and intra-molecular interactions between DNAJB6 and its target condensate-forming protein partners are likely to be both heterogeneous and complex, making conventional binding assays such as a bilayer interferometry inappropriate. Second, surface based approaches like bilayer interferometry require anchoring of one partner on a chip. If one or both proteins have a propensity to undergo LLPS upon crowding / concentration, their concentration on the chip surface will potentially cause artifactual condensation. Nevertheless, the evidence from the studies that we have done in Figs 3-6 point to transient, weak interactions that prevent FUS from physically interacting close enough to form homotypic FUS-FUS interactions necessary for aging into solid aggregates with increased amounts of inter-molecular anti-parallel beta sheet (Fig 6c).

In the next paragraphs we discuss each of these experiments in more detail.

We have previously shown that within a purified protein system containing only DNAJB6 and FUS, that DNAJB6 modifies the biophysical properties of FUS, suggesting a direct interaction between these two proteins. However, we did not provide similar *in vitro* data supporting a direct interaction between DNAJB6 and the two other highlighted RBPs in our manuscript, TDP-43 and hnRNPA1. To further characterize the mechanism underlying DNAJB6’s rescue of RBP misfolding, we performed a co-immunoprecipitation (Co-IP) assay between 3xFLAG-DNAJB6 and each of our target RBPs. In mammalian cells overexpressing 3xFLAG-DNAJB6 and each RBP, a clear interaction between DNAJB6 and both TDP-43 and hnRNPA1 was observed (Sup. Fig. 20; data pasted below for convenience). Our Co-IP was unable to demonstrate an interaction between DNAJB6 and FUS. It is possible that the nature of the interaction is transient and unstable, and so does not withstand our Co-IP conditions.

These co-IP data have been added to the manuscript with the following passage and Supplementary Figure 20.

The co-localization experiments performed on the endogenous FUS and DNAJB6 proteins in disease-relevant wild type human iPSC-derived motor neurons shows that a small percentage (~4%) of intracellular puncta contain both FUS and DNAJB6 (Fig 3; data shown below for convenience).

We then complemented this optical co-localization experiment with a novel biophysical approach that exploits tie-line methodology. This methodology characterizes the collective interactions between FUS and DNAJB6 in the condensed phase, and does so by examining changes in the concentration of FUS in the dilute phase after phase separation. These experiments indicate that DNAJB6 prevents the overall liquid-to-solid transition of FUS by replacing on-pathway homotypic FUS-FUS interactions with preferential heterotypic FUS-DNAJB6 interactions (Fig. 4d). This preferential interaction of FUS molecules with DNJAB6 rather than with other FUS molecules preserves the liquid / gel-like state of FUS, and prevents their progression into more solid states.

Supplementary Figure 20. Co-immunoprecipitation (Co-IP) assays probing interactions between DNAJB6 and various RBPs. **a.** Protein homogenates from HEK293T cells transfected with plasmids overexpressing untagged DNAJB6 (WT) or 3xFLAG-DNAJB6 and either FUS, TDP-43 or hnRNPA1 were used as the input for a Co-IP. **b.** Uncropped western blots for data shown in panel a.

Figure 3. FUS and DNAJB6 coincide in a subset of neuritic motor neuron granules. a-d. Immunostain of **a.** FUS, **b.** DNAJB6, **c.** FUS and DNAJB6 merge, in hiPSC human differentiated motor neurons. **Left panel:** A single Airyscan Z-stack section is shown, out of three 150 nm sections with continuous overlap between FUS and DNAJB6 (scale bar = 2 μm). **Right panel:** magnification of a FUS granule that co-labels with DNAJB6 (scale bar = 0.176 μm).

Figure 4. Biophysical characterization of FUS condensates with and without DNAJB6. a. DNAJB6 promotes formation of smaller spherical condensates that are stable during aging, and prevent formation of dense irregular spiculated / fibrillary condensates. FUS and FUS + DNAJB6 condensates at 20 minutes, 24 hours and 36-48 hours. FUS condensates form fibrillar aggregates when incubated for 36 hours. Scale bar = 10 μm at 20 min and 24 hours, scale bar = 20 μm at 36 hours. Data were obtained using epifluorescence microscopy. **b.** Quantification of condensate size. One micron squared minimum size cutoff was used when quantifying condensates. Average condensate size was determined for multiple condensates within several different fields of view. At 20 minutes, $n=343$ for FUS and $n=1,455$ for FUS + DNAJB6. At 24 hours, $n=792$ for FUS and $n=1,459$ for FUS + DNAJB6. Comparisons were conducted with Welch's two-sided t test. **** $P \leq 0.0001$ Plots are shown as mean \pm s.e.m. **c.** Fluorescence lifetime measurements (FLIM) of mEmerald-FUS condensates \pm DNAJB6 over time. The condensate-averaged fluorescence lifetime of mEmerald fluorophores linked to FUS significantly decreases over time, indicating FUS condensation and aggregation over time. However, in the presence of DNAJB6, FUS aggregation is prevented, as shown by the significantly higher fluorescence lifetime detected with DNAJB6. Non-parametric Kruskal-Wallis test with Dunn's multiple comparison where * is $P < 0.05$ and **** is $P < 0.0001$. Error bars denote \pm s.d. **d.** Tie line gradient analysis reveals preferential partitioning and interaction between DNAJB6 and FUS. (*left panel*) Intensity time trace of a phase separated FUS sample upon flowing through a confocal detection set-up using microfluidic technology (top) and corresponding intensity histogram (bottom). Bursts of high intensity correspond to condensates passing through the confocal volume. The baseline signal forms a peak in the intensity histogram and is used to extract dilute phase concentrations. For details, see Materials

and Methods. (*middle panel*) Dilute phase concentration changes of FUS with varying DNAJB6 concentrations allowing for determination of the tie line gradient value between FUS and DNAJB6 at the measured conditions. (*right panel*) DNAJB6 is recruited into FUS condensates, effectively spacing out aggregation promoting FUS-FUS interactions to induce liquid state preservation.

9. For the DNAJB6 mutants that have decreased activity in rescuing RBP toxicity, the authors attribute the mechanism to reduced chaperon activity or binding. It would be necessary to rule out the possibility that these mutants are more toxic to WT yeast than WT DNAJB6. For example, based on Figure S17A, deltaG/F domain is more toxic to yeast than the FUS alone strain. Therefore, there might be added toxicity from the mutants.

The reviewer raises a great point. To understand how mutations to DNAJB6 cause loss of rescue, we should investigate their effects on cell growth independent of FUS. To perform this analysis,

we individually characterized the toxicity of multiple DNAJB6 constructs. At the same time, we also analyzed the growth of the FUS model alone and a strain expressing the non-toxic control EYFP alone, to help provide reference points for the observed growth defects. As expected FUS causes significant toxicity, while wild type DNAJB6 alone induces a mild growth defect. Furthermore, the DNAJB6 Δ G/F construct when expressed alone in yeast also causes robust toxicity on its own. In line with this observation, many of the constructs that show a lack of FUS rescue (P96L, V99A, E102K) cause toxicity on their own. Interestingly, F89I, although being defective for FUS rescue, does not cause more toxicity than WT DNAJB6 when expressed alone, suggesting that it represents a true DNAJB6 variant with decreased activity.

As a whole, these results suggest that DNAJB6 mutants that show loss of RBP rescue may do so by gaining new toxicity independent of FUS (e.g. Δ G/F, P96L, V99A, E102K) or by losing their ability to block FUS mediated toxicity (F89I). We have added these data as **Supplementary Figure 28**, along with adding the following discussion in the text:

[LINE 531]

The loss of rescue may be driven by an inability of a given DNAJB6 variant to interact with their RBP clients or the DNAJB6 variant gaining an intrinsic toxicity that then synergizes with the RBP toxicity. Expressing several of our examined DNAJB6 variants alone in cells revealed that the loss of rescue was due to both of the mentioned possibilities (Sup. Fig. 28). Future studies will be required to understand the mechanism driving this variable toxicity between mutants.

Supplementary Figure 28. Mutations in DNAJB6 can affect its intrinsic toxicity and ability to rescue RBP toxicity. Individual growth assays for several DNAJB6 variants expressed alone in yeast cells are shown. FUS and EYFP, a toxic and non-toxic protein, respectively were tested at the same time to provide a comparator. Many but not all DNAJB6 variants that show a loss of FUS and TDP-43-mediated rescue also show enhanced toxicity on their own as compared to WT

DNAJB6. Comparisons were conducted with ordinary one-way ANOVA; ns not significant, *** $P < 0.001$, **** $P < 0.0001$

10. In Figure S13, RNA-seq demonstrates that DNAJB6 is significantly upregulated in response to FUS or TDP-43 overexpression in HEK293T cells. It would be helpful if this could be confirmed by western blot.

We thank the reviewer for the opportunity to clarify this point. We have done western blots on cells that were transfected in the same manner as those collected for RNA-seq as shown in Supplementary Figure 16 (previously Figure S13 in initial submission). We identified a mild upregulation in DNAJB6 upon expression of TDP-43 via western blot but did not identify significant upregulation upon FUS overexpression. This follows a similar trend observed in the RNA-seq data wherein TDP-43 induced a more robust increase in DNAJB6 RNA transcripts. It is possible that at the time point in which these data were captured that the cell has begun upregulating the DNAJB6 transcript but this has not yet led to an increase in protein. It is also possible that the mechanism by which DNAJB6 becomes upregulated upon FUS or TDP-43 overexpression is inadequate with cells upregulating the transcript but failing to convert the increased levels of mRNA to additional amounts of protein due to other post-transcriptional, translational or post-translational processes.

We have updated the text describing the additional data in the supplementary figure:

[LINE 305]

As compared to the EYFP control condition, DNAJB6 was among the most strongly and significantly upregulated chaperones within the ~270 molecular chaperones observed upon overexpression of FUS and TDP-43. However, only overexpression of TDP-43 resulted in an increased amount of DNAJB6 protein at the same time point (Sup. Fig. 16 & 17 and Sup. Table 8).

Supplementary Figures 16 & 17 are pasted below for convenience:

Supplementary Figure 16. RBP overexpression selectively leads to an increase in DNAJB6 levels. **a.** RNA-seq demonstrates that DNAJB6 is significantly upregulated in response to FUS or TDP-43 overexpression in HEK293T cells. Volcano plots for HEK293T expressed chaperones in cells overexpressing FUS or TDP-43 as compared to EYFP. Two biological replicates were done for each condition. **b.** Western blot and Coomassie total protein stain from HEK293T cells transfected with EYFP, FUS or TDP-43. Independent transfections were performed for each sample displayed. **c.** Quantification of DNAJB6 bands normalized to total protein detected by Coomassie staining in **b.** Quantification is shown for the longer DNAJB6a isoform (38 kDa) which contains a nuclear localization signal (NLS) at its C-terminus as compared to the shorter DNAJB6b isoform (25 kDa) that lacks the NLS. Comparisons for **c.** were conducted with ordinary one-way ANOVA; not significant = ns, ** $P < 0.01$, data are shown as mean \pm s.d. for three biological replicates.

Supplementary Figure 17. Full unprocessed blots for measuring DNAJB6 levels after FUS and TDP-43 overexpression. All individual western blots for both DNAJB6 antibody staining (left) and corresponding Coomassie Blue staining for total protein measurement (right), as seen in Sup. Fig. 16. All blot lanes are in the same order as presented in Sup. Fig. 16 with relevant lanes outlined by a red rectangle.

11. The authors showed that DNAJB6 can rescue FUS, TDP-43, and hnRNPA1 toxicity in yeast cells and decrease the insoluble fraction in HEK293 cells. It would be helpful to confirm the toxic reduction activity of DNAJB6 in other model systems. For example, does DNAJB6 rescue toxicity of these RBPs in HEK293 cells?

The reviewer makes a great point and in new data shared at the introduction of these revisions, we highlight the ability of DNAJB6 overexpression to rescue motor neuron loss and the inflammatory response in an ALS-FUS animal model. With regards to the reviewer's exact experimental suggestion, there is a group that has expressed FUS in HEK293T cells and observed cellular toxicity which they then rescue by overexpressing either the eukaryotic translation initiation factor, EIF4A1 (Sun et al.) or the FUS chaperone and nuclear import factor, TNPO1 (Guo et al) (1-2).

Before using this assay to evaluate the effects of DNAJB6 in mammalian cells, we first set out to reproduce the reported FUS-dependent cellular toxicity from this previous report and found a clear effect of FUS overexpression on cellular growth in HEK293T cells (see Figure ii below). Following these studies, we then tested the previously published rescuers to see if we could observe the expected amelioration of FUS toxicity. In contrast to the previous publications, our team was unable to rescue the FUS-dependent toxicity in HEK293T cells when overexpressing either EIF4A1 or TNPO1 (see Figure iii below). Despite these initial negative results, we next performed a large number of follow-up studies where we varied the dosage of the various plasmids, cell outgrowth times, and also tried to temporally express the previously published rescuer before FUS (to give it time to act), but in all cases were unable to identify any rescue (data not shown). The fact that TNPO1 is thought to serve as a molecular chaperone which can modulate FUS aggregation, yet showed no rescue of FUS-dependent toxicity in HEK293T cells, suggested to us that the acute effects of FUS overexpression in HEK293T cells may not be solely dependent upon its aggregation. For this reason, we moved on to other experimental paradigms that enabled us to probe the effect of DNAJB6 and its chaperone activity within mammalian cell context, such as the RBP solubility assays from Chen et al. (3)

Figure ii. *FUS* expression causes toxicity in HEK293T cells depending on the amount of plasmid transfected. This assay uses crystal violet staining to determine the abundance of cells in a well and follows the protocol from Resnick et al. JVI 2021 (4). Increased absorbance is correlated with higher levels of cell density (lower amounts of toxicity) in a given condition. The results of 8 independent transfections are shown for all conditions.

Figure iii. *FUS* toxicity in HEK293T cells is not rescued by literature reported rescuers TNPO1 and EIF4A1. Experiment was performed as in figure ii with 100ng of each plasmid being delivered. The results of 8 independent transfections are shown for all conditions.

Reference:

1. Sun, Z., et al. (2011). "Molecular determinants and genetic modifiers of aggregation and toxicity for the ALS disease protein FUS/TLS." PLoS Biol 9(4): e1000614.
2. Guo, L., et al. (2018). "Nuclear-Import Receptors Reverse Aberrant Phase Transitions of RNA-Binding Proteins with Prion-like Domains." Cell 173(3): 677-692.e620.

3. Chen, H. J., et al. (2016). "The heat shock response plays an important role in TDP-43 clearance: evidence for dysfunction in amyotrophic lateral sclerosis." *Brain* 139(Pt 5): 1417-1432.
4. Resnick, R. et al. (2021). "Inhibitors of Coronavirus 3CL Proteases Protect Cells from Protease-Mediated Cytotoxicity." *Journal of Virology* 95(14): 10.1128/jvi.02374-20

12. Similarly, in the yeast model, DNAJB6 was expressed without human HSP70. I am curious in mammalian cells, does DNAJB6's activity depends on the expression/activity of HSP70?

The reviewer raises an interesting point. To answer this question, we leveraged a mutation in the J-domain of DNAJB6 (H31Q) which ablates transfer of misfolded protein to, and activation of the HSP70 ATPase (1-3). In mammalian cells, we observed that the interaction between DNAJB6 and the HSP70 partner was essential for DNAJB6's ability to reduce the formation of SDS-insoluble FUS, TDP-43 and hnRNPA1. These data have been included in the manuscript as Supplementary Figure 24 & 25 (shown below for convenience), along with the following text:

Reference

1. Aprile, F. A. *et al.* The molecular chaperones DNAJB6 and HSP70 cooperate to suppress α -synuclein aggregation. *Scientific Reports* **7**, (2017).
2. Bengoechea, R. *et al.* Inhibition of DNAJ-hsp70 interaction improves strength in muscular dystrophy. *Journal of Clinical Investigation* 4470–4485 (2020). doi:10.1172/jci136167
3. Tsai, J. & Douglas, M. G. A conserved HPD sequence of the J-domain is necessary for YDJ1 stimulation of HSP70 ATPase activity at a site distinct from substrate binding. *Journal of Biological Chemistry* **271**, 9347–9354 (1996).

[LINE 513]

We also verified that the interaction between DNAJB6 and an HSP70 partner is required for it to reduce the formation of SDS-insoluble species in mammalian cell culture (Sup. Fig. 25 & 26).

Supplementary Figure 25. DNAJB6-HSP70 interaction is required for RBP solubilization in cells. Representative images comparing SDS-soluble (RIPA) and SDS-insoluble/Urea-soluble fractions extracted from cells co-expressing WT DNAJB6 or DNAJB6 H31Q with **a.** FUS **b.** TDP-43 **c.** hnRNPA1. **d.** Quantification of the urea soluble species normalized to total protein detected by Coomassie staining in a-c. * $P \leq 0.05$. Data are shown as the mean \pm s.d. for the three separate transfections and extractions.

Supplementary Figure 26. Full unprocessed blots for dependence of DNAJB6 on HSP70 for rescue in HEK293T cells. All individual replicate blots for FUS, TDP-43, and hnRNPA1 overexpression in HEK293T cells as displayed in Sup. Fig. 25 including both antibody staining (left) and corresponding Coomassie Blue staining for total protein measurement (right). All blot lanes are in the same order as presented in Sup. Fig. 25 with relevant lanes outlined by a red rectangle.

Minor points:
 13. “liquid liquid droplet” has been used in a couple of places in the manuscript and it is not an accurate description. It should be replaced with either “liquid–liquid phase-separated droplets” or “liquid droplets”.

We have corrected the usage of this term from “liquid liquid droplet” to “liquid-liquid phase separated droplet” in all occurrences in the manuscript.

14. Figure 3F is missing the label for Y axis.

We have revised the figure (now Fig 6c in the revised manuscript), and added the Y axis label indicating that it is “secondary structure %” i.e. the proportions of FUS in different chemical structures.

15. In Figure S15, scale bar is missing from A.

We have updated the figure (now Supplementary Fig 21a within the revised manuscript) to include a scale bar, we thank the reviewer for identifying this error.

16. In Figure S15, the droplets in B are very small and also appear to be moving, which prevents a detailed characterization such as circularity measurement or FRAP experiment to confirm the liquid nature of the droplets. I am wondering if an image with clearer droplets would be provided.

We agree that the droplets are small and possibly in motion. We have sought to find sharper images, but have not been able to do so. However, we believe that the current images suffice to demonstrate the formation of DNAJB6 condensates following a drop in salt concentration. Because the focus of the manuscript is on FUS condensates and FUS + DNAJB6 condensates, we did not intend to do FRAP and circularity analyses of the DNAJB6 condensates.

17. For the same reason as mentioned in point 6, the amount of FUS plasmid used for transfection should be mentioned for Figure 4D.

The figure legend has been updated (now figure 7 in revised manuscript) to include the amount of plasmid that was transfected. Updated portion of the figure legend is listed below.

d. Testing of potentiated DNAJB6 variants in mammalian cells for their ability to reduce SDS-insoluble, urea-soluble FUS species upon overexpression with 200 ng of FUS plasmid.

Reviewer #2 (Remarks to the Author):

Authors of the manuscript, “A multiplex platform to identify mechanisms and modulators of proteotoxicity in neurodegeneration,” developed a screening platform to probe the interactions between neurodegenerative disease (ND)-associated proteins and potential toxicity rescuers in yeast cells. From this platform, DNAJB6 was found to modify the solubility and toxicity of proteins implicated in neurodegeneration, such as FUS and TDP-43. DNAJB6 was also demonstrated to co-phase separate with these proteins and prevented their fibrillation. Using a deep mutational scan, specific residues in the glycine/phenylalanine and serine rich regions were identified to reduce or enhance DNAJB6 activity.

Development of an efficient, extensive screening technique that can be used to characterize interactions in a living system is useful for finding potential therapeutic avenues, especially for neurodegenerative diseases. The text thoroughly discusses the successes and shortcomings of their platform; the figures are detailed and visually appealing. Overall, the paper is well-written, and the developed screening platform shows promise in identifying potential rescuers of ND-associated protein toxicity.

However, there are some concerns that the authors should consider:

1. The Results section mentions that DNA-barcoded strains were pooled ‘based on toxicity’. Is there a measure of toxicity or a system of ranking the toxicity of ND-associated proteins in yeast? If so, is the toxicity similar to what would be observed in humans?

We appreciate the opportunity to clarify this point. During the development of our screening paradigm, we varied multiple parameters, one of which was “skewing” the abundance of the various strains within the pool based on their toxicity. The hypothesis was that by increasing the concentration of models that showed high toxicity and decreasing the concentration of models that showed low toxicity we’d be better able to see all of our models in the final outgrown population. To develop our metric of toxicity spot assays for all models were performed and this was used to group each of the models into various categories of toxicity. Strains were then pooled at either a 15:1, 10:1, 5:1, or 1:1 ratio depending on how toxic the model was. However, this “skewed” pool showed worse performance due to increased variation and decreased signal as compared to the conventional pool where all models were mixed at the same initial concentration (**Supplementary Figure 3**). Given these results, we did not pursue this approach further and all future pools were assembled with each model occupying an equal ratio of the pool at the baseline. We have now added 2 sentences (shown below) to **Supplementary Note 2** to describe in further detail how the skewed pool was created:

This was done by first performing a spot assay to determine the relative growth rate of each strain in the pool. Then, strains were pooled at either a 15:1, 10:1, 5:1, or 1:1 ratio depending on the degree of growth suppression, with the most toxic strains given the highest initial abundance.

Additionally, we have updated the last sentence of the “Development of a multiplexed screening strategy to identify rescuers of proteotoxicity” section with the following information to make it clear that we mixed strains at equal ratios:

[LINE 149]

In total, our final screening library contains 32 NDD-associated proteins plus multiple controls, each placed within 5-7 uniquely DNA-barcoded strains and mixed at equal ratios into a single

pool for testing (For detailed descriptions of models and pool composition, see Sup. Table 1 and Sup. Note 3).

Regarding the second question, to our knowledge there is not a study ranking the toxicity of various neurodegenerative disease-associated proteins in human cells. Furthermore, the age of clinical presentation or the average age at which patients succumb to disease cannot be used as a surrogate to quantify the relative toxicity between various proteins given that they are expressed in different neuronal and non-neuronal populations at varying levels which confounds any comparisons. In the absence of a clear set of data to compare our yeast results against it is difficult to confidently make a statement on the relationship between toxicity in yeast vs. humans.

2. There is no evidence that these ND-associated proteins misfold in yeast. The burden of expressing exogenous proteins may be significant enough to cause the observed toxicity. This type of toxicity may be reversed by the presence of protein chaperones, but there is no evidence that rescuing toxicity in yeast is related to rescuing protein misfolding in ND.

The reviewer brings up an interesting discussion point. Many of the neurodegenerative disease-associated proteins we have chosen to study have previously been examined within yeast and shown to have properties consistent with being misfolded upon their overexpression (e.g. forming fluorescent foci, biochemically insoluble species, and showing poor recovery post photobleaching) (1-4). Furthermore, there are several examples where modifiers of the observed toxicity within yeast reverse the phenotypes associated with misfolding (e.g. removing aggregates from cells, increasing protein solubility, and restoring fluorescence recovery after photobleaching) (1-4). For these reasons, we do attribute some of the toxicity we observe to the propensity of these ND-associated proteins to misfold in yeast and the rescue being related to correcting that misfolding.

With regards to DNAJB6 and its relation to rescuing protein misfolding within yeast, the reviewer is correct that we do not provide direct evidence for this occurring in yeast. However, given our evidence for the effects of DNAJB6 on RBP misfolding using purified proteins and within mammalian systems, along with the track record of chaperones rescuing the toxicity of aggregation-prone proteins in yeast via rescuing their misfolding, we propose that DNAJB6 is likely having its effects in yeast by ameliorating the misfolding of FUS, TDP-43 and hnRNPA1(5-7).

References

1. Sun, Z., et al. (2011). "Molecular determinants and genetic modifiers of aggregation and toxicity for the ALS disease protein FUS/TLS." *PLoS Biol* **9**(4): e1000614.
2. Gitler, A. D., et al. (2009). "Alpha-synuclein is part of a diverse and highly conserved interaction network that includes PARK9 and manganese toxicity." *Nat Genet* **41**(3): 308-315.
3. Konopka, C.A. et al. (2011). "A yeast model for polyalanine-expansion aggregation and toxicity." *Mol Biol Cell* **22**(12): 1971-1984.
4. Bolognesi B. et al. (2019). "The mutational landscape of a prion-like domain." *Nat Comm* **10**(1): 4162.
5. Wyszowski, H. *et al.* Class-specific interactions between SIS1 J-domain protein and hsp70 chaperone potentiate disaggregation of misfolded proteins. *Proceedings of the National Academy of Sciences* **118**, (2021).
6. Jackrel, M. E. & Shorter, J. Potentiated HSP104 variants suppress toxicity of diverse neurodegenerative disease-linked proteins. *Disease Models & Mechanisms* **7**, 1175–1184 (2014).

7. March, Z. M. *et al.* Therapeutic genetic variation revealed in diverse Hsp104 homologs. *eLife* **9**, (2020).

Based on the reviewer's comments, we have changed how we introduce the models being examined, explaining that toxicity can arise from multiple sources of which misfolding is one potential mechanism.

From:

"Into each of these barcoded strains, we deliver a construct encoding a NDD-associated protein with a propensity to misfold (e.g., TDP-43, FUS, alpha-synuclein)."

To:

[LINE XXX]

Into each of these barcoded strains, we deliver a construct encoding an NDD-associated protein with a propensity to **cause toxicity either through its misfolding or perturbation of core cellular processes** (e.g., TDP-43, FUS, alpha-synuclein).

3. In the manuscript, 'toxicity' is indirectly measured by the 'growth' rate, and 'rescue' is relative to toxicity. In other words, 'rescue' is assessed by the change in yeast abundance. The data in Fig. 1 b and 1e express 'rescue' as Log2 fold change. It would be useful to include a representative raw data in the supplementary information to depict the general landscape of the effects of ND-associated protein expression have on yeast growth.

The reviewer brings up a great point about sharing primary data. We have included two new Supplementary Tables that provide the raw, unnormalized aligned read counts for each barcoded cell line across all experimental conditions. These tables are Supplementary Table 3 and Supplementary Table 6 within this revised submission.

4. The assumption that toxicity caused by ND-associated proteins is driven by proteins misfolding follows the toxic gain-of-function disease model; however, ND toxicity may also be caused by a loss-of-function or other disease pathways. Would the results from this screening platform be able to address other forms of ND pathology?

We thank the reviewer for providing us an opportunity to address this point. While we have focused our follow-up studies on genetic modifiers such as DNAJB6 that we argue prevent toxicity by restoring protein folding, our platform is able to address other causes of toxicity. In fact, we note that our platform is non-biased in that we do not require the proteins to misfold in yeast for us to include them – we only require them to cause toxicity to the yeast cell. It is likely that for some of our models misfolding is not the only mechanism by which they cause toxicity. Yet, a benefit of our system is that even in these cases we can still use our approach to identify genetic modifiers that may function by regulating the toxic biochemical activity of the neurodegenerative disease associated protein and in doing so still provide valuable insight into the biology of these clinically relevant targets.

Furthermore, we have chosen to focus primarily on toxic gain-of-function phenotypes as these are relatively easy to model in wild type yeast and have a wealth of literature supporting their use. Although not explored in this work, we believe our system is amenable to studying loss-of-function (LOF) mutations in cases where yeast have a homolog of a protein that is deficient in human patients and loss of the yeast homolog causes a growth defect. With these properties it would

then be possible to substitute the loss-of-function variants for the yeast homolog and screen for genetic modifiers that restore function. Examples of LOF mutations modeled in yeast are shared below:

1. Wilson, R. B. and D. M. Roof (1997). "Respiratory deficiency due to loss of mitochondrial DNA in yeast lacking the frataxin homologue." *Nat Genet* 16(4): 352-357.
2. Babcock, M., et al. (1997). "Regulation of mitochondrial iron accumulation by Yfh1p, a putative homolog of frataxin." *Science* 276(5319): 1709-1712.
3. Nishida, C. R., et al. (1994). "Characterization of three yeast copper-zinc superoxide dismutase mutants analogous to those coded for in familial amyotrophic lateral sclerosis." *Proc Natl Acad Sci U S A* 91(21): 9906-9910.

5. The yeast chaperones were demonstrated to be more effective at rescuing proteotoxicity. Does this suggest that the toxicity observed is yeast-specific? If these screens/experiments were conducted in other cell lines, would there be a significant difference in the results? How translatable are these results to human cells?

We appreciate this observation and chance for further discussion. It does appear at first glance that the yeast chaperones produce more frequent and stronger interactions. We do want to point out that this seems to be the result of the engineered HSP104 variants being included in the yeast set. While these are derived from a yeast chaperone, they have been engineered in the laboratory to have broader activity and do not represent a canonical yeast chaperone (1). They were included in our study as positive controls. Removing them from the analysis shows that among called hits, human chaperones have an average L2FC of 1.00 (n=20) while yeast chaperones have an average L2FC of 1.21 (n=20) but the means are not statistically different (p=0.34). Similarly, among suspected interactions that were validated, human chaperones had an average L2FC of 0.42 (n=22) while yeast chaperones had an average L2FC of 0.39 (n=11) without a statistically significant difference in mean (p=0.70). We believe that this suggests that in our study, both yeast and human chaperones had a similar degree of rescue activity suggesting that chaperones from both organisms can work in our assay. It is true that a greater percentage of total validated interactions from yeast chaperone set were called statistically significant hits within our screen as compared to human chaperones (65% vs. 48%) and this may represent an advantage the yeast chaperones had by functioning within their native environment. We do believe that there might be some changes in the activity of certain chaperones if our studies were done in other cell lines given that chaperones often function in multiple member complexes, but it is difficult to predict what this would look like *a priori*. Chaperones are a relatively conserved group of proteins which may enable them to function across different organisms, which may explain why we observe both human and yeast chaperones working to resolve proteotoxicity within yeast cells. Ultimately, we validate interactions that were identified in our yeast system in mammalian cells, suggesting that the yeast results are translatable to human cells.

Reference:

1. Jackrel, M. E., et al. (2014). "Potentiated Hsp104 variants antagonize diverse proteotoxic misfolding events." *Cell* 156(1-2): 170-182.

6. It is unclear why the data in Fig. 1d and 1g were selected from the statistically significant interactions in Fig. 1c and 1f. Why were the most significant interactions from the screening platform not chosen for validation?

The reviewer brings up a good point about the data we have chosen to highlight in Fig. 1d and 1g. We should first point out that we did in fact test all significant interactions in the screen. In **Supplementary Table 4**, the one-by-one validation results from every significant interaction in the screen that was manually tested is shown. We chose to highlight a few interactions in Fig. 1d and 1g to demonstrate that the platform identified previously unreported interactions from multiple different models in both the yeast and human chaperone screens.

7. It may be useful to show negative controls when validating the screening results (i.e., the relative growth of statistically non-significant samples from Fig. 1c and 1f)

We agree that it is important to share negative control data. We have displayed the 77 negative control interactions representing interactions that were not significant for an increase in growth rate. These data were initially included in Supplementary Table 3 of our initial submission but are now also shown as graphs in **Supplementary Figure 11** (shown below for convenience). Of the 77 negative interactions tested none caused an increase in growth rate when examined one at a time.

Supplementary Figure 11. Validation of interactions not predicted to cause growth rescue. Eleven ORFs that did not result in growth rescue in the multiplexed screen were individually tested for their ability to increase the growth rate for **a. FUS b. TDP-43 c. EWSR1 d. hnRNP1 e. PR50 f. Kar2-Abeta and g. Alpha-synuclein** expressing yeast. Data are shown as mean \pm s.d. for three biological replicates for chaperones and six biological replicates for the mCherry condition.

dc

8. Fig. 1e shows that the highest fold change was observed with the interaction between DNAJB6 and TDP-43, followed by FUS and then hnRNP1. However, Fig. 2a shows the reverse: DNAJB6

rescues hnRNPA1 the most, followed by FUS and then TDP-43. Furthermore, only one of these interactions (TDP-43) were deemed as statistically significant in Fig. 1f. Why was DNAJB6 selected amongst other chaperones when most of its interactions were not statistically significant (Fig. 1f)? What is the difference between 'suspected interactions' and false negatives? How applicable is this screening platform if the sensitivity is reported to be only 66%?

The reviewer brings up multiple good points within this comment that we will address below.

The discrepancy between the fold changes in the multiplexed screen (Fig. 1e) and the individual validation (Fig. 2a) is largely driven by the difference in experimental paradigms. In the multiplexed screen, a small inoculum of each strain within a complex and dynamic mixed pool is grown, mated to a potential genetic modifier, selected, and then outgrown in inducing conditions. Given the multitude of steps, the sparse initial inoculum, and the moderate amount of sequencing depth, there is variability inherent to the platform. We can overcome this variation through the usage of multiple uniquely barcoded strains assigned to each model (**Supplementary Figure 6**). When analyzing the results of our screening data the average log₂ fold change (L2FC), while valuable, does not capture the variance of the 5-7 individual strains representing a given model. This likely explains why the results of L2FC do not match up with the individual validation, although we do note that the L2FC suggests that DNAJB6 should rescue the growth of TDP-43, FUS, and hnRNPA1 and this is what we observe upon individual testing.

With regards to why only the interaction between DNAJB6 and TDP-43 was deemed statistically significant, this is due to our use of a very strict correction factor for multiple hypothesis testing. If we were to use a less strict form of correction it would lead to more hits passing our statistical threshold, but this may also come at the cost of an increase in false positives. As such we have chosen to select a very demanding threshold for our initial analysis such that the rate of false positives is very low. Realizing that what we call as statistically significant does not encompass all potential hits, we also provide the L2FC data (Sup. Table 2 and Sup. Table 5) so if others wish to explore weaker interactions they may also do so, as we have shown that *bona fide* interactions often fall outside of our statistical threshold.

As to why DNAJB6 was selected as noted above while it only showed significance against TDP-43 it showed clear L2FC against other RBPs implicated in the same disease process. We thought this was an interesting finding as few human proteins have been identified that influence the behavior of multiple RBPs implicated in ALS and FTD. Furthermore, we tested many other members of the DNAJ family in our screen, yet DNAJB6 was unique in its breadth and strength of rescue, for these reasons we chose to focus on it.

For purposes of determining our assay sensitivity we define our false negatives as the 59 interactions with positive L2FC in our screen (i.e. suspected interactions) that showed rescue when tested one at a time but failed to achieve statistical significance in our screen. We justify this choice since when we examined interactions without a positive L2FC we found that none showed rescue, thus to simplify the manual validation of the 5,850 possible interactions we focused on individually testing only those with positive L2FC in the screen. As noted above, we have selected a very strict threshold for statistical significance and with any screening assay there is an inherent trade-off between sensitivity and specificity, which for our assay we estimate at 66% and 99%, respectively. In other words, we could lower our statistical threshold and this would increase our sensitivity but at a cost of decreasing our specificity. We have favored a very high level of stringency such that we can feel confident that hits which pass our criteria are likely to validate upon individual testing. We have chosen this criterion as we are of the notion that the utility of a screen is to rapidly survey a large interaction space and narrow down the area for

subsequent validation testing to a number that is not overly burdensome to test manually. Should subsequent groups desire to change the threshold for calling hits we have now included raw barcoded count data from our screens in two new Supplementary Tables, labeled **Supplementary Table 3** and **Supplementary Table 6**.

As for the utility of our assay given its estimated sensitivity of 66%, this can be changed should users decide they are willing to tolerate an increase in false positives. For example, our pipeline can detect 157/175(89.7%) of validated interactions by dropping the stringency of our multiple hypothesis testing but this would lead to a total of 333 called hits, suggesting that the specificity would be closer to 53%. While many of the initial yeast neurodegenerative disease screening papers do not report validation statistics (1-5), when looking at the positive predictive value of our assay as currently implemented, it performs well in comparison to other yeast screening methods, 96.7% versus 75% (6), 29% (7), 43% (8), 50% (9), 59% (10), and 80% (11), respectively. Finally, even with our stringent criteria for hit calling, we still have several dozen interactions to be explored at a molecular level, many more than any individual lab can pursue and so we consider the screen a success, as it has served its purpose of being a strong dataset for hypothesis generation. We hope that upon publication of our full dataset, the larger scientific community will explore our results and if desired tweak our analysis pipeline to their interests.

1. Sun, Z., et al. (2011). "Molecular determinants and genetic modifiers of aggregation and toxicity for the ALS disease protein FUS/TLS." *PLoS Biol* 9(4): e1000614.
2. Cooper, A. A., et al. (2006). "Alpha-synuclein blocks ER-Golgi traffic and Rab1 rescues neuron loss in Parkinson's models." *Science* 313(5785): 324-328.
3. Ju, S., et al. (2011). "A yeast model of FUS/TLS-dependent cytotoxicity." *PLoS Biol* 9(4): e1001052.
4. Armakola, M., et al. (2012). "Inhibition of RNA lariat debranching enzyme suppresses TDP-43 toxicity in ALS disease models." *Nat Genet* 44(12): 1302-1309.
5. Kim, H. J., et al. (2014). "Therapeutic modulation of eIF2 α phosphorylation rescues TDP-43 toxicity in amyotrophic lateral sclerosis disease models." *Nat Genet* 46(2): 152-160.
6. Jo, M., et al. (2017). "Yeast genetic interaction screen of human genes associated with amyotrophic lateral sclerosis: identification of MAP2K5 kinase as a potential drug target." *Genome Res* 27(9): 1487-1500.
7. Zhang, L., Nebane, N. M., Wennerberg, K., Li, Y., Neubauer, V., Hobrath, J. V., McKellip, S., Rasmussen, L., Shindo, N., Sosa, M., Maddry, J. A., Ananthan, S., Piazza, G. A., White, E. L., & Harsay, E. (2010). A high-throughput screen for chemical inhibitors of exocytic transport in yeast. *Chembiochem : a European journal of chemical biology*, 11(9), 1291–1301.
8. Sarnoski, E. A., Liu, P., & Acar, M. (2017). A High-Throughput Screen for Yeast Replicative Lifespan Identifies Lifespan-Extending Compounds. *Cell reports*, 21(9), 2639–2646.
9. Wong LH, Unciti-Broceta A, Spitzer M, White R, Tyers M, Harrington L. A yeast chemical genetic screen identifies inhibitors of human telomerase. *Chem Biol*. 2013 Mar 21;20(3):333-40.
10. Park, S. K., Pegan, S. D., Mesecar, A. D., Jungbauer, L. M., LaDu, M. J., & Liebman, S. W. (2011). Development and validation of a yeast high-throughput screen for inhibitors of A β ₄₂ oligomerization. *Disease models & mechanisms*, 4(6), 822–831.
11. Fernández-Acero, T., Rodríguez-Escudero, I., Vicente, F., Monteiro, M. C., Tormo, J. R., Cantizani, J., Molina, M., & Cid, V. J. (2012). A yeast-based in vivo bioassay to screen for class I phosphatidylinositol 3-kinase specific inhibitors. *Journal of biomolecular screening*, 17(8), 1018–1029.

9. There is no significant difference in the AFM-IR data of FUS versus FUS+DNAJB6. Other techniques (e.g., electron microscopy, far UV CD spectroscopy, ThT fluorescence-based assays) would more accurately show DNAJB6 effect, if any, on FUS aggregation.

We thank the reviewer for providing the opportunity to include additional approaches to demonstrate the modulatory role of DNAJB6 on FUS aggregation, namely fluorescence-based assays, which we discuss below.

We also thank the reviewer for pointing out that the AFM-IR data in our original figure (now Fig 6b) are potentially open to the misinterpretation because the error bars are very small, and the lines for FUS and FUS+DNAJB6 are very close together. This is standard for FTIR spectra, in particular the amide I band used for secondary structure determination, as it contains the contributions of various secondary structure elements in a small spectral window, thus precluding visual inspection. Thus, it is convention to perform derivation of spectra to emphasize individual secondary structure elements, quantified in the histograms in Fig. 6c, which reveal subtle but significant differences. We have revised the AFM-IR figure to more clearly show this.

Our infrared nanospectroscopy (AFM-IR) studies exploited the ability of AFM-IR to combine the imaging power of atomic force microscopy, an imaging technique with comparable/higher spatial resolution to electron microscopy, with the molecular/structural analysis power of infrared spectroscopy at this single condensate level. This ability to interrogate individual condensates permits us to unravel the heterogeneity in secondary structure of single condensates of FUS versus FUS+DNAJB6 mixtures.

As we show now in the revised Fig 6b-c and Supplementary Fig 23 a-g, the single-molecule level characterisation of AFM-IR reveal a small but statistically significant difference between the secondary structure of condensates of FUS versus FUS+DNAJB6. Specifically, the revised graph shows that: i) the FUS+DNAJB6 condensates have a higher content of intermolecular β -sheet, indicating the presence of a gelled phase; ii) this signature occurs at lower wavenumbers, indicating the presence of more extended strands in the gelled phase.

By characterising the condensation at the single condensate level, AFM-IR allowed us to complement traditional bulk approaches (e.g. CD spectroscopy, ThT). These traditional approaches retrieve average information of the heterogeneous mixtures of condensates and possible fibrillar aggregates (as we demonstrated in Shen, Nature Nanotechnology, 2020). Our AFM-IR approach also circumvents the limitations of conventional imaging techniques such as electron microscopy, which can visualize the condensates but cannot characterise their gelation and secondary structure.

We have also included pFTAA binding studies in Fig 5a and Supplementary Fig 22a. We have previously shown in biochemical and cellular experiments that pFTAA fluorescence reflects increasing content of proteins with repeated crossed- β sheet structures (Qamar et al. Cell 2018). Classical amyloidophilic dyes such as Thioflavin T (ThT) do not bind well even to pathological FUS aggregates. In contrast, the oligothiophene dye pentameric formyl thiophene acetic acid (pFTAA) does bind to pathologically condensed gels and fibrillar aggregates of FUS and tau, and does so in both biochemical and cellular preparations. This property relates to the high affinity of pFTAA for repeated crossed- β sheet structures in fibrillary assemblies. These studies reveal that FUS-only condensates form large, non-spherical shapes with strong pFTAA fluorescence in a time-dependent (“ageing”) manner. Many of these aged FUS-only condensates display strongly pFTAA-labelled foci dispersed within the aggregates. In contrast, over the same time interval,

FUS + DNAJB6 condensates remain smaller with only weak pFTAA signals (Fig. 5a, Supplementary Fig 22a; figures shown below for convenience).

Taken together, these studies show that DNAJB6 has a profound effect on the dynamics of FUS condensation and aggregation, and in doing so reduces the tendency of FUS to undergo progressive conversion into β -sheet rich aggregates.

Fig 6a-c shown below for convenience (previously Fig. 3e-i). AFM-IR nano-chemical analysis of FUS and FUS+DNAJB6 empowers the analysis of the a) 3-D morphology and b) nanoscale chemical properties in the form of localized IR spectra (colored crosses in the AFM maps) and the deconvolution by second derivative analysis of the Amide band I of protein allows to c) measure the secondary structure and aggregation state of the condensates.

Figure 5. pFTAA staining, AFM, and FTIR studies of the effect of DNAJB6 on FUS condensates. **a.** Aged FUS condensates have higher content of pFTAA-positive condensates than FUS with DNAJB6. pFTAA staining of mCherry-FUS and mCherry-FUS with DNAJB6 after 48 hrs, with corresponding zoom-in regions below. Scale bars correspond to 25 μm for top panels and 10 μm for bottom zoomed-in panels.

Supplementary Figure 22: pFTAA staining quantification and AFM quantification and FTIR spectra of DNAJB6. **a.** Quantification of pFTAA signal from condensates at 48 hr (each dot corresponds to the average of the signal of the condensates in a field of view, 807 condensates for mCherry-FUS and 8,976 condensates for mCherry-FUS with DNAJB6). Mann-Whitney non-parametric test p-value < 0.01. The FTIR spectra **b.** and second derivate **c.** were acquired for DNAJB6 alone to ensure the secondary structure changes detected for FUS + DNAJB6 condensates was not solely due to the presence of the additional protein component. Spectra were measured at 3 μm (~9x higher than in FUS + DNAJB6 measurements), as there was insufficient signal at lower concentrations.

10. Microscopy images of irregularly shaped condensates in Fig. 3c do not necessarily prove that fibrils formed. Condensate formation and LLPS do not always result in fibrillation. Other experiments must be performed to confirm whether fibril formation has occurred.

To answer this question, we confirmed that our methods (fluorescence imaging and AFM-IR) are able to discern and characterise the formation of fibrils. We present below fluorescence images and AFM-IR data, where we analyse the secondary structure of FUS fibrillar condensates like those in the fluorescent images in Fig. 4a - 36 hour panel, and in the left panel in the image below. These fibrillar condensates have a large spectral signature and content of intermolecular β -sheet structure indicating their transition to the amyloid-like fibrillar state. This is not a feature of the gelled FUS + DNAJB6 condensates in Fig 6a-c.

11. There is evidence that FUS + DNAJB6 may form loose, gel-like structures, but fibrils can still be present in gel-like structures. The AFM-IR data showed an increase in intermolecular parallel and anti-parallel β -sheets in the FUS + DNAJB6 (Fig. 3f). Fibrils contain β -sheet structures that can be parallel or anti-parallel; therefore, more experiments need to be conducted to show whether fibrils are present in the FUS and FUS+DNAJB6 samples.

As described in the point 10 above, our methods are able to discern fibrillar formation. Some fibrillar aggregates are indeed rarely present also in the FUS+ DNAJB6 at later time points. In the new Supplementary Figure 23 (panels h-j), we present a comparison of the gelled condensates and the “rarer” fibrillar aggregates of FUS+ DNAJB6 at 48h. As expected, the fibrillar state has an increased content of intermolecular β -sheet compared to the condensates, indicating a transition towards the amyloid-like state.

In the present paper we decided to focus on the ability of DNAJB6 to modulate the structural properties of the more abundant gelled condensates (which is the novelty of this paper). To accomplish this, we exploited the ability of AFM-IR to investigate the properties of the sample at the single aggregate and condensate level.

Supplementary Figure 23. AFM-IR nano-chemical analysis on single FUS and FUS+DNAJB6 condensates. Example for a single FUS+DNAJB6 condensate map showing **a.** 3-D morphology, **b.** IR absorption in the Amide I (1655cm⁻¹), **c.** Contact resonance by phase locked loop (PLL) which offers a quality control on the chemical signal provided by AFM-IR. **d.** 3D morphology maps of 2 independent FUS and 2 independent FUS + DNAJB6 condensates. **The fine black lines are defects in the ZnSe surface.** **e.** IR spectra from 10 independent locations (each location 5 co-averaged spectra) on the condensate, **f.** their average + SE and **g.** second derivative of the amide I band to deconvolve protein secondary structure contributions. **h.** Maps of 3D morphology and **i.** second derivative of the amide I band to **j.** deconvolve fibrillar secondary structure vs gel-like condensates.

12. Fig 3a suggests that DNAJB6 affects LLPS characteristics of FUS (i.e., FUS+DNAJB6 LLPS is slower and results in smaller droplets), but there is no data that suggests that aggregation is affected. Kinetics data is required to support the hypothesis that DNAJB6 prevents fibrillation.

To address this question we provide kinetic data from two types of experiments – FRAP and FLIM analyses. We show that the fluorescence recovery rates of FUS + DNAJB6 condensates are initially significantly faster than FUS-only condensates (Sup. Fig. 21d). These FRAP studies therefore demonstrate a clear impact of DNAJB6 on the dynamics of the early stages of FUS condensation during the first 20 minutes. However, FRAP has limited ability to see dynamic differences in the FUS mobility of gelled versus fibrillary aggregates that increasingly occur after

the first 20 minutes. To circumvent this, we then applied FLIM methods to investigate the biophysics at later stages of condensate “aging” (Fig 4c, Supplementary Fig 21e-g). These studies show that in the FUS-only preparations there is a continued progressive and time-dependent reduction in fluorescence lifetimes (Fig. 4c; Supplementary Fig 21e-g). This indicates that, over time, FUS-only polymers interactions become quenched— a result that is fully congruent with the microscopic observation that FUS-only preparations eventually form fibrillar aggregates (Fig 4a). In contrast, at all time points from 20 minutes to 48 hours, the FUS + DNAJB6 condensates demonstrate significantly longer fluorescence lifetimes that are similar to those of control experiments containing dispersed solutions of DNAJB6 and mEmerald which do not form condensates (Fig. 4c, Sup. Fig. 21e-g). Accordingly, these FLIM experiments suggest that DNAJB6 permits FUS condensation to occur, but limits close homotypic FUS-FUS interactions, and prevents further (“packing”) into dense viscous fibrillar condensates.

To discover the biophysical and mechanical basis of these changes, we next complement the FRAP and FLIM studies with dynamic pFTAA binding studies (Fig. 5a and Supp. Fig. 22a), and by time-resolved AFM nanomechanical studies (Fig 5b) and nano-IR spectroscopy structural studies (Fig 6).

The pFTAA studies (Fig. 5a and Supp. Fig. 22a) show that over 48 hours, the FUS-only condensates increasingly form large, non-spherical shapes with strong punctate pFTAA fluorescence. In contrast, over the same time interval, FUS + DNAJB6 condensates remain smaller with a significantly weaker pFTAA signals, indicating the absence of progressive conversion to β -sheet.

The AFM nanomechanical studies of FUS-only condensates (Figure 5b) display a time-dependent increase in the Young’s modulus (which reports on the emergence of solid-like behavior within ageing condensates). At 2 hours, the Young’s modulus of FUS-only condensates is ~ 9 kPa, which is typical of aged fluids. However, at 48 hours, the Young’s modulus of FUS-only condensates increases to $\sim 25,000$ kPa, which is consistent with the emergence of solid-like behaviour. In contrast, FUS + DNAJB6 condensates initially display a higher elastic response of ~ 31 kPa at 2 hours, which then remains essentially unchanged up to 48 hours (~ 40 kPa) (Fig. 5b). Thus, on the timescale explored, partitioning of DNAJB6 into the FUS condensates prevents ageing into solid-like assemblies – a conclusion that is also in good agreement with the FRAP, FLIM and tie-line experiments in Fig 4 and Sup. Fig. 21.

To understand whether DNAJB6 dynamically alters the ageing behavior of FUS condensates by inducing changes in the secondary structure and conformation of the component polypeptide chains, we turned to bulk (FTIR, Fig 5c) and single molecule infrared nanospectroscopy (AFM-IR, Fig 6). Bulk Fourier transform IR spectroscopy (FTIR) of FUS-only condensates reveals that the dominant peak at 2 hours is from random coil/ α -helical conformations (1657 cm^{-1})²¹. After 24 hours of ageing, the intensity of this peak decreases, and is accompanied by the appearance of a peak corresponding to intermolecular β -sheet (Fig. 5c). This peak further increases in intensity at 48 hours (Fig. 5c). These results suggest a gradual transition from a random coil to a solid intermolecular β -sheet conformation.

The same experiment on FUS + DNAJB6 condensates reveals two dominant peaks at 2 hours. One reflects random coil/ α -helix (1655 cm^{-1}) (Fig. 5c). The other reflects intermolecular, parallel β -sheet (1626 cm^{-1})^{15,21}, which did not further increase in intensity up to 48 hours (Fig. 5c). These FTIR results are consistent with DNAJB6 stabilizing the presence of gel-like condensates that cannot further develop to solid-like states. Indeed, in the presence of DNAJB6, most of the FUS polypeptide chains remain in a random coil conformation as indicated by the lack of change in the

peak intensity at $\sim 1655\text{ cm}^{-1}$. As a control, we confirmed that the secondary structure changes seen in the FUS + DNAJB6 condition were not due to DNAJB6 alone (Supp. Fig. 22b-c). These results are further supported by the single condensate AFM-IR studies described above and shown in Fig 6.

Taken together these results suggest that DNAJB6 dynamically alters the chemical environment and the interaction network between disordered FUS polypeptide chains, and thereby prevent assembly into fibrillar aggregates.

13. (SI Fig. 15) DNAJB6 does not phase separate in high salt concentrations. FUS is an RNA-binding protein, and nucleic acids can be conceptualized as long polyanions. How will the presence of RNA affect the phase separation of DNAJB6 alone versus DNAJB6 + FUS?

We thank the reviewer for pointing out this important concept. FUS and RNA are well known to influence each others phase separation in a concentration dependent fashion. It is very likely that RNA and other polymers in ribo-nucleoprotein (RNP) granules will influence FUS – DNAJB6 interactions. It's hard to predict how such three-way interactions will alter FUS aggregation/gelation dynamics. This is clearly a new direction for future work that we highlight in the discussion of the revised paper.

[LINE 674]

There is a growing body of work describing critical functional interactions between components of biomolecular condensates (including intrinsically disordered proteins (IDPs) and their interacting RNAs) and molecular chaperones (especially homomeric/heteromeric assemblies of heat shock proteins - HSPs)^{98–105}. Investigating potential interactions between these components (IDPs, RNAs, and chaperones) will be of interest to pursue in future studies. Yet, even in the absence of knowledge about whether or not RNAs play a role in FUS-DNAJB6 condensate formation, there is evidence that IDP-chaperone interactions play a key role both in forming biomolecular condensates and in preventing rampant aggregation of their aggregation-prone components^{98–105}

14. The manuscript mentions that FUS and DNAJB6 “directly interact” (line 319) but there is no experiment that proves that these two proteins bind. Co-localization demonstrates the two proteins temporally occupy the same vicinity; therefore, additional experiment is needed to provide binding and direct interaction between FUS and DNAJB6.

To address this question, we provide three types of data – Co-IP data from cells exogenously expressing FUS and DNAJB6 (Sup. Fig. 20), co-localization in endogenously expressing human iPSC derived motor neurons (Fig 3), and *in vitro* biophysical “tie-line” experiments (Fig 4d). In the next paragraphs, we discuss these experiments, but taken together they suggest that DNAJB6 forms direct but transient, low avidity direct interactions with FUS. These interactions prevent close packing and thus disrupt the subsequent inter- and intra-molecular homotypic FUS-FUS interactions that drive aging of condensates from liquid droplet to aggregated states.

We performed a co-immunoprecipitation (Co-IP) assay between 3xFLAG-DNAJB6 and each of our target RBPs. In mammalian cells overexpressing 3xFLAG-DNAJB6 and each RBP, a clear interaction between DNAJB6 and both TDP-43 and hnRNPA1 was observed (Sup. Fig. 20). Our Co-IP was unable to demonstrate an interaction between DNAJB6 and FUS. We explain this difference as reflecting the nature of the FUS-DNAJB6 interaction – namely that it may be weak, and so does not withstand our Co-IP conditions.

We also undertook high resolution co-localization experiments on the endogenous FUS and DNAJB6 proteins in disease-relevant wild type human iPSC-derived motor neurons. This work shows that a small percentage (~4%) of intracellular puncta contain both FUS and DNAJB6 (Fig 3).

Finally, we undertook additional biophysical experiments that exploit a novel tie-line methodology. This approach characterizes the collective interactions between FUS and DNAJB6 in the condensed phase, and does so by examining changes in the concentration of FUS in the dilute phase after phase separation. These experiments indicate that DNAJB6 prevents the overall liquid-to-solid transition of FUS by replacing on-pathway homotypic FUS-FUS interactions with preferential direct heterotypic FUS-DNAJB6 interactions (Fig. 4d). Hence, FUS molecules interact with DNJAB6 rather than themselves, which preserves their liquid / gel-like state, rather than progressing to more solid states as shown by the AFM nanomechanical studies in Fig 5b.

We interpret these results to mean that DNAJB6 directly interacts with several different RBPs, and that this interaction underpins the ability of DNAJB to halt aging of the RBP condensates into fibrillar aggregates. The interaction between FUS and DNAJB6 is likely to be weaker and/or more transient than for the other RBPs (hence our inability to Co-IP DNAJB6 and FUS). Additional experiments will be needed to understand the molecular basis for this difference. For instance, there may be differences in the number and/or biophysical properties of the interacting residues between different RBPs and different chaperone proteins. However, this work is beyond the scope of this paper, which is already very data-rich. We hope that the reviewer will agree that this might be best pursued in a separate manuscript.

For convenience the referenced figures are pasted below.

Supplementary Figure 20. Co-immunoprecipitation (Co-IP) assays probing interactions between DNAJB6 and various RBPs. **a.** Protein homogenates from HEK293T cells transfected with plasmids overexpressing untagged DNAJB6 (WT) or 3xFLAG-DNAJB6 and either FUS, TDP-43 or hnRNPA1 were used as the input for a Co-IP. **b.** Uncropped western blots for data shown in panel a.

Figure 3. FUS and DNAJB6 coincide in a subset of neuritic motor neuron granules. **a-d.** Immunostain of **a.** FUS, **b.** DNAJB6, **c.** FUS and DNAJB6 merge, in hiPSC human differentiated motor neurons. **Left panel:** A single Airyscan Z-stack section is shown, out of three 150 nm sections with continuous overlap between FUS and DNAJB6 (scale bar = 2 μ m). **Right panel:** magnification of a FUS granule that co-labels with DNAJB6 (scale bar = 0.176 μ m).

Figure 4. Biophysical characterization of FUS condensates with and without DNAJB6. **a.** DNAJB6 promotes formation of smaller spherical condensates that are stable during aging, and prevent formation of dense irregular spiculated / fibrillary condensates. FUS and FUS + DNAJB6 condensates at 20 minutes, 24 hours and 36-48 hours. FUS condensates form fibrillar aggregates when incubated for 36 hours. Scale bar = 10 μ m at 20 min and 24 hours, scale bar = 20 μ m at 36 hours. Data were obtained using epifluorescence microscopy. **b.** Quantification of condensate size. One micron squared minimum size cutoff was used when quantifying condensates. Average condensate size was determined for multiple condensates within several different fields of view. At 20 minutes, n=343 for FUS and n=1,455 for FUS + DNAJB6. At 24 hours, n=792 for FUS and n=1,459 for FUS + DNAJB6. Comparisons were conducted with Welch's two-sided t test. ****P \leq 0.0001 Plots are shown as mean \pm s.e.m. **c.** Fluorescence lifetime measurements (FLIM) of mEmerald-FUS condensates \pm DNAJB6 over time. The condensate-averaged fluorescence lifetime of mEmerald fluorophores linked to FUS significantly decreases over time, indicating FUS condensation and aggregation over time. However, in the presence of DNAJB6, FUS aggregation is prevented, as shown by the significantly higher fluorescence lifetime detected with DNAJB6. Non-parametric Kruskal-Wallis test with Dunn's multiple comparison where * is P<0.05 and **** is P<0.0001. Error bars denote \pm s.d. **d.** Tie line gradient analysis reveals preferential partitioning and interaction between DNAJB6 and FUS. (*left panel*) Intensity time trace of a phase separated FUS sample upon flowing through a confocal detection set-up using microfluidic technology (top) and corresponding intensity histogram (bottom). Bursts of high intensity correspond to condensates passing through the confocal volume. The baseline signal forms a peak in the intensity histogram and is used to extract dilute phase concentrations. For details, see Materials and Methods. (*middle panel*) Dilute phase concentration changes of FUS with varying DNAJB6 concentrations allowing for determination of the tie line gradient value between FUS and DNAJB6 at the measured conditions. (*right panel*) DNAJB6 is recruited into FUS condensates, effectively spacing out aggregation promoting FUS-FUS interactions to induce liquid state preservation.

REVIEWER #3 (REMARKS TO THE AUTHOR):

I looked only the AFM-IR experiments and they were carried out carefully. I trust the results. The data collected support the conclusions.

1. One could argue that AFM-IR probes the surface, and the protein aggregates at the surface might be different than in the bulk of the aggregate. So one point that the authors should mention is the fact that AFM-IR probes the sample down to a limited depth, and therefore I would like that you mention this depth and compare it to the size of the observed aggregates.

We agree with the reviewer that this is an important consideration for many experimental set-ups. However, in our configuration, photothermal AFM-IR is not strictly a surface sensitive technique. Indeed, for protein condensates smaller than 1 μm in height, AFM-IR probes both their surface and bulk properties because the typical penetration length of IR light in a material is in the order of several micrometres (Lipiec, Ruggeri, *Nucleic Acids Research*, 2019).

2. A remark at the lines 275-..., Intriguingly.... The fact that you observe more aggregates in one case does not mean that there are more in solution, since what you observe are the aggregates that are adsorbed on the surface and not those which are in solution: this is especially the case after 20 minutes where the aggregates in the two experiments are almost the same size, in this case one should have the same number. There might be an effect of difference in adsorption on the surface for FUS and FUS-DNAJB6 aggregates.

We thank this reviewer for pointing this out. The images were taken in the fluid phase of solvent, not on the surface of the slide. We propose that in the FUS + DNAJB6 sample, DNAJB6 interferes with the homotypic FUS: FUS interactions, and this has two consequences. First, it blocks the ability of FUS molecules in the condensate to recruit more FUS molecules from the solvent/dilute phase, and so the small droplets fail to grow. Second, DNAJB6 is likely to inhibit droplet fusion (i.e. Ostwald ripening). In contrast, we suggest that the larger size and smaller number of FUS-only condensates at 20 minutes reflects ongoing recruitment of FUS molecules from the dilute phase plus the effects of droplet: droplet fusion.

3. For Figure 3a,b and c, the authors should mention with which technique the different sets of data were obtained.

These data are now shown in Fig. 4a-b in the updated manuscript. We have revised the figure legend to clarify that all three figure elements depict the result of visualisation using epifluorescence microscopy of FUS condensates in the presence/absence of DNAJB6 at the times indicated. The figure and its updated legend are pasted below for convenience.

c Condensate-Averaged Fluorescence Lifetimes

Figure 4. Biophysical characterization of FUS condensates with and without DNAJB6. **a.** DNAJB6 promotes formation of smaller spherical condensates that are stable during aging, and prevent formation of dense irregular spiculated / fibrillary condensates. FUS and FUS + DNAJB6 condensates at 20 minutes, 24 hours and 36-48 hours. FUS condensates form fibrillar aggregates when incubated for 36 hours. Scale bar = 10 μm at 20 min and 24 hours, scale bar = 20 μm at 36 hours. Data were obtained using epifluorescence microscopy. **b.** Quantification of condensate size. One micron squared minimum size cutoff was used when quantifying condensates. Average condensate size was determined for multiple condensates within several different fields of view. At 20 minutes, $n=343$ for FUS and $n=1,455$ for FUS + DNAJB6. At 24 hours, $n=792$ for FUS and $n=1,459$ for FUS + DNAJB6. Comparisons were conducted with Welch's two-sided t test. **** $P \leq 0.0001$. Plots are shown as mean \pm s.e.m. **c.** Fluorescence lifetime measurements (FLIM) of mEmerald-FUS condensates \pm DNAJB6 over time. The condensate-averaged fluorescence lifetime of mEmerald fluorophores linked to FUS significantly decreases over time, indicating FUS condensation and aggregation over time. However, in the presence of DNAJB6, FUS aggregation is prevented, as shown by the significantly higher fluorescence lifetime detected with DNAJB6. Non-parametric Kruskal-Wallis test with Dunn's multiple comparison where * is $P < 0.05$ and **** is $P < 0.0001$. Error bars denote \pm s.d. **d.** Tie line gradient analysis reveals preferential partitioning and interaction between DNAJB6 and FUS. (*left panel*) Intensity time trace of a phase separated FUS sample upon flowing through a confocal detection set-up using microfluidic technology (top) and corresponding intensity histogram (bottom). Bursts of high intensity correspond to condensates passing through the confocal volume. The baseline signal forms a peak in the intensity histogram and is used to extract dilute phase concentrations. For details, see Materials and Methods. (*middle panel*) Dilute phase concentration changes of FUS with varying DNAJB6 concentrations allowing for determination of the tie line gradient value between FUS and DNAJB6 at the measured conditions. (*right panel*) DNAJB6 is recruited into FUS condensates, effectively spacing out aggregation promoting FUS-FUS interactions to induce liquid state preservation.

REVIEWER #4 (REMARKS TO THE AUTHOR):

Resnick et al. develop a yeast-based platform in which they express one of 29 proteins associated with misfolding and neurodegenerative disorders, along with a second protein that may rescue the misfolding. These additional proteins initially include 62 yeast chaperones and 70 human ones, and subsequently ~900 human proteins from the hORFeome library. These many pairings result in 120 interactions called significant from the chaperone set and 54 from the hORFeome set. Although redoing – and extending – an experiment done multiple times over the last two decades, this work is an important accomplishment that allows consistent comparisons of disease-associated proteins and potential modifiers derived from disparate models. Furthermore, the platform employed here leverages redundant barcodes for each pairing to provide additional confidence in the results. Because the platform allows high throughput, Resnick et al. in several cases also screen mutant versions of misfolding proteins that are known to increase the probability of neurodegenerative disease. Finally, the work identifies the human HSP40 chaperone DNAJB6 as a rescuer of multiple RNA-binding proteins that are aggregation prone, and provides analysis of the mechanism by which this chaperone acts.

The results represent a large body of work that is done at a high standard. The platform will be of considerable value throughout the community studying these diseases, and its description and the results from these screens warrant publication in a broad-based journal.

Thank you for your critical reading of our work and for your clear summary of its main points.

My comments are generally minor and mostly pertain to the presentation.

1. The Abstract leaves out key information – e.g. that the platform is yeast-based – that would seem to be helpful to the reader, and emphasizes a large number (“more than a thousand genetic modifiers”) that is not representative of the bulk of the results presented. The Abstract might better state some of the specific results; for example, that the 29 models were tested against 62 yeast and 70 human chaperones, and might provide some indication of what were the key findings besides those related to DNAJB6.

We have updated the manuscript’s abstract with the input of the reviewer, taking into account several of their comments listed below. Furthermore, as Reviewer #1 suggested we now also highlight our human ORFeome screening results and make it a more prominent part of the manuscript. The new abstract based on all reviewer’s feedback is pasted below:

[LINE 55]

Neurodegenerative disorders (NDDs) are a family of diseases that remain poorly treated despite their growing global health burden. To gain insight into the mechanisms and modulators of neurodegeneration, we developed a yeast-based multiplex genetic screening platform. Using this platform, 32 NDD-associated proteins were probed against a library of 132 molecular chaperones from both yeast and humans, and an unbiased set of ~900 human proteins. We identify both broadly active and specific modifiers of our various cellular models. To illustrate the translatability of this platform, we extensively characterized a potent hit from our screens, the human chaperone DNAJB6. We show that DNAJB6 modifies the toxicity and solubility of multiple amyotrophic lateral sclerosis and frontotemporal dementia (ALS/FTD)-linked RNA-binding proteins (RBPs). Biophysical examination of DNAJB6 demonstrated that it co-phase separates with, and alters the behavior of FUS containing condensates by locking them into a loose gel-like state which prevents their fibrilization. Domain mapping and a deep mutational scan of DNAJB6 revealed key residues required for its activity and identified novel variants with enhanced activity. Finally, we show that

overexpression of DNAJB6 prevents motor neuron loss and the associated microglia activation in a mouse model of FUS-ALS.

2. I could not find a list of what the 29 proteins are that undergo misfolding in neurodegeneration and that form the basis for the work. This list could be within Fig. 1a, but this panel shows 24 proteins associated with disease, 2 yeast proteins not associated with neurodegeneration, and 9 genetic variants of the 24 disease proteins.

We thank the reviewer for bringing this to our attention. We attempted to use an oversimplification for how we referred to models included in the pool, but a granular description of the models examined is best. We have taken several steps to be clearer about the models. While most of the models examined are relevant to NDD, with some being thought to cause toxicity via misfolding (e.g. FUS, TDP-43) for others it is uncertain if their toxicity arises from misfolding or perturbing other cellular pathways such as proteosomal degradation (e.g. UBQLN2). First, we have added a description column to **Supplementary Table 1** which has key information about the models in the pool. Second, we have updated the abstract to better reflect the models screened. Third, we have updated the main text with more accurate descriptions of our pool. Fourth, we have expanded **Supplementary Note 3** with a description of the different classes of models in our pool. A summary table of the types of models and their number in the pool is included below.

NDD Protein	23
NDD Protein - Genetic Variant	9
Non-NDD Previously Published Poly Alanine Model	4
Yeast Prion	2
Non-NDD-Associated Toxic Protein	5
Non-Toxic Controls	2

[LINE 149]

In the main text:

In total, our final screening library contains 32 NDD-associated proteins plus multiple controls, each placed within 5-7 uniquely DNA-barcoded strains and mixed at equal ratios into a single pool for testing (For detailed descriptions of models and pool composition, see Sup. Table 1 and Sup. Note 3).

[LINE 2210]

From Supplementary Note 3:

For each model, we assembled 5-7 individual barcoded strains and validated equal growth between isogenic strains containing the same model but different barcodes (Sup. Table 1). The final pool is comprised of 45 total models. Twenty-three models represent human NDD-associated proteins. An additional 9 are clinical variants of disease-associated proteins that lead to an increase in the incidence of disease. The remaining models represent poly-alanine repeat expansion models (4), yeast prions (2), Non-NDD-associated proteins that cause growth defects when expressed and serve as assay controls (5), and finally non-toxic controls (2).

3. 29 proteins are tested against 132 chaperones, which makes 3,828 combinations, plus controls. The manuscript cites 5,850 interactions (line 137) by lumping together models and

controls, which are not clearly defined. There should be more precision in the descriptions of the controls and the pairings.

We have added new language when introducing the results of the chaperone screen to better capture the number of each type of interaction tested. The updated section is pasted below for convenience:

[LINE 162]

Overall, this screen represents 5,850 genetic interactions between models in the pool and the corresponding library of molecular chaperones, including 4,224 interactions directly relevant to human neurodegenerative disease models.

4. It is not obvious to me why the hORFeome data are included, other than to point to the potentially large scale (“~35,000 interactions”) of the platform. This experiment is described in a single paragraph on p. 8 (lines 202-215), and the 54 confirmed interactions are not discussed beyond a list of entries in Supp. Tables. These data should either be dealt with in a substantive manner or deleted.

The reviewer brings up a good point. We have added a section and more in-depth discussion on the results of this screen including figures representing validation in yeast. Below are the additional text and figure related to these data.

[LINE 233]

A large, unbiased screen of human ORFs identifies novel interactions

To evaluate the performance of our platform against a larger, unbiased library of potential modifiers, we screened the hORFeome V8.1 human cDNA library against our pool. We chose to screen human ORFs not only to identify hits that are more likely to translate directly to mammalian models of disease, but also because few human genes had been tested before in this paradigm, suggesting that many novel interactions remain uncovered. Screening 899 members of the hORFeome library enabled the examination of 40,455 interactions, representing ~9 times more interactions surveyed within our system than is typical^{7,8,11,14,15}. Screening and subsequent validation of this collection of rescuers resulted in the identification of 54 confirmed genetic interactions (Sup. Tables 5-7). As expected with a library comprising random human genes, the occurrence of genetic interactions was significantly lower (0.14%) compared to the curated molecular chaperone screen (2.9%). Furthermore, when compared to other unbiased screens overexpressing yeast ORFs instead of human genes our hit rate remains lower (previously reported hit rates of 0.24%-1.13% vs. 0.14%)^{8,10,11,14}. One of the main drivers of this difference is likely the failure of human genes to function in yeast cells, highlighting the benefits of using a rapid high-throughput, quantitative screening approach to explore this sparsely populated interaction space³⁷.

Despite a lower rate of interactions, rescuers were identified for a broad range of yeast models, including ones that did not have rescuers from the initial molecular chaperone screen (Sup. Fig. 13a). Several hits captured known interactions. For example, two 14-3-3 proteins, YWHAG and YWHAB were found to be rescuers of the ATXN1 polyglutamine (ATXN1-Q85) yeast model (Sup. Fig. 13b). Cytoplasmic 14-3-3 proteins bind ATXN1 via a motif centered around a phosphorylated serine at position 776 in ATXN1. This binding masks the ATXN1 NLS and diminishes its transport into the nucleus which is necessary for ATXN1 toxicity^{38,39}. Other rescuers of ATXN1-Q85 in our screen were RNPS1 and JMJD6, two proteins involved in RNA-splicing, which ATXN1 also actively participates in (Sup. Fig. 13b). Furthermore, both RNPS1 and JMJD6 were recently

identified as interactors of ATXN1-Q85 via pulldown and BIO-ID approaches, respectively⁴⁰. We also identified two rescuers of the alpha-synuclein expressing model, one of which is a member of the Regulators of G protein signaling (RGS) family, RGS20 (Sup. Fig. 13c). RGS family proteins are structural proteins that accelerate the GTPase activity of α -subunits in G coupled-protein receptors, reducing the duration of downstream signaling⁴¹. A number of RGS family proteins have been implicated in the development of Parkinson's Disease including a study of RGS6 in which expression of RGS6 in substantia nigra pars compacta dopaminergic neurons suppressed Parkinson's Disease phenotypes in aged mice⁴². Within our larger screen, we also discovered several rescuers of the FUS yeast model (Sup. Fig. 13d). These rescuers included RAD23B, a ubiquitin-binding proteasome-shuttle protein localized to the nucleus that is suspected to undergo phase separation, HIST1H1E, a member of the H1 linker histone family with predicted disordered regions, TGIF2LX, a DNA-binding homeobox protein with predicted disordered regions, and OAS1, a stress granule localized 2'5'-oligoadenylate synthetase protein. TMEM106B, a protein implicated in FTD⁴³⁻⁴⁶ and brain aging^{47,48} was strongly rescued by COX7B, a member of the cytochrome c oxidase complex involved in the mitochondrial respiratory chain (Sup. Fig. 13e)^{49,50}. Taken together, these results suggest that human proteins beyond molecular chaperones can be screened to identify novel interactions.

Supplementary Figure 13. Identification and validation of interactions from the hORFeome screen. **a.** Log₂ fold change interactions between a select set of hits from the ORFeome screen and the models included in the pool. Manual validation in yeast of detected interactions for **b.** ATXN1-Q85 **c.** Alpha-synuclein **d.** FUS and **e.** TMEM106B. Data are shown as mean \pm s.d. for three biological replicates, independent outgrowths. Comparisons for **b.** and **d.** were conducted with ordinary one-way ANOVA and comparisons **c.** and **d.** were conducted with Welch's t test; * $P < 0.05$, ** $P < 0.01$, **** $P < 0.0001$.

5. Fig. 2 shows gels from a protein aggregation assay to compare the amount of SDS-insoluble

species that are solubilized in urea with or without DNAJB6. The manuscript text (lines 234-241) and figure legend are sparse about the details of these experiments. What is the high molecular weight material in Fig. 2b, g and is it included in this comparison? Where is the Ponceau S staining for 2b, c, d and 2g, h, i that is used for normalization? Why are full gel lanes not shown for 2c, f, h, I, rather than only limited slices of the gels? This is an important experiment to validate the DNAJB6 rescue, and either Fig. 2 or a supplemental figure should show a complete depiction of what was found, including whole gels.

The reviewer points out an extremely important point regarding western blots. We regret not including full gels in our initial submission. We have included 6 new supplementary figures showing the full gels and Ponceau S staining for our blots, including those from 2b-d and 2g-i (**Supplementary Figures 15 and 19**, respectively - for convenience these full blots are shown at the end of this response.

The comment regarding the HMW species seen in the TDP-43 blots was also brought up by reviewer #1 and is a good point. This assay was developed based on a prior published protocol from the following paper in which HMW species of TDP-43 were also identified upon overexpression in mammalian cells:

Chen, H. J. *et al.* The heat shock response plays an important role in TDP-43 clearance: evidence for dysfunction in amyotrophic lateral sclerosis. *Brain* **139**, 1417-1432, doi:10.1093/brain/aww028 (2016)

This paper quantifies the abundance of HMW species which were observed at the same molecular weights in our study. Given this precedent, we also quantify all TDP-43 present in the urea lanes in our study. We have updated the methods and figure legends with this information and excerpts are pasted below:

[LINE 1755]

In Figure 2 legend, the following statement has been added, **For TDP-43 blots, all bands at or above 43 kDa were included in quantification.**

[LINE 954]

In the methods, **For TDP-43 blots, high molecular weight (HMW) species were included in quantification of TDP-43 blots.**

Supplementary Figure 15. Full unprocessed blots for DNAJB6 overexpression experiments. All individual replicate blots for FUS, TDP-43, and hnRNPA1 experiments as displayed in Figure 2b, 2c, and 2d including both antibody staining (left) and total protein Ponceau S staining (right). Red rectangles outline relevant portions of blots. All blot lanes are in the same order as presented in Figure 2b-d.

Supplementary Figure 19. Full unprocessed blots for DNAJB6 KO experiments. All individual replicate blots for FUS, TDP-43, and hnRNPA1 overexpression in two DNAJB6 KO lines and two corresponding NTC lines as displayed in Figure 2g, 2h, and 2i including both antibody staining (left) and corresponding Ponceau S staining for total protein measurement (right). All blot lanes are in the same order as presented in Figure 2g-l with relevant lanes outlined by a red rectangle.

6. Sup. Fig. 13 shows that DNAJB6 was upregulated when FUS or TDP-43 was overexpressed. What are the other RNAs that are even more upregulated than DNAJB6?

We have added these data as a supplementary table – **Supplementary Table 8** - and included both the chaperone subset (Supplementary Table 8a and 8c) used to create the plot and all genes that were detected (Supplementary Table 8b and 8d). While many genes were differentially expressed (adjusted p-value of <0.0001), we focused our analysis on the subset of 273 chaperones as this seemed most relevant regarding our gene of interest, DNAJB6. There were 16 chaperone genes from our analysis that were upregulated more than DNAJB6, including DNAJA1, PFDN2, DNAJC25, HSP90AA1, TTC1, TXDNC12, SUGT1, AHSA1, CCT6A, DNAJC12, GREPL1, HSPA6, CRYAB, HSPA7, RAPSN, and SERPINH1 upon either FUS or TDP-43 overexpression. Furthermore, when looking at the 25 chaperones in each condition with the greatest increases in RNA expression, 6 overlap across the FUS and TDP-43 conditions. Among these 6 shared chaperones, DNAJB6 shows the most consistent increases in expression across both data sets (ranked 12th and 10th in the FUS and TDP-43 datasets, respectively).

7. Sup. Fig. 17a shows that deletion of domains from DNAJB6 prevents the protein from rescuing FUS toxicity in yeast. Why are the comparisons between growth in the presence of the deletion vs. mCherry, rather than growth in the presence of the deletion vs. the intact DNAJB6? Also, line 334 points to a result in Sup. Fig. 17b that should refer to Sup. Fig. 17a.

Thank you for pointing these two things out in the manuscript. Given that the appropriate null hypothesis for this experiment is that DNAJB6 domain deletions and mutations do not perform differently from the WT DNAJB6, the analysis presented in our current **Supplementary Figure 24** is not the most relevant presentation of the data. We have replaced the figure with comparisons between WT DNAJB6 and other constructs tested (see modified **Supplementary Figure 24** below). We did choose to leave the relative growth as normalized to the mCherry as we believe this more clearly shows that some domain deletions or point mutations are essentially inert while others exert a gain of function effect. Additionally, we have fixed the incorrect reference that was on line 334 in the original submitted manuscript.

Supplementary Figure 24. Identification of DNAJB6 domains important for activity in yeast. a-b. Testing domain deletions and J domain H31Q loss of function point mutant for their ability to rescue the FUS expressing yeast model. ΔJ , $\Delta G/F$, ΔS represent deletion of the J-domain, glycine-phenylalanine rich, or serine rich region of DNAJB6, respectively. Comparisons were conducted with ordinary one-way ANOVA; **** $P < 0.0001$. **c.** Uncropped western blots for data shown in panel b.

REVIEWERS' COMMENTS

Reviewer #1 (Remarks to the Author):

In the revised manuscript, Resnick and co-authors have adequately addressed my previous comments and concerns, making the manuscript suitable for publication. The additional data and reworded text further strengthen and clarify the conclusions.

We thank the reviewers for seeing the value of our revised manuscript.

Reviewer #2 (Remarks to the Author):

The authors of, "Multiplex Neurodegeneration Proteotoxicity Platform Reveals DNAJB6 Promotes non-toxic FUS Condensate Gelation and Inhibits Neurotoxicity" have addressed many of my previous concerns to a satisfactory degree. The authors provided the data I had asked for in supplementary tables and figures. They were able to answer my question regarding the difference between observed L2FC from the initial screening platform and individual validation in subsequent assays and justified their investigation of DNAJB6 with many additional experiments. These experiments addressed my concerns regarding the validity of DNAJB6 effects, but also led to a few minor questions that the authors may choose to address:

We thank the reviewer for their initial comments and we provide answers to their minor set of inquiries below.

1. It appears that the results from the screening platform designed by the authors require additional optimization steps before identifying strong candidates. Are the authors interested in further developing their screening platform? Do they have any ideas on how to improve this platform to make it more applicable for widespread laboratory use?

At present we plan to make use of the platform as originally described with slight modifications. These changes are that we have streamlined the sequencing library preparation from a 2-stage PCR to a single round of PCR. In addition, we are investigating if it is possible to modify our hit calling pipeline to further improve its sensitivity. Finally, while the approach may seem challenging it makes use of methods that are common to any group that has experience with sequencing library preparation and is much less cumbersome than the multistage protocols required for procedures such as CHIP-seq, RNA-seq, or ATAC-seq, as all that is required is to perform a single round of PCR on the extracted DNA.

2. The results of aging condensates (either FUS alone or FUS + DNAJB6) are reported only up to 48 hours for all experiments. What happens beyond that timepoint? Is there a reason why data collection stops after two days?

The analytical time points were set based upon the times at which liquid:liquid droplets, gels and hardened fibrillar aggregates were sufficiently abundant that they could be assayed by direct inspection, pFTAA dye fluorescence, Fluorescence Life-time Imaging Microscopy, Atomic Force Microscopy, and nano-Infrared Spectroscopy. We have not formally assessed aggregates beyond 72 hours. However, casual inspection suggests that gels remain gels and aggregates remain aggregates.

3. Do the authors propose DNAJB6 as a potential therapeutic for neurodegeneration related to FUS toxicity? Have the authors investigated the effects of overexpression of DNAJB6 in neuronal cell models?

We have not tested the effects of DNAJB6 overexpression within neuronal cell models, it is an area we are interested in exploring but is beyond the scope of this current manuscript.

4. The bulk of biophysical characterization (AFM-IR, FLIM, etc) data study DNAJB6 and FUS interaction. What is the interaction between DNAJB6 and TDP-43/hnRNPA1? Are the gelation effects observed between DNAJB6 and FUS observed with TDP-43 and hnRNPA1?

We appreciate the reviewer's interest in pursuing additional biophysical analyses on other ALS-associated RBPs. Indeed, we agree that it would be valuable to explore the effects of DNAJB6 on TDP-43 and hnRNPA1. However given the wealth of data already provided and the time required to perform these deep biophysical studies, the proposed studies we believe are best to be performed as part of a follow-up study.

Reviewer #4 (Remarks to the Author):

The authors have done a careful and thorough job of addressing the concerns of the referees, and where they have declined to perform additional experimentation they provide credible rationales for doing so. They also generated new data in a mouse model that bolster the validity of the yeast screens. I have no additional concerns.

We thank the reviewer for their previous comments which helped improve the rigor of our manuscript.

Reviewer #5 (Remarks to the Author):

NCOMMS-22-42799A

This study presents a novel yeast genetic screening platform to identify modifiers of neurodegeneration-linked proteins, focusing on 32 NDD-associated targets. Key findings highlight the human chaperone DNAJB6 as a potent suppressor of toxicity in ALS/FTD models. The authors demonstrate DNAJB6's ability to alter phase separation dynamics of FUS protein condensates, preventing fibrilization, and validate its neuroprotective effects in a FUS-ALS mouse model.

Figure 8 & Supplementary Figure 31 Assessment

Figure 8: Robustly illustrates DNAJB6's in vivo efficacy, showing prevention of motor neuron loss and reduced microglia activation in FUS-ALS mice. Data presentation is clear, with appropriate controls and statistical rigor.

Supplementary Figure 31. Validation of DNAJB6 expression upon exposure to ChAT-Cre. Data presentation is clear, with appropriate controls and statistical rigor.

We thank the reviewer for agreeing that our results suggest a clear effect of DNAJB6 on modulating FUS toxicity within a FUS-ALS mouse model.